

# Parameterization of the spectral light absorption coefficient of phytoplankton in the Baltic Sea: general, monthly and two-component variants of approximation formulas

Justyna Meler[1], Sławomir B. Woźniak[1], Joanna Stoń-Egiert[1], Bogdan Woźniak[1†]

[1]Institute of Oceanology, Polish Academy of Sciences, Powstańców Warszawy 55, 81-712 Sopot, Poland

*Correspondence to*: Justyna Meler (jmeler@iopan.pl)

**Abstract.** The paper presents approximate formulas for parameterizing the coefficient of light absorption by phytoplankton $a_{ph}(\lambda)$ in Baltic Sea surface waters. Over a thousand absorption spectra (in the 350-750 nm range), recorded during nine

10 years of research carried out in different months of the year and in various regions of the southern and central Baltic, were used to derive these parameterizations. The analysed empirical material was characterized by a wide range of variability: the total chlorophyll *a* concentration (*Tchla*) varied between 0.3 and > 140 mg m$^{-3}$, the ratio of the sum of all accessory pigment concentrations to chlorophyll *a* ($\Sigma C_i$/*Tchla*) varied between 0.21 and 1.5, and the absorption coefficients $a_{ph}(\lambda)$ at individual light wavelengths varied over almost three orders of magnitude. Different versions of the parameterization formulas were

15 derived on the basis of the data gathered: a one-component parameterization in the "classic" form of a power function with *Tchla* as the only variable, and a two-component formula – the product of the power and exponential functions, with *Tchla* and $\Sigma C_i$/*Tchla* as variables. There were distinct differences between the general version of the one-component parameterization and its variants developed for individual months of the year. In contrast to the general parameterization, both the monthly and the two-component variants enable, at least partially, the variability of pigment composition occurring

throughout the year in the Baltic phytoplankton populations examined here to be taken into account.

## 1 Introduction

If we wish to fully describe photosynthesis in the seas and oceans, and to correctly interpret remote observations of water bodies, it is important to obtain an accurate quantitative description of the spectral characteristics of light absorption by living phytoplankton, a significant constituent of seawater (see e.g. Kirk 1994, Mobley 1994, or Woźniak and Dera 2007).

The efficiency of sunlight absorption by this phytoplankton generally depends on a number of factors. The principal, strongly absorbing components of phytoplankton are the pigments it contains: chlorophyll *a*, the basic photosynthetic pigment, and accessory – photosynthetic and/or photoprotective – pigments. Light absorption by phytoplankton thus depends primarily on the properties and concentrations of these several pigments. It is known *inter alia* from theoretical considerations that the very structure, dimensions and shapes of individual phytoplankton cells can also influence the



efficiency with which their populations absorb sunlight. It is important to know by what means and how densely the strongly light-absorbing pigments are "packed" within the internal structures of such cells (see e.g. Morel and Bricaud 1981, 1986). Given the complexity of these all relationships, it is often necessary (or even required) for practical purposes to take a highly simplified approach, for example, when constructing models of bio-optical processes for interpreting remote sea

observations. It is a common simplification to assume that all the relevant properties of a phytoplankton population can be parameterized using just one variable – the concentration of chlorophyll *a*, its main pigment: indeed, the total biomass of an entire phytoplankton population as well as its diverse optical properties are often parameterized in this way. Earlier authors addressing this question applied this kind of simplification in attempts to determine typical values of the "chlorophyll *a*-specific" absorption coefficient (defined as the light absorption coefficient of phytoplankton normalized to the chlorophyll *a*

concentration). In practice, therefore, the adoption of one averaged value of this coefficient should enable the relationship between the light absorption coefficient of phytoplankton and the chlorophyll *a* concentration in seawater to be described using the simplest possible, i.e. linear, functional relationship. As measured in nature, however, values of the specific absorption coefficient have proved to be extremely variable. The papers by Bricaud et al. (1995, 1998), often cited by other authors, were among the first to introduce for practical purposes a different, non-linear, approximate description of the light

absorption vs chlorophyll *a* concentration relationship. They proposed using a power function to account for the general decrease in light absorption efficiency per unit chlorophyll *a* concentration that occurs with increasing absolute values of this concentration in seawater. Bricaud et al. (1995) also gave a theoretical explanation of these effects, suggesting that there might be a correlation between the increase in the absolute chlorophyll *a* concentration and the increasing contribution of the pigment package effect and, concurrently, the decreasing proportion of pigments other than chlorophyll *a*. The papers by

Bricaud et al. (1995, 1998) were based on empirical material gathered in open, oceanic waters, often classified as "case 1"waters (in those authors' original dataset, the chlorophyll *a* concentration varied from ca 0.02 to 25 mg m$^{-3}$). Among later works addressing the same problem but in relation to other natural water bodies (examples of "case 2" waters, i.e. coastal waters of enclosed and semi-enclosed seas, and inland waters), there are several alternative examples of spectral parameterizations using the same form of the power function, but with noticeably different, empirically derived coefficients.

Examples of parameterizations derived for marine environments can be found, for example, in Staehr and Markager (2004), Dmitriev et al. (2009) and Churilova et al. (2017). The first of these papers was based on a set of data from marine areas ranging from estuarine, through coastal, to open Atlantic waters, with chlorophyll *a* concentrations from 0.03 to 88.1 mg m$^{-3}$. The other two papers analysed datasets gathered in the Black Sea: these exhibited a relatively small range of chlorophyll *a* variability – from 0.15 to 2.04 mg m$^{-3}$. Churilova et al. (2017) drew attention to the differences between parameterization

coefficients obtained separately for summer and winter. An earlier work by our research team (Meler et al. 2017a) provides alternative parameterization coefficients of a power function adjusted to our own data collected in the southern Baltic Sea. The subject literature also provides examples of parameterizations derived for inland water bodies, for example, by Reinart et al. (2004), Ficek et al. (2012a, b), Ylöstalo et al. (2014) and Paavel et al. (2016). All of these papers give spectral coefficients of parameterizations tailored to specific datasets differing from each other to a greater or lesser extent.



Obviously, all such parameterizations are far-reaching simplifications of the complex dependences observed in nature. That there might be significant deviations from the approximate "average" relationship was already made clear by Bricaud et al. (1995) in their original work; these authors subsequently analysed the potential causes of this differentiation (Bricaud et al. 2004). Using indirectly reconstructed information regarding the size structure of the ocean phytoplankton population, they were able to estimate separately the impacts of the differences in the dominant sizes of plankton populations and the differences in pigment composition on the relationship in question. In general, they found that for oligo- and mesotrophic waters (i.e. waters with chlorophyll $a$ concentrations <2 mg m$^{-3}$), the variability associated with the package effect might be exerting a more significant influence, while in eutrophic waters (with higher chlorophyll $a$ concentrations) both effects might be of equal weight. Generally, however, the observed variability indicates that one should expect both regionally and seasonally differentiated forms of such simplified relationships to occur instead of one universal, approximate statistical relationship between $a_{ph}(\lambda)$ and the chlorophyll $a$ concentration.

The Baltic Sea is a semi-enclosed, brackish sea basin classified as "case 2". It is characterized by usually high concentrations of dissolved organic substances of terrestrial origin. In addition to the paper by Meler et al. (2017a), which gives one of the earliest versions of the "local" parameterization of light absorption, our team has recently further documented in somewhat greater detail other properties of the phytoplankton inhabiting this particular environment. With a comprehensive set of measurement data to hand, successively gathered over the last ten years, we have been able to identify significant differences in the absorption properties of Baltic phytoplankton at different times of the year (see Meler et al. 2016b). Moreover, although our analyses have so far been limited to just a single light wavelength (440 nm), we have been able to demonstrate differences between the coefficients of the relevant simplified parameterizations when they have been tailored to data gathered at specific times of the year (see Meler et al. 2017b). It is in this context, therefore that we have decided in the present paper to re-address the problem of determining practical forms of a simplified parameterization of the phytoplankton light absorption coefficient appropriate to Baltic Sea conditions.

The main objective of this work is to perform the relevant analyses, this time over a wide spectral range, with a sufficiently high resolution and in accordance with the latest recommended calculation procedures, in order to find new forms of parameterizations for the phytoplankton light absorption coefficient adapted to the specific conditions of the Baltic Sea. An important aspect of this (main) objective is to check whether and how large the differences between the coefficients of this type of spectral parameterization may be if they are matched separately for data from selected periods of the year. An additional aim is, if possible, to attempt a modified, but still relatively simple and practical form of parameterization that would at least partially explain the diversity of phytoplankton absorption properties observed during the year. The new forms of parameterization that we are seeking can be used, among other things, to develop and improve the accuracy of practical, local algorithms for interpreting remote observations of the Baltic Sea.





## 2 Materials and methods

The empirical data used in this study were collected at more than 170 measuring stations in various parts of the southern and central Baltic Sea, though mainly in the Polish economic zone, from 2006 to 2014 (see Figure 1). These data were acquired principally during 42 short research cruises on board r/v Oceania at different times of the year, but mostly from March to

May and from September to October (about 80% of the data analysed in this paper are from these periods). The practice during each cruise was to select measuring stations that were maximally diverse with respect to their optical properties, i.e. in the vicinity of river mouths and estuaries (the Rivers Vistula, Oder, Reda, Łeba and Świna; the Szczecin Lagoon), bays and offshore waters (Gulf of Gdańsk, Puck Bay and Pomeranian Bay), and open southern Baltic waters. During three cruises (in May of 2010, 2012 and 2014), measurements were also made in the open waters of the central Baltic. Because of

weather- and sea-state-related limitations, the proportion of data collected in open water regions exceeds ca 30% only for data collected in February, March, May, September and October (see Table 1). In addition to the cruise measurements, data were gathered throughout the year by sampling the seawater at the end of the ca 400 m long pier in Sopot, on the Gulf of Gdansk coast (<7% of the overall number of data analysed).

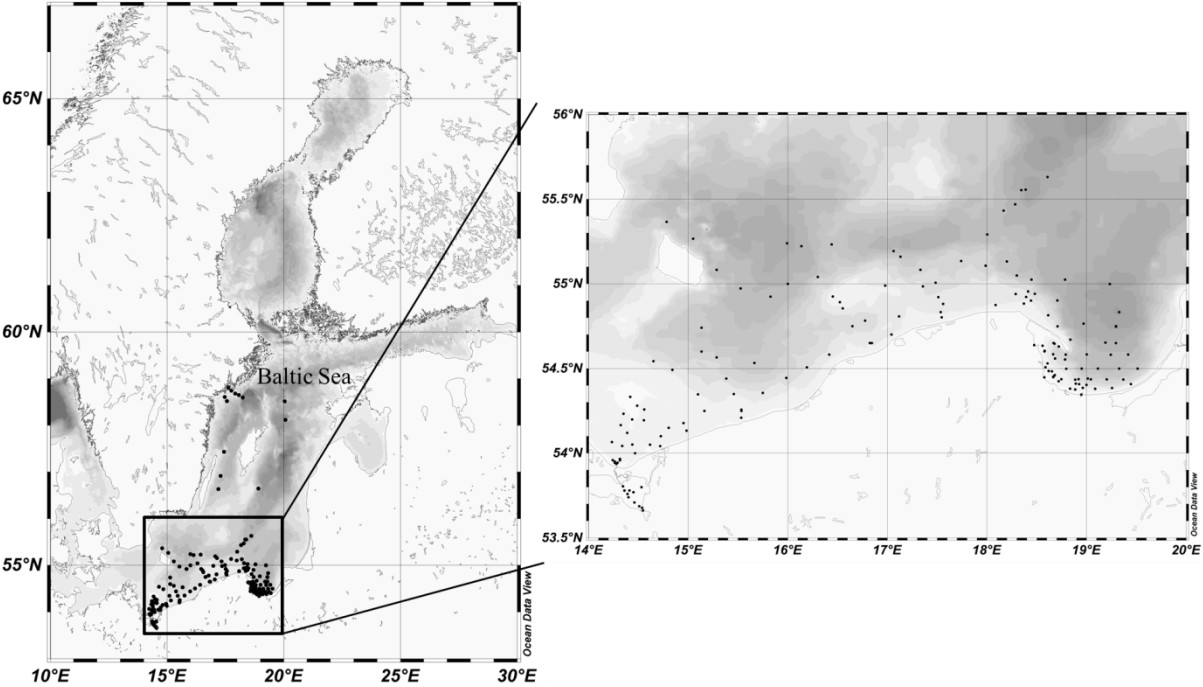

**Figure 1.** Locations of all sampling stations in the Baltic Sea; enlargement of the southern Baltic Sea area.



**Table 1.** Amount of analysed seawater samples divided into months and areas of their acquisition.

| Time of the year | Open Baltic Sea areas | Coastal areas and bays | All regions |
|---|---|---|---|
| January | 4 | 6 | 10 |
| February | 16 | 36 | 52 |
| March | 52 | 122 | 174 |
| April | 22 | 74 | 96 |
| May | 80 | 121 | 201 |
| June | - | 5 | 5 |
| July | - | 7 | 7 |
| August | 4 | 21 | 25 |
| September | 75 | 136 | 211 |
| October | 51 | 101 | 152 |
| November | 15 | 42 | 57 |
| December | 1 | 11 | 12 |
| All year | 320 | 682 | 1002 |

Some of the data used in this work have already been used by the authors before. The analysis in Meler et al. (2017a) was based on ca 400 of the more than 1000 light absorption coefficient spectra used here, but the methods of calculation differed in some important aspects (see the description of the measurement procedures below). Some of these data were also handled by Meler et al.(2016b, 2017b), but those analyses referred to just a single light wavelength (440 nm).

During the research cruises, a diversity of physical and optical parameters of seawater were measured *in situ* at each sampling station. Discrete seawater samples for further laboratory analysis were also collected. This work analyses certain optical properties (spectra of coefficients of light absorption by phytoplankton) and biogeochemical properties (concentrations of chlorophyll *a* and other phytoplankton pigments) only of discrete surface seawater samples. These were collected with a Niskin bottle (height ca 0.9m,capacity 25L) immersed just below the surface. In shallow estuarine areas and river mouths (sampled from a pontoon) or off the end of the Sopot Pier, the surface water was collected using a bucket. Immediately after collection, all seawater samples were passed through glass fibre filters (Whatman, GF/F, 25 mm) at a pressure not exceeding 0.4 atm. The filters to be used for determining either absorption properties or phytoplankton pigment concentrations were immersed in a Dewar flask containing liquid nitrogen (at about -196°C) and then kept deep frozen (at about -80°C) for further analysis in the laboratory on land.

The spectra of light absorption coefficients for all suspended particles $a_p(\lambda)$ were measured in the 350-750 nm spectral range with a UNICAM UV4-100 double-beam spectrophotometer equipped with an integrating sphere of external



diameter 66 mm (LABSPHERE RSA-UC-40); the methodology described by Tassan and Ferrari (1995, 2002) was applied. The optical density of the suspended matter collected on the filter $OD_s(\lambda)$ was determined from these measurements and the subsequent calculations. For the reference measurements we used clean filters rinsed in seawater previously passed through GF/F filters (Whatman) and then through 0.2 μm membrane filters (Sartorius) to remove suspended particles. In order to

calculate absorption coefficients on the basis of $OD_s(\lambda)$, an appropriate correction has to be made to compensate for the elongation of the optical path of the light due to the multiple scattering occurring in the material collected on the filter. This is done by applying the dimensionless path length amplification, the β-factor, which converts the measured optical density of particles collected on the filter ($OD_s(\lambda)$) into the optical density characterizing these particles in solution ($OD_{sus}(\lambda)$) (Mitchell 1990). In our analyses we used the new β-factor formula proposed by Stramski et al. (2015) for the T-R method:

$$OD_{sus}(\lambda) = 0.719 OD_s^{1.2287}. \tag{1}$$

The coefficient of light absorption by all suspended particles was then calculated using the formula:

$$a_p(\lambda) = \frac{\ln(10) OD_{sus}(\lambda)}{l}, \tag{2}$$

where $l$ [m] is the hypothetical optical path in solution, determined as the ratio of the volume of filtered water to the effective area of the filter. Note that the use of Eq. (1) is a significant change compared to the previous papers by Meler et al. (2016b,

2017 a and b).

Later, the coefficients of light absorption by non-algal particles $a_{NAP}(\lambda)$ were also determined from results of analogous measurements performed for filters with suspended particles after the phytoplankton pigments had been bleached with a 2% solution of calcium hypochlorite $Ca(ClO)_2$ (Koblentz-Mishke et al. 1995, Woźniak et al. 1999).

The reproducibility of the individual spectral scans performed using our spectrophotometric setup was generally

good. For example, the spectrally averaged value of the coefficient of variation (CV) calculated for 10 consecutive scans of the same clean filter was about 1.95% (± standard deviation of 1.86%). Nevertheless, all the measured particle absorption spectra always carried the signs of accidental "noise" within the measuring system. To eliminate this, we applied a "spectral smoothing" procedure (the spectral 5-point "moving average" repeated 3 times) to the spectra of both coefficients $a_p$ and $a_{NAP}$.

Finally, the sought-after coefficient of light absorption by phytoplankton $a_{ph}(\lambda)$ was determined from the difference between the previously calculated coefficients $a_p(\lambda)$ and $a_{NAP}(\lambda)$, using the value of $a_{ph}(\lambda)$ at wavelength 750 nm for the null-point correction (Mitchell et al. 2002).

HPLC was used to determine phytoplankton pigment concentrations; the methodology is described in detail in Meler et al. (2017b), Stoń and Kosakowska (2002) and Stoń-Egiert and Kosakowska (2005). In this work we refer mainly to

the total chlorophyll $a$ concentration ($Tchla$) (defined as the sum of chlorophyll $a$, allomer and epimer, chlorophyllide $a$ and phaeophytin $a$), and to the sum of the concentrations of all accessory pigments $\Sigma C_i$, i.e. the sum of chlorophylls $b$ ($Tchlb$), chlorophylls $c$ ($Tchlc$), photosynthetic carotenoids ($PSC$) and photoprotective carotenoids ($PPC$).





The data were analysed statistically (using either spreadsheet software (Excel) or statistical package and data visualization software (SigmaPlot)) in order to characterize their variability and to find approximate empirical relationships between them. The variability of the target optical and biogeochemical quantities ranged over almost three orders of magnitude. Therefore, to assess the uncertainty of the empirical parameterizations, we applied separate statistics to the logarithmically transformed data (so-called logarithmic statistics) in addition to the standard arithmetic statistical metrics. This generally yields a more appropriate and fuller description of the accuracy of approximate empirical relationships. The following arithmetic and logarithmic statistical metrics were used:

- relative mean error (representing the systematic error according to arithmetic statistics):

$$\langle \varepsilon \rangle = N^{-1} \sum_i \varepsilon_i, \tag{3}$$

where $\varepsilon_i = (X_{i,cal} - X_{i,m})/X_{i,m}$; $X_{i,m}$ - measured values; $X_{i,cal}$ - estimated values;

- the standard deviation of $\varepsilon$, often referred to as the root mean square error (RMSE) (representing the statistical error according to arithmetic statistics):

$$\sigma_\varepsilon = \sqrt{\frac{1}{N} \left( \sum (\varepsilon_i - \langle \varepsilon \rangle)^2 \right)}; \tag{4}$$

- the mean logarithmic error (representing the systematic error according to logarithmic statistics):

$$\langle \varepsilon \rangle_g = 10^{\left\langle \log\left(\frac{X_{i,cal}}{X_{i,m}}\right)\right\rangle} - 1, \tag{5}$$

where $\left\langle \log\left(\frac{X_{i,cal}}{X_{i,m}}\right)\right\rangle$ is the mean of $\log\left(\frac{X_{i,cal}}{X_{i,m}}\right)$;

- the standard error factor (the quantity which allows one to calculate the range of statistical errors according to logarithmic statistics):

$$x = 10^{\sigma_{log}}, \tag{6}$$

where $\sigma_{log}$ is the standard deviation of the set $\log\left(\frac{X_{i,cal}}{X_{i,m}}\right)$;

- statistical logarithmic errors (representing the range of statistical errors according to logarithmic statistics):

$$\sigma_- = \frac{1}{x} - 1, \quad \sigma_+ = x - 1. \tag{7}$$

## 3 Results and discussion

### 3.1 General characteristics of the data

Figure 2 shows spectra of coefficient $a_{ph}$ that we recorded in the Baltic Sea (these examples include minimum and maximum $a_{ph}(\lambda)$ values as well as a few intermediate ones). Even though the previously described raw data "smoothing" procedure was applied to these spectra, they still contain some obvious, undesirable artefacts related to the noise occurring in our measurement system, especially in the 350-400 nm and 550-650 nm ranges. In spite of these imperfections, 80% of these spectra exhibit the expected characteristic absorption maxima in both the blue (ca 440 nm) and red (ca 675 nm) bands. Some



of the spectra in our set, however, do not show a significant increase in light absorption with increasing wavelength in the 350-440 nm range: $a_{ph}(400)/a_{ph}(440)$ varied from > 0.95 to as much as 1.42 in 20% of the spectra recorded, mainly near the mouth of the Vistula River, in the Szczecin Lagoon and off the Sopot Pier. The first part of Table 2 presents basic statistical information characterizing the ranges of variation of the light absorption coefficient for selected light wavelengths. In fact, the variability of $a_{ph}(\lambda)$ over the entire spectral range examined, calculated for individual light wavelengths, was almost three orders of magnitude. For blue light, for example, $a_{ph}(440)$ varied from 0.014 m$^{-1}$ (February, open Baltic Sea waters) to 3.85 m$^{-1}$ (May, the Szczecin Lagoon), whereas for the local absorption maximum in the red band, $a_{ph}(675)$ varied from 0.006 m$^{-1}$ (February, open Baltic Sea waters) to 1.74 m$^{-1}$ (September, the mouth of the River Vistula).

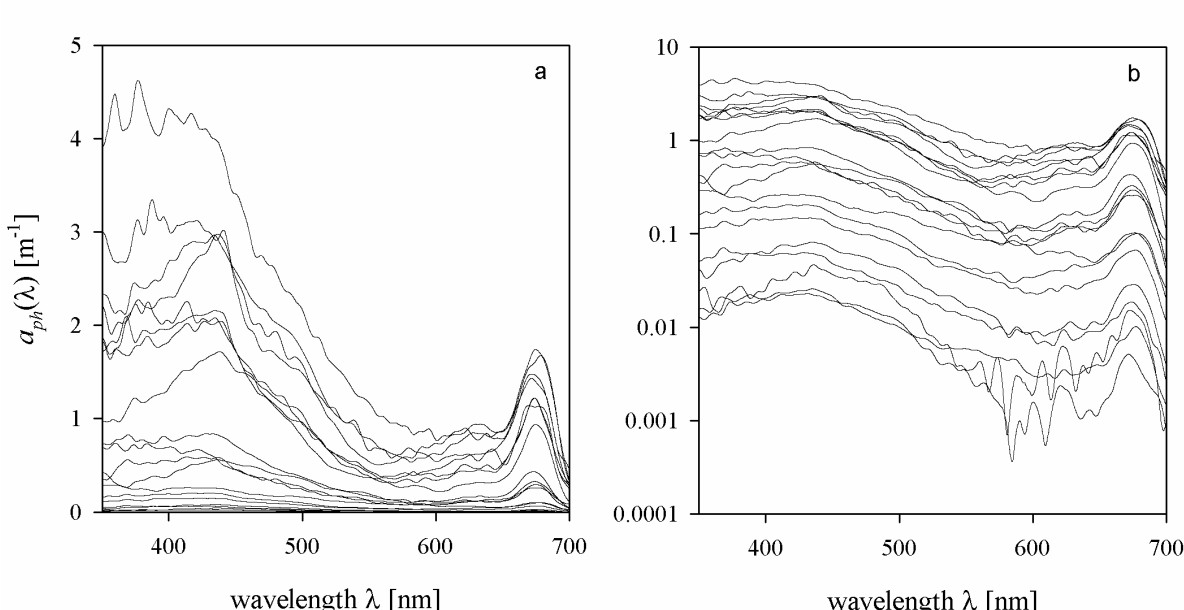

**Figure 2.** (a):Examples of spectra of the light absorption coefficient $a_{ph}(\lambda)$ representing the variability range of the data analysed in the present paper; (b): the same spectra presented on a semi-logarithmic scale.

**Table 2.** Variability ranges of selected optical and biochemical quantities characterizing phytoplankton in samples from the Baltic Sea.

| Quantity | Average value (CV) | Minimum; 10th perc.;**median**; 90th perc.; maximum |
|---|---|---|
| $a_{ph}(350)$ [m$^{-1}$] | 0.222 (166%) | 0.007; 0.033; **0.103**; 0.509; 3.91 |
| $a_{ph}(440)$ [m$^{-1}$] | 0.280 (137%) | 0.014; 0.059; **0.157**; 0.611; 3.85 |
| $a_{ph}(500)$ [m$^{-1}$] | 0.142 (142%) | 0.007; 0.029; **0.080**; 0.313; 2.17 |
| $a_{ph}(550)$ [m$^{-1}$] | 0.065 (145%) | 0.003; 0.012; **0.036**; 0.147; 1.17 |
| $a_{ph}(600)$ [m$^{-1}$] | 0.044 (166%) | 0.002; 0.007; **0.021**; 0.093; 0.772 |





| | | |
|---|---|---|
| $a_{ph}(675)$ [m$^{-1}$] | 0.133 (148%) | 0.006; 0.023; **0.069**; 0.290; 1.74 |
| $a_{ph}(700)$ [m$^{-1}$] | 0.022 (180%) | 0.0001; 0.003; **0.010**; 0.049; 0.459 |
| $Tchla$ [mg m$^{-3}$] | 7.69 (163%) | 0.31; 1.09; **3.63**; 17.92;141.8 |
| $Tchlb$ [mg m$^{-3}$] | 0.50 (175%) | 0; 0.07; **0.27**; 1.01; 11.7 |
| $Tchlc$ [mg m$^{-3}$] | 0.76 (149%) | 0.01; 0.09; **0.35**; 1.93; 14.8 |
| PSC [mg m$^{-3}$] | 1.82 (175%) | 0.03; 0.15; **0.73**; 4.52; 32.1 |
| PPC [mg m$^{-3}$] | 1.70 (160%) | 0.04; 0.26; **0.86**; 3.89; 42.8 |
| $\Sigma C_i$[mg m$^{-3}$] | 4.79 (152%) | 0.15; 0.69; **2.40**; 6.02; 72.1 |
| $\Sigma C_i / Tchla$ | 0.66 (27%) | 0.21; 0.47; **0.62**; 0.88; 1.50 |

        The second part of Table 2 provides statistical information illustrating the variability in concentration of the basic photosynthetic pigment chlorophyll *a* (*Tchla*), and also the levels of different groups of accessory pigments, i.e. chlorophylls *b* and *c*, and other photosynthetic and photoprotective pigments (*Tchlb, Tchlc*, PSC and PPC). In addition, the

table lists the total concentration of all accessory pigments ($\Sigma C_i$) and the ratio of this to *Tchla*. Figure 3 illustrates the variability of *Tchla* and $\Sigma C_i$ as well as their ratio for all the pooled data, broken down into individual sampling periods (months). This shows that with respect to all the data analysed, the ranges of variability of both *Tchla* and $\Sigma C_i$ are, like the absorption coefficient, almost three orders of magnitude (0.41-141.8 mg m$^{-3}$ and0.15-72.1 mg m$^{-3}$ respectively). The average *Tchla* for all the data was 7.69 mg m$^{-3}$. In the spring and summer months when we were able to make measurements at sea,

i.e. in April, May, August and September, mean *Tchla* concentrations were above average (14.40, 9.13, 7.83 and 9.57 mg m$^{-3}$ respectively). In autumn and winter months, however, average *Tchla* were lower: October – 5.08 mg m$^{-3}$, November –2.97 mg m$^{-3}$, December-January –1.75 mg m$^{-3}$, February – 3.98 mg m$^{-3}$ and March –5.83 mg m$^{-3}$. In general, a similar trend of average changes in individual months emerges from an analysis of the sum of accessory pigment concentrations $\Sigma C_i$. Taking into account all the data from different periods of the year, we can say that measured values of $\Sigma C_i$ correlate fairly well with

*Tchla* (see the approximate equation and the coefficient of determination $R^2$ given in Figure 3c). Nevertheless, if we look at the $\Sigma C_i$/*Tchla* ratio, we see that its average values also changed significantly during the year (see Figure 3b). The average value of $\Sigma C_i$/*Tchla* for all the data was 0.66, but the full range of variability that we recorded was from 0.21 to 1.5. For the months of April, May and September, the average $\Sigma C_i$/*Tchla* was higher than or equal to the average for the whole year (0.66, 0.79 and 0.69, respectively). In the remaining months, the averages were lower than the general average –from 0.50 to

0.61. This latter fact is a clear indicator of the obvious limitations of applying solely the chlorophyll *a* concentration as a simplified measure to describe the overall pigment population, and to which measure the light absorption of pigments is customarily parameterized.





**Figure 3.** (a): Box plot presenting the range of variation of chlorophyll *a* concentration (*Tchla*) for all the data analysed and for each sampling month; (b): as (a) but showing the ratio of the sum of accessory pigments to the chlorophyll *a* concentration ($\Sigma C_i/Tchla$); (c): graph illustrating the relationship between $\Sigma C_i$ and *Tchla* – the solid line represents a simple functional approximation of the relationship (the equation is given in the panel).

At this point, it should be noted that the statistical values presented above should not be treated as "universal" values characterizing the Baltic Sea environment. These figures are only a characteristic of the dataset we gathered, in which the



proportion of data from different sea regions varied at different times of the year. However, the ranges of variability that we observed (almost three orders of magnitude) with respect both to the concentrations of *Tchla* or accessory pigments and to the light absorption coefficients are generally consistent with the ranges reported in the Baltic by other researchers (see e.g. Babin et al. 2003). We ourselves had already reported similar ranges of variability earlier (see Woźniak et al. 2011, Meler et al. 2016b, 2017a and b).

### 3.2 Approximate description of the light absorption coefficient by phytoplankton

### 3.2.1 General and monthly variants of classic one-component parameterizations

We carried out statistical analyses of our measurement data in order to define classic forms of the approximate functional relations between the light absorption coefficient $a_{ph}(\lambda)$ and the concentration *Tchla*. Like Bricaud et al. (1995, 1998), we approximated these relations using power functions. With linear regression applied to the logarithms of the input data for each light wavelength (regression between $\log(a_{ph}(\lambda))$ and $\log(Tchla)$), the coefficients $A$ and $E$ of the following approximated parameterization could be calculated:

$$a_{ph}(\lambda) = A(\lambda) \cdot Tchla^{E(\lambda)},\tag{8.a}$$

Note that coefficient $A(\lambda)$ determined in this way reflects the numerical value of the light absorption coefficient $a_{ph}(\lambda)$ which the approximated relationship assigns to the case when the *Tchla* is exactly 1 mg m$^{-3}$.The coefficient $E(\lambda)$ of Eq. (8.a) is a dimensionless quantity, usually less than 1, which is the exponent of the power to which the chlorophyll *a* concentration is raised. By performing linear regression of the logarithms of the input data, we calculated the determination coefficients R$^2$ for the approximated parameterization at the individual wavelengths of light. The parameterization coefficients given by Eq. (8.a) can be easily used to determine the specific coefficient of light absorption by phytoplankton $a_{ph}^{*}(\lambda)$ [m$^2$mg$^{-1}$] (defined as values of $a_{ph}(\lambda)$ normalized with respect to *Tchla*):

$$a_{ph}^{*}(\lambda) = A(\lambda) \cdot Tchla^{E(\lambda)-1}.\tag{8.b}$$

The coefficients of the approximate Eq. (8.a) were determined over the entire available spectral range from 350 to 700 nm with a resolution of 1 nm. Figure 4 presents different variants of the spectra of coefficients $A(\lambda)$ and $E(\lambda)$, along with the respective values of R$^2$. These variants represent parameterizations based on all available data (a general variant) as well as alternative parameterizations derived for data subsets relating to particular months (monthly variants). The parameterization coefficients for the general and selected monthly variants are listed in the Appendix (see Tables A1 and A2). Analysis of the curves in Figure 4 shows that the coefficients of the monthly parameterizations differ, exhibiting larger or smaller deviations from the course of the general variant's coefficients. In the case of coefficient $A$, the differences between 350-590 nm and around 675 nm are particularly conspicuous. For example, the highest values of coefficient $A$ for the 440 nm band were obtained in the case of parameterizations developed for September and December-January, and the lowest for April. With regard to the spectral slope of coefficient $A$ in the 350-440 nm range, the largest deviations from the typical course were recorded for December-January, April, and February. In contrast, the parameterizations obtained for



March, May and October are the closest to the general variant with respect to $A$. As regards coefficient $E$, the differences between the alternative parameterizations occur over the entire spectral range. In the case of the general variant, $E$ changes only slightly, between 0.81 and 0.91. In contrast, the values of $E$ for the parameterizations derived for individual months are spectrally more differentiated, with more pronounced local maxima and minima. The deviations from the general case of the parameterizations are the largest for March and April (upward) and for December-January (downward). As regards the determination coefficients $R^2$, which may initially characterize the accuracy of fit of the relationships with Eq. (8.a), it will be seen that the general variant takes relatively high values of $R^2$, not less than 0.8, over almost the entire visible range. Below 0.8, $R^2$ drops only at the edges of the relevant spectral ranges, i.e. 350-386 nm and 699-700 nm. In the case of the monthly parameterizations, only the formulas obtained for months with relatively large amounts of data take equally high values of $R^2$ (i.e. March, April, May and September). For the other months, $R^2$ drops below 0.8, at least in significant parts of the spectral range. Figure 5 illustrates the variability of the predicted spectral shapes of the light absorption coefficient when only the general parameterization variant is used to calculate it.

Figure 5a illustrates the family of curves representing the specific coefficients of light absorption by phytoplankton $a_{ph}^*(\lambda)$ calculated for a few chlorophyll $a$ concentrations from the 0.3-100 mg m$^{-3}$ range (corresponding more or less to the range that we recorded in the Baltic Sea). The bold line in Figure 5a outlines the spectrum calculated for $Tchla = 1$ mg m$^{-3}$ (corresponding to the numerical value of coefficient $A(\lambda)$). In addition, to better visualize the "evolution" of the spectral shape of the predicted spectra of $a_{ph}^*$, a family of curves corresponding to the normalized spectra of $a_{ph}(\lambda)$ was plotted in Figure 5b (spectra normalized with respect to 440 nm). Moreover, Figures 5c and d show analogous diagrams obtained using the "classic" parameterization developed by Bricaud et al. (1995), although it should be mentioned that the two highest $Tchla$ values – 30 and 100 mg m$^{-3}$ – generally lie beyond the range for which Bricaud et al. (1995) originally developed their parameterization. These graphs show that the predicted spectral shapes of coefficients $a_{ph}^*$ are clearly less variable when the general variant of our new "Baltic" formula is used rather than the "classic" parameterization according to Bricaud et al. (1995). In our parameterization, the spectral shapes are distinctly different, mainly in the 600-680 nm spectral range, whereas in the parameterization according to Bricaud et al. (1995) the variations occur over a much broader spectral range. One can assume, for example, the value of $a_{ph}(440)/a_{ph}(675)$, also referred to as the colour index, to be a simplified quantitative measure of the "flattening" of light absorption spectra along with increasing chlorophyll concentration (see e.g. Woźniak and Ostrowska 1990 a and b). When the parameterization by Bricaud et al. (1995) is applied to $Tchla$ concentrations from 0.3 to 100 mg m$^{-3}$, the colour index changes roughly three-fold, i.e. it decreases from 2.69 to 0.88, the latter value signifying a greater absorption of light in the red band than in the blue. By contrast, with our new parameterization, the colour index drops by a factor of only around 1.57 (from 2.76 to 1.76). The comparison presented in Figure 5, however, refers only to the results obtained using the general version of our new parameterization, that is, the one developed on the basis of all the data we collected in different periods of the year. On the other hand, there are clear differences between the variants of the parameterizations matched for individual months.


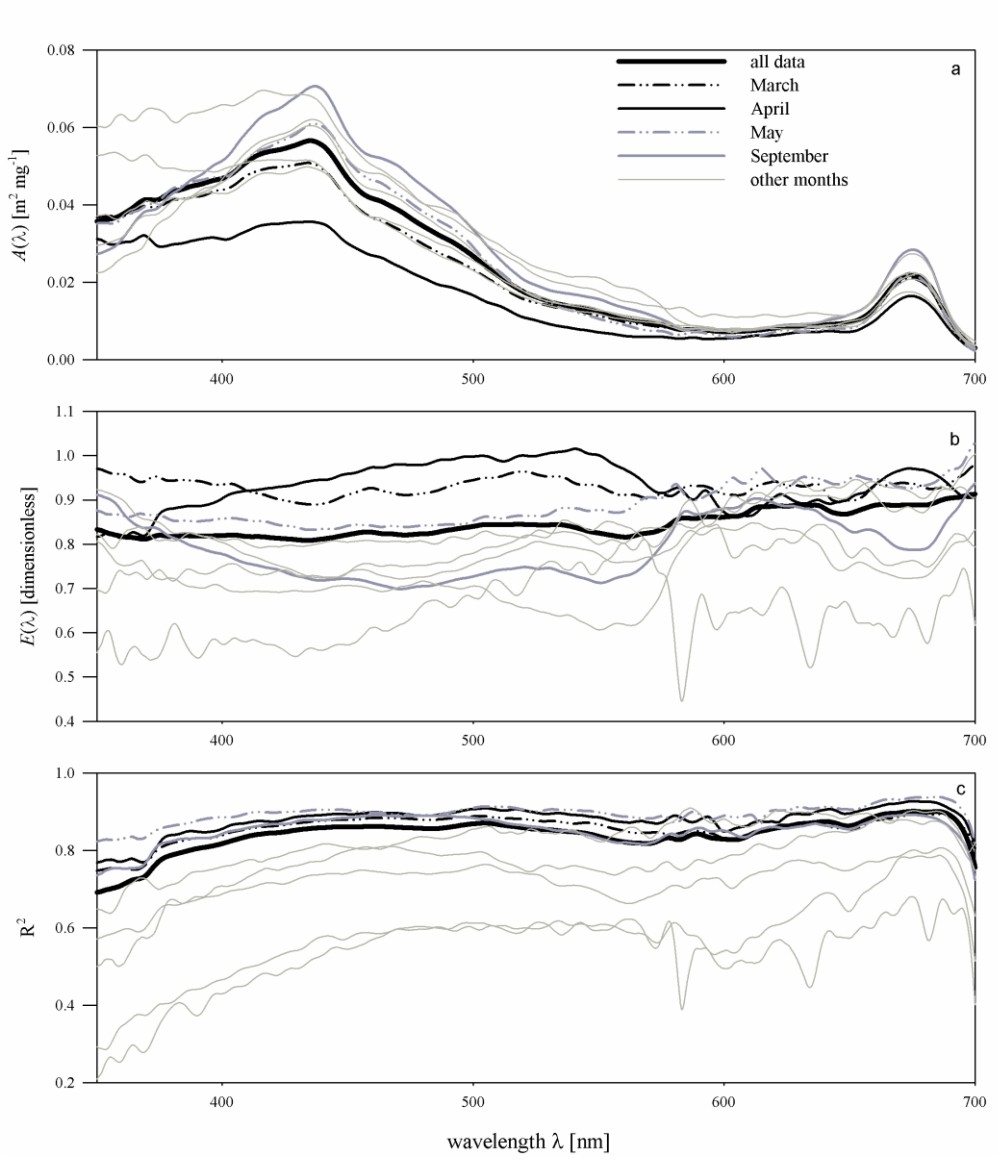

**Figure 4.** Spectral plots of coefficients *A* (panel a) and *E* (panel b) of the parameterizations described by Eq. (8.a) and the corresponding values of the determination coefficient $R^2$ (panel c), determined for all the data analysed and for each sampling month.

5    Figure 6 shows how the different slopes of curves approximately describing the dependence of $a_{ph}^*$ on *Tchla* are obtained for two spectral bands (440 and 675 nm) and the months of March, April, May and September (only those months were selected where $R^2$ was relatively high for the relevant parameterization). In the case of September, for example, we see the evidently steeper slopes of the approximated $a_{ph}^*$ vs *Tchla* curves than for the general parameterization, whereas for April the slopes of the curves are clearly less steep.







**Figure 5.** (a) and (b): Example spectra of the specific coefficients of light absorption by phytoplankton $a^*_{ph}(\lambda)$, or spectra of $a_{ph}$ normalized to its own value in the 440 nm band, for selected *Tchla* concentrations between 0.3 and 100 mg m$^{-3}$, estimated using the general variant of parameterization obtained in this paper; (c) and (d): as panels (a) and (b), but plotted on the basis of the parameterization by Bricaud et al. (1995). The grey lines on panels (c) and (d) represent values of *Tchla* that go beyond the range for which the Bricaud et al. parameterization was originally developed (i.e. for *Tchla* values >25 mg m$^{-3}$).





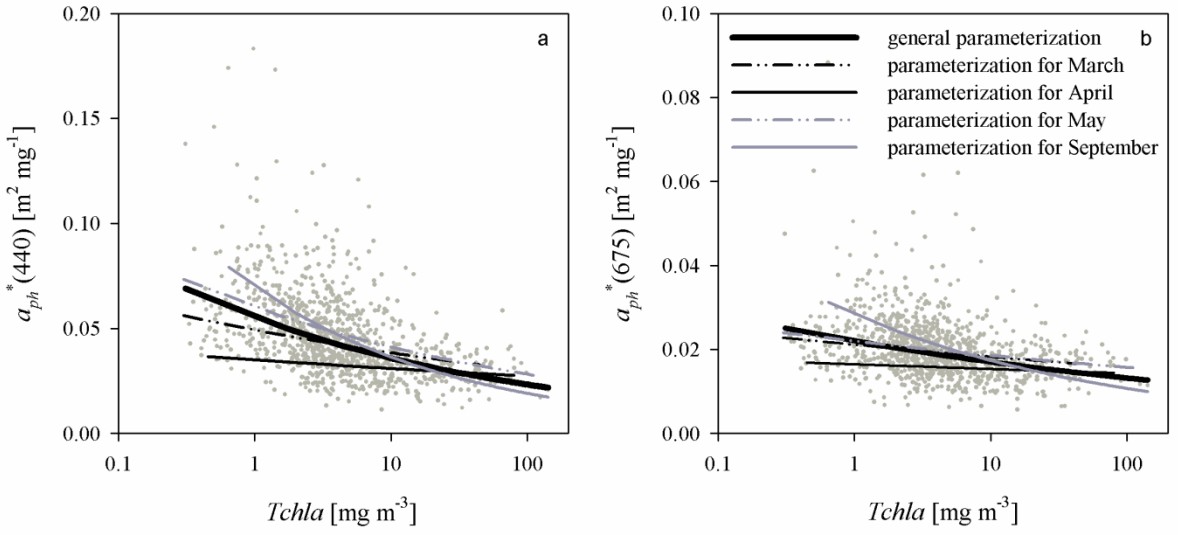

**Figure 6.** (a) Relation between the coefficient $a^*_{ph}(440)$ and the chlorophyll $a$ concentration $Tchla$ and its functional approximations determined for all the data analysed and for four sampling months; (b): as (a) but for $a^*_{ph}(675)$. The grey dots on each panel represent individual data points, and the lines represent the functional approximation with Eq. 8.b (see the caption to panel (b)).

The obvious differences between these months also result in different "evolutions" of the $a_{ph}^*$ spectral curves predicted by the different parameterization variants (see Figure 7). The family of $a_{ph}^*(\lambda)$ curves generally representing the lowest values are those plotted on the basis of the April parameterization (Figure 7b), whereas in general the highest values are represented by the September family of curves (Figure 7d). The variability in the normalized shapes of the $a_{ph}$ coefficient is greater and more complex (Figures 7e to h) than in the case of the general parameterization (see Figure 5b). The variability of the colour index is the highest by a factor of 1.71 (a drop from 3.06 to 1.79) for the curves plotted according to the May parameterization, and the smallest by a factor of just 1.16 (a drop from 2.2 to 1.9) for April. There are, moreover, differences in the "evolution" of slopes in the short-wave part of the spectrum between these four months that we failed to perceive using only the general version of the parameterization.







**Figure 7.** Example spectra of the specific coefficient of light absorption by phytoplankton $a^*_{ph}$ (panels a – d), and spectra of $a_{ph}$ normalized to its value in the 440 nm band (panels e – h), for selected chlorophyll $a$ concentrations *Tchla* between 0.3 and 100 mg m$^{-3}$, estimated using the parameterization variants obtained in this study for the following months: March, April, May and September (the month descriptions are given in each panel).

In addition to the analysis of the determination coefficients $R^2$ presented earlier, we performed a more extensive analysis of the errors made using the proposed approximation formulas for the whole of our dataset (analysis of estimation errors). The results of such analyses, using both arithmetic and logarithmic statistics, for nine wavelengths of light, are set out in Table 3. These wavelengths were chosen to cover the spectral range under consideration and include the characteristic maxima of light absorption by chlorophyll $a$. The results in Table 3 relate to two scenarios: one, when only the general variant of the parameterization was used to calculate absorption coefficients for all the data, and the other, when the relevant variants of monthly parameterizations were used, depending on the month of data acquisition (the values for the latter scenario are given in parentheses). The accuracy of estimation of coefficients $a_{ph}$ obtained using the general parameterization, expressed by both arithmetic and logarithmic statistics, seems satisfactory, at least for wavelengths from 400 to 690 nm. In this range, the systematic error according to arithmetic statistics remains at the relatively low level of 5-9%, and the statistical error varies from 34% to just over 50%. Because the general parameterization was developed using linear least-squares regression applied to the logarithms of *Tchla* and $a_{ph}$ values, the systematic error according to logarithmic statistics is always equal or very close to zero. The standard error factor, which enables the statistical error range according to logarithmic statistics to be determined, varies between 1.37 and 1.52 for wavelengths from 400 to 690 nm, and reaches higher values only at the edges of the spectral range under investigation. This means that the statistical error according to logarithmic statistics in the 400-690 nm range varies from -34% to 52%; if the entire spectral range is considered, it varies from -45% to 81%. In turn, the estimation errors obtained for the scenario where the monthly parameterization variants are applied, are, as expected, even slightly lower than the previous ones. The systematic error according to arithmetic statistics in the 400-690 nm range remains at 4-8%, and the statistical error varies between 31 and 48%. Applying logarithmic statistics to this scenario leads to a standard error factor varying from 1.34 to 1.49 in the 400-690 nm range, and taking values of less than or equal to 1.75 at the edges of this range. Hence, the statistical error according to logarithmic statistics in the 400-690 nm range varies from -33% to 49%, and from -45% to 75% if the entire spectral range is considered.



**Table 3.** Estimation errors of $a_{ph}(\lambda)$ coefficient in selected spectral bands, obtained when the one-variable parameterization obtained in this study (Eq. 8.a) was applied to the entire available dataset (n= 1002) in its general variant or, alternatively, in variants specified for individual months (alternative values are given in parentheses).

| Light wavelength [nm] | Arithmetic statistics | | Logarithmic statistics | | | |
| --- | --- | --- | --- | --- | --- | --- |
| | systematic error | statistical error | systematic error | standard error factor | statistical error | |
| | $\langle\varepsilon\rangle$[%] | $\sigma_\varepsilon$ [%] | $\langle\varepsilon\rangle_g$[%] | $x$ | $\sigma_+$ [%] | $\sigma_-$[%] |
| 350 | 17.86 (15.98) | 67.72 (65.17) | 0 (-0.03) | 1.81 (1.75) | 80.85 (74.66) | -44.71 (-42.75) |
| 400 | 8.54 (7.67) | 44.70 (41.65) | 0 (-0.01) | 1.51 (1.48) | 50.95 (48.15) | -33.75 (-32.50) |
| 440 | 6.29 (5.24) | 38.53 (34.10) | 0 (-0.01) | 1.42 (1.38) | 41.98 (38.41) | -29.59 (-27.75) |
| 500 | 6.09 (5.15) | 37.38 (33.65) | 0 (-0.04) | 1.42 (1.38) | 41.55 (38.36) | -29.35 (-27.72) |
| 550 | 7.62 (6.67) | 42.38 (39.19) | 0 (-0.05) | 1.47 (1.44) | 47.47 (44.33) | -32.19 (-30.72) |
| 600 | 8.83 (8.05) | 47.43 (46.31) | 0 (-0.15) | 1.52 (1.49) | 51.60 (49.15) | -34.04 (-32.95) |
| 675 | 4.99 (4.19) | 34.21 (30.78) | 0 (-0.04) | 1.37 (1.34) | 36.85 (33.94) | -26.93 (-25.34) |
| 690 | 5.79 (5.35) | 37.30 (36.50) | 0 (-0.11) | 1.40 (1.39) | 40.08 (38.71) | -28.61 (-27.91) |
| 700 | 23.81 (21.78) | 199.07 (193.1) | -0.93 (-0.16) | 1.69 (1.70) | 69.01 (69.56) | -40.83 (-41.02) |

The estimation errors listed in Table 3 relate to these two scenarios, but both treat the entire available dataset as a single whole. Therefore, they do not address the question of how much more accurate the results might be if a correctly chosen monthly parameterization were used instead of the general one, when we limit ourselves to data from one particular month. We have carried out such an analysis, but we do not present its detailed results due to the fact that they are very extensive. Here we limit ourselves only to stating that the use of general or monthly parameterizations for individual months has little effect on the level of statistical error according to logarithmic statistics, although it may have, as generally expected, a very significant impact on the level of systematic error. Applying the general variant of the parameterization to particular months may overestimate or underestimate the values of coefficients $a_{ph}$ by a factor of a few percent to as much as 30% and more in extreme situations. If we take the case of April, the general parameterization variant overestimates $a_{ph}(\lambda)$ by ca 12% to 34%, depending on the light wavelength. For September, on the other hand, $a_{ph}(\lambda)$ is underestimated by ca 2 to 9% in the majority of the visible range. In other months $a_{ph}(\lambda)$ may be overestimated in some spectral ranges and underestimated in others: in May, for example, this coefficient was underestimated by up to -11% in the short-wave range but overestimated by up to +16% in the long-wave range. In contrast, application of appropriately matched monthly parameterizations in all of these cases reduces the statistical error according to logarithmic statistics to practically zero.



### 3.2.2 Example of a two-component parameterization

As indicated in section 3.1.2, there is noticeable variation in the proportion between *Tchla* and the concentrations of other phytoplankton pigments in particular months of the year within the dataset (see also Figure 3). This variability indicates, of course, the limitations that crop up when using the chlorophyll *a* concentration as the only variable for parameterizing the spectra of $a_{ph}(\lambda)$. As a step towards trying to improve the approximation accuracy of $a_{ph}$ while retaining the relative simplicity of the mathematical formalism used (i.e. without resorting to complex multicomponent models, where it would be necessary to know the concentrations of all the different phytoplankton pigments), we decided to search for just one additional variable. For this purpose, the relationship described earlier by Eq. (8.a) was treated as a first, intermediate stage in the construction of a two-component parameterization. To distinguish between them, the values calculated according to Eq. (8.a) are now denoted $a_{ph}(\lambda)_{cal}$, whereas the actually measured values of the absorption coefficient are $a_{ph}(\lambda)_m$. In the next step, the relationship between the ratio $a_{ph}(\lambda)_{cal}/a_{ph}(\lambda)_m$ and the sum of the concentrations of all the other accessory pigments $\Sigma C_i$ was investigated. The analysis confirmed that the ratio of the sum of accessory pigments to the concentration of chlorophyll *a*, $\Sigma C_i/Tchla$, can be used as a second independent variable for the parameterization of $a_{ph}(\lambda)$. Figure 8 illustrates in a simplified manner, and for two light wavelengths, the consecutive steps in the construction of the new parameterization. Approximate relationships between $a_{ph}(\lambda)_{cal}$ and *Tchla* are plotted in Figures 8a and b – this illustrates the first step – while Figures 8c and d show the frequency distributions of the ratio $a_{ph}(\lambda)_{cal}/a_{ph}(\lambda)_m$. The second and final step in the construction of the new parameterization is shown in Figures 8e and f, which depict the relationships between the ratios $a_{ph}(\lambda)_m/a_{ph}(\lambda)_{cal}$ and $\Sigma C_i/Tchla$. Despite the large dispersion of individual data points on the latter two panels, the general tendency for $a_{ph}(\lambda)_m/a_{ph}(\lambda)_{cal}$ to decrease with increasing $\Sigma C_i/Tchla$ is clear. This trend can be described by an approximate exponential relationship that takes the following form:

$$\frac{a_{ph}(\lambda)_{cal}}{a_{ph}(\lambda)_m} = f\left(\frac{\Sigma C_i}{Tchla}\right) = const_1(\lambda)e^{const_2(\lambda)\frac{\Sigma C_i}{Tchla}}. \tag{9}$$

Relationships of this form were determined over the entire spectral range (350-700 nm) with a step of 1 nm. Obviously, their determination coefficients $R^2$ are low, but they do permit additional information on the influence of pigment composition on the ultimate values of $a_{ph}$ to be taken into account.



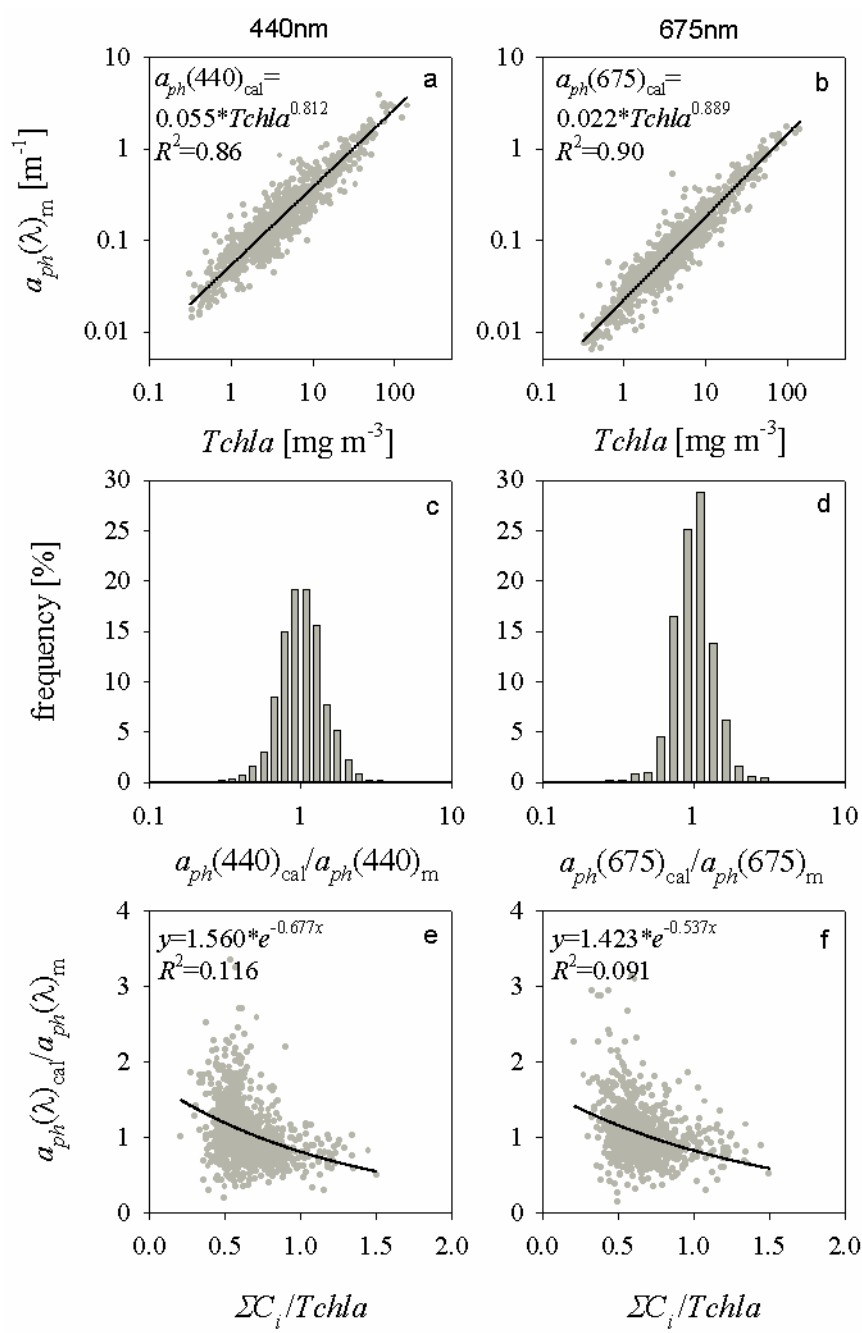

**Figure 8**. (a) and (b): Relations between measured coefficients of absorption by phytoplankton (denoted by $a_{ph}(\lambda)_m$) and chlorophyll $a$ concentrations for two light wavelengths, 400 and 675 nm, and their functional approximations according to Eq. (8.a) (the equations are given in the panels); (c) and (d): frequency distribution histograms of the ratio of calculated and




measured absorption coefficients ($a_{ph}(\lambda)_{cal}/a_{ph}(\lambda)_m$); (e) and (f): relations between the ratio $a_{ph}(\lambda)_{cal}/a_{ph}(\lambda)_m$ and the pigment concentration ratio $\Sigma C_i/Tchla$, and their functional approximations (the equations are given in the panels).

Having the statistical dependences described by formulas (8.a) and (9) to hand, a new expression can be written that approximates $a_{ph}(\lambda)$ by treating it as a function of two variables at each light wavelength – chlorophyll $a$ concentration (*Tchla* [mg m$^{-3}$]) and the ratio of the sum of the concentrations of the other accessory pigments to chlorophyll $a$ ($\Sigma C_i/Tchla$):

$$a_{ph}(\lambda) = A_0(\lambda) \cdot e^{K(\lambda) \cdot \frac{\Sigma C_i}{Tchla}} \cdot Tchla^{E(\lambda)}. \tag{10}$$

The numerical coefficients of the newly obtained parameterization, i.e. $A_0(\lambda)$ [m$^2$ mg$^{-1}$] (where $A_0(\lambda)=A(\lambda)/const_1(\lambda)$), and $K(\lambda)$ [no units] (where $K(\lambda)=-const_2(\lambda)$) are summarized in Table A3 in the Appendix (with a spectral resolution of 2 nm). Note that coefficient $E(\lambda)$ [no units] takes the same values as those in the general variant of the one-component parameterization. Note, too, that the product of the new coefficient $A_0(\lambda)$ and the exponential function appearing in equation (10) allows one, with the adopted value of the ratio $\Sigma C_i/Tchla$, to calculate the value corresponding to coefficient $A(\lambda)$ from the parameterization given by formula (8.a). We define this product as:

$$A'\left(\lambda, \frac{\Sigma C_i}{Tchla}\right) = A_0(\lambda) \cdot e^{K(\lambda) \cdot \frac{\Sigma C_i}{Tchla}}. \tag{11}$$

Spectral values of the new coefficients of equation (10) (coefficients $A_0(\lambda)$ and $K(\lambda)$) are shown in Figure 9.a, and Figure 9.b illustrates the family of $A'(\lambda, \Sigma C_i/Tchla)$ curves plotted for some values of $\Sigma C_i/Tchla$ in our database. These include the median and average values, and the 10th, 25th, 75th and 90th percentiles calculated for the whole dataset; the 10th percentile for the period December-January (0.39) and the 90th percentile for the month of May (1.15) were also taken into consideration as examples of "extreme" values.



**Figure 9.** (a): Spectral plots of coefficients $A_0$ and $K$ of the two-component parameterization described by Eq. (10); (b): examples of curves representing coefficient $A'(\lambda, \Sigma C_i/Tchla)$ defined by Eq. 11, for the following values of $\Sigma C_i/Tchla$: the average value of 0.62; selected values between 0.39 and 1.15 (representing the range from the 10th percentile for the December-January period to the 90th percentile for May); the hypothetical value of 0

As in the case of single-variable parameterizations, example families of $a_{ph}^*$ curves are now presented for the new two-component parameterization (Figure 10) for two values of $\Sigma C_i/Tchla$, i.e. 0.47 and 0.88, corresponding to the 10th and 90th percentiles from the observed distribution of that ratio. There are conspicuous differences in this respect between both the values and shapes of the $a_{ph}^*$ spectra. As expected, the new two-component parameterization generally predicts lower values of $a_{ph}^*(\lambda)$ for low values of $\Sigma C_i/Tchla$ than for higher ones. If we assume a low proportion of accessory pigments, i.e. for $\Sigma C_i/Tchla = 0.47$ and $Tchla$ increasing from 0.3 to 100 mg m$^{-3}$, the colour index falls from 2.69 to 1.72, i.e. by a factor of 1.57. In contrast, if we assume a higher proportion of accessory pigments (= 0.88), the colour index decreases by the same factor (1.57), but from a higher starting value of 2.85 to 1.82. However, none of these differences are as distinct as those





between the families of $a_{ph}^*$ curves, drawn earlier according to the one-component parameterizations obtained for selected months (see Figure 7).Generally speaking, we can expect the use of the two-component parameterization to introduce an additional "degree of freedom" to the description of the variability of parameterized light absorption spectra. But it also seems likely that even with the new two-component parameterization it will not be possible to explain all the differences manifested by the monthly one-component parameterizations. This intuitive expectation can be quantitatively checked by analysing in detail the estimation errors calculated for different variants of the parameterizations.

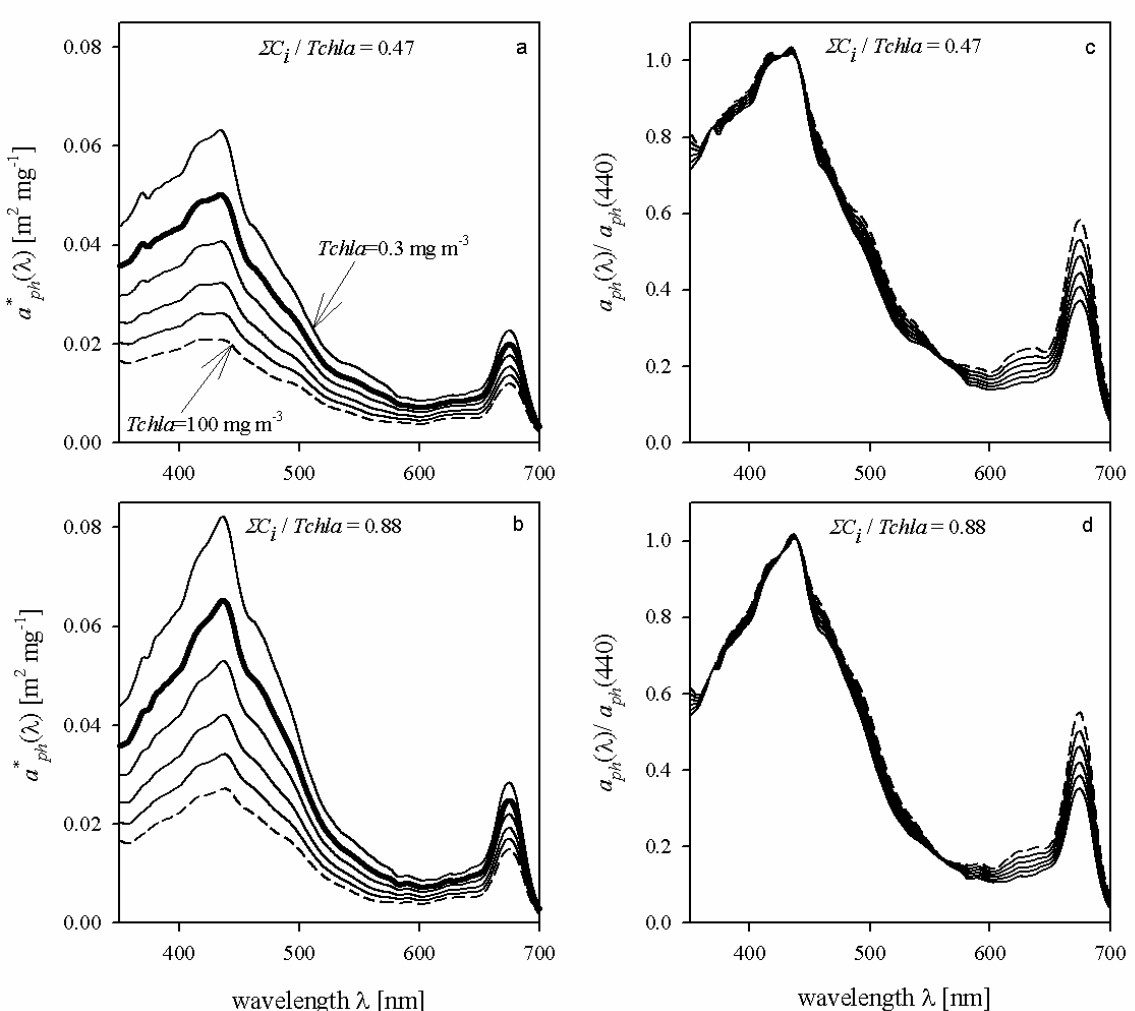

**Figure10**. Example spectra of the specific coefficient of light absorption by phytoplankton $a_{ph}^*$ (panels a and b) and spectra of coefficient $a_{ph}$ normalized to its value in the 440 nm band (panels c and d), for selected concentrations *Tchla* between 0.3 and 100 mg m$^{-3}$, estimated by the two-component parameterization (Eq. 10) and for two different assumed constant values of the pigment concentration ratio $\Sigma C_i/Tchla$: 0.47 (representing the 10th percentile for all data), and 0.88 (representing the 90th percentile for all data).



The estimation errors relating to the use of the two-component parameterization were assessed in the context of the entire available dataset: the results for selected light wavelengths are listed in Table 4. As in the case of the one-component parameterization, the errors of the two-component parameterization generally persist at a relatively low level across almost the whole visible light spectrum (from 400 to 690 nm). In this range, the arithmetic systematic error remains at 4.5-9%, while the arithmetic statistical error varies from 32% to 51%. Larger errors are observed only at the edges of the spectral range under investigation (from 350 to 400 nm and above 690 nm). According to logarithmic statistics, the standard error factor in the 400-690 nm range varies from 1.35 to 1.52, while at the edges of this range, it can reach values up to 1.82. This means that the logarithmic statistical error in the 400-690 nm range varies from -35% to 53%, whereas over the entire spectral range in question, this error varies from -45% to 82%. At this juncture, we can confirm that the use of the two-component parameterization leads to a noticeable reduction in the errors compared with the general version of the one-component parameterization (Eq. 8.a) only in the 390-530 nm and 664-684 nm spectral ranges. These are the ranges in which significant differences in the family of $A'(\lambda, \Sigma C_i/Tchla)$ curves have been observed (see Figure 9b). However, comparison of the estimation errors associated with the two-component parameterization with the use of monthly variants of the one-component parameterization slightly favours the latter (compare the values given in Tables 3 and 4). Further details of the comparisons of different variants of the parameterization will be given in the next section.

**Table 4.** Estimation errors of $a_{ph}(\lambda)$ coefficient in selected spectral bands, obtained when the two-variable parameterization obtained in this study (Eq. 10) was applied to the entire available dataset (n= 1002).

| Light wavelength [nm] | Arithmetic statistics | | Logarithmic statistics | | | |
| --- | --- | --- | --- | --- | --- | --- |
| | systematic error | statistical error | systematic error | standard error factor | statistical error | |
| | $\langle\varepsilon\rangle$ [%] | $\sigma_\varepsilon$ [%] | $\langle\varepsilon\rangle_g$ [%] | $x$ | $\sigma_+$ [%] | $\sigma_-$ [%] |
| 350 | 17.97 | 68.20 | -0.11 | 1.82 | 81.61 | -44.94 |
| 400 | 8.13 | 43.10 | -0.14 | 1.51 | 50.69 | -33.64 |
| 440 | 5.39 | 35.13 | -0.13 | 1.40 | 39.58 | -28.36 |
| 500 | 5.22 | 34.14 | -0.17 | 1.39 | 39.48 | -28.33 |
| 550 | 7.63 | 42.67 | -0.13 | 1.48 | 48.13 | -32.49 |
| 600 | 8.84 | 47.70 | -0.10 | 1.52 | 52.18 | -34.29 |
| 675 | 4.43 | 32.17 | -0.15 | 1.35 | 35.49 | -26.19 |
| 690 | 5.80 | 38.06 | -0.15 | 1.41 | 40.68 | -28.92 |
| 700 | 24.46 | 194.02 | 0.79 | 1.74 | 73.66 | -42.42 |





### 3.3 Comparison with various parameterizations from the literature

So far we have referred only to the "classic" version of the parameterization given by Bricaud et al. (1995) for oceanic, "case 1" waters. Now we shall compare our results with those from other papers (mentioned in the Introduction). The first two panels of Figure 11 compare selected variants of parameterization coefficients obtained in this work with literature examples for different marine environments. In the case of coefficients $A$, the shapes of all the spectra corresponding to these different examples generally reflect the characteristic absorption maxima in the blue and red spectral ranges (see Figure 11a). Quantitatively, however, there are significant differences between the values of coefficients $A$ corresponding to these examples, the largest being in the wavelength range from about 400 to 480 nm. In this range, the lowest values of $A$ are given by the "classic" parameterization according to Bricaud et al. (1995), and the highest by the summer version of the parameterization by Churilova et al. (2017). Interestingly, however, the range of variability of $A$ between the parameterizations quoted from the literature is generally similar to the one we obtained with our own data by developing separate variants of the one-component parameterization for individual months. With regard to the spectral values of coefficients $E$, the parameterizations derived by other authors exhibit a much greater spectral differentiation of these coefficients, in contrast to the general variant we obtained in this paper (and one example from our earlier work, Meler et al. (2017a)), where these coefficients change only slightly between ca 0.8 and 0.9. According to different literature sources, coefficients $E$ take values from under 0.6 to even more than 1 in different spectral ranges. When analysing our own data, we also noted the spectral diversity of $E$ in parameterizations obtained for individual months (values from ca 0.75 to ca 1), but our values of this coefficient were not as low as in the parameterization according to Bricaud et al. (1995) in the 400-500 nm range, or in the parameterization according to Dmitriev et al. (2009) in the ca 400-540 nm and 570-615 nm ranges. The fact that the various literature parameterizations clearly differ in their coefficients $E$ can be additionally illustrated by graphs of estimated dependences of specific absorption coefficients $a_{ph}*(440)$ and $a_{ph}*(675)$ as functions of $Tchla$ (see Figure 11 c and d).





**Figure11.** (a) and (b): Comparison of spectra of coefficients *A*, *A'* and *E* obtained in this study with the corresponding coefficients of parameterizations derived for different marine environments by other authors (for the description of the lines, see the caption in panel a); (c) and (d): comparison of functional approximations of the relations between coefficients $a^*_{ph}(440)$ or $a^*_{ph}(675)$ and concentrations *Tchla* (the grey dots represent individual points from our dataset; the different lines represent approximations according to different versions of the parameterization; see the caption given in panel d).



Figure 12 illustrates the errors calculated according to the logarithmic statistics of estimating coefficient $a_{ph}$ when different variants of both our parameterizations and those from the literature are applied to our whole dataset. In addition to the already presented examples of "marine" parameterizations, we have taken into account examples from a variety of lacustrine environments. Of course, neither of the new variants of the parameterization proposed in this work reveal any

systematic error at all as they were derived from the same set of data, unlike the literature parameterizations that we are comparing here (see Figures 12a and c). For example, when the parameterization according to Bricaud et al. (1995) is applied, the systematic estimation errors of $a_{ph}(\lambda)$ range from -57 to -21% over almost the entire spectral range. Calculated values of $a_{ph}$ are also significantly lower in the entire spectral range when we use the parameterization according to Dmitriev et. (2009), and in selected spectral ranges for the parameterizations according to Ylöstalo et al. (2014) and Paavel et al.

(2016). For example, there is a significant increase in calculated $a_{ph}$ values in the short-wavelength range using the summer version of the parameterization by Churilova et al. (2017) and in the long-wavelength range for the parameterization by Paavel et al. (2016). In general, we obtained the smallest systematic errors (<15%) when using the parameterization by Staehr and Markager (2004) in the spectral range from ca 450 to 685 nm (for the whole spectral range studied, the systematic error of this particular parameterization varies from -35 to +51 %). With regard to the version of the parameterization from

our earlier work (Meler et al. 2017a), we now see that, apart from the UV range, values of $a_{ph}$ were generally overestimated by up to 20% and more (in the 555-600 nm range and above 680 nm). In the case of the standard error factor, which characterizes the range of the estimate's statistical error, only two variants (summer and winter) of the parameterizations by Churilova et al. (2017) among all the examples of "marine" parameterizations take values similar to those characterizing the general variant of parameterization given in this work in the most important spectral range from 400 to 500 nm. Among the

different parameterizations obtained for lacustrine environments, we can get similar values of the standard error factor only with the parameterizations by Ylöstalo et al. (2014) and Ficek et al. (2012a). However, none of the literature parameterizations can reach the level of the standard error factor obtain able with our new two-component parameterization or with one-component parameterizations in properly selected monthly variants. If we take into account both systematic and statistical errors, the superiority of the new parameterizations developed specifically for Baltic Sea conditions is undisputed.




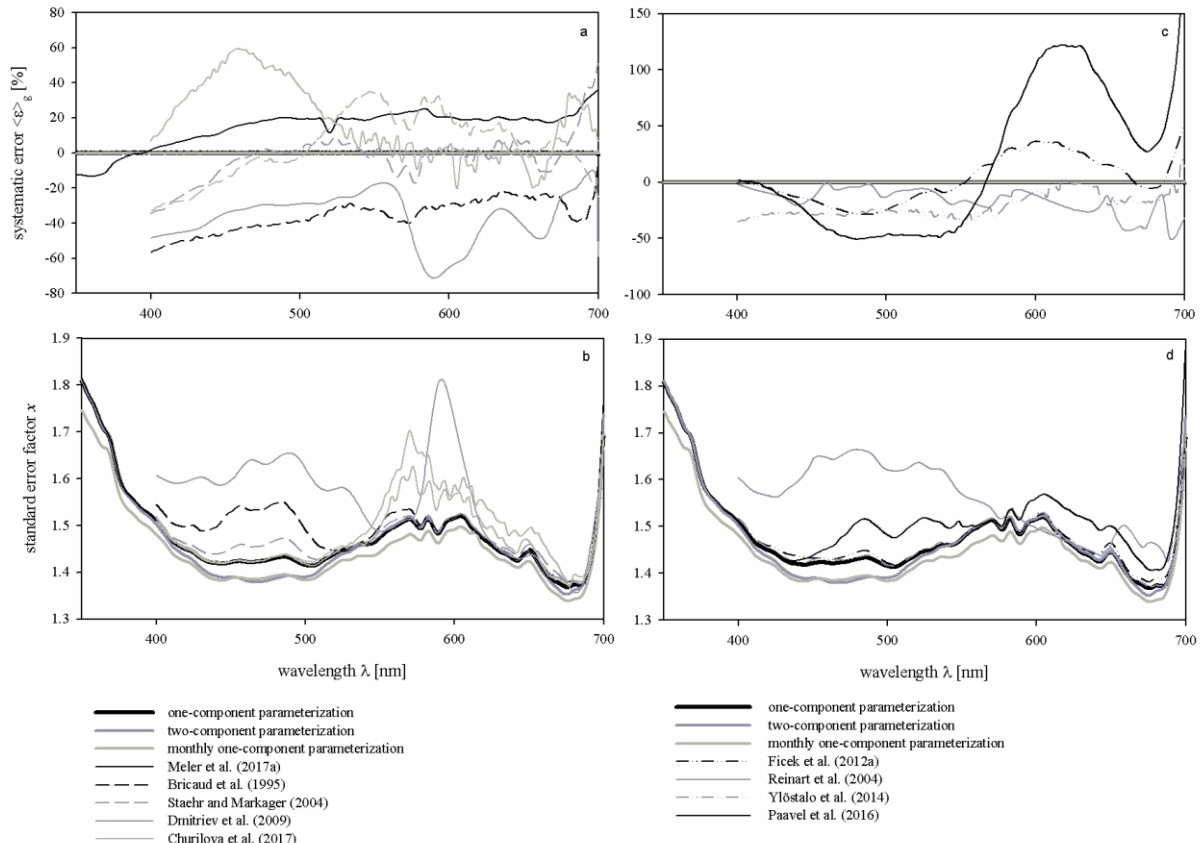

**Figure 12.** Spectral values of the systematic error of estimation $\langle\varepsilon\rangle_g$ (panels a and c) and the standard error factor $x$ (panels b and d) calculated when different parameterizations – both those obtained in this study and others gleaned from the literature – were applied to our dataset in order to calculate coefficient $a_{ph}(\lambda)$. Panels (a) and (b) compare the estimation errors with examples of parameterizations derived for different marine environments; panel (c) and (d) compares examples for lacustrine environments. For the line descriptions, see the caption given below the panels.

## 4 Final remarks

The empirical material for this work was acquired in a relatively small geographical area, mainly the southern Baltic Sea. However, because it was gathered in various parts of this basin, from coastal areas to open waters, and at different times of the year, the recorded light absorption coefficients and concentrations of phytoplankton pigments have large ranges of variability, in both cases reaching almost three orders of magnitude. Based on such a dataset it was possible to develop a number of new variants of the parameterization of coefficient $a_{ph}$: they should be treated as simplified and practical relationships of a local character, tailored to the specifics of the target environment. The new empirical formulas include, among other things, classic one-component parameterizations, where the only variable is the concentration of chlorophyll $a$.





Parameterizations of this type have been developed both as a general version, i.e. one matched to all data collected in different periods of the year, and in the form of separate variants adjusted to the individual months of data collection. Importantly, it was found that the matched coefficients of monthly variants could differ from each other very significantly, thus indirectly reflecting the annual variation in the proportions between chlorophyll *a* and other photosynthetic or

photoprotective pigments. The paper also presents a new, slightly more complex form of parameterization that uses one additional variable: the ratio of the concentrations of accessory pigments to the concentration of chlorophyll *a*. With all the variants of this parameterization, spectra of coefficient $a_{ph}$ can be estimated fairly simply, and with few requirements as to input data. Such estimates can be made over a wide spectral range (from 350 to 700 nm) and with a high spectral resolution (1 nm). It should be borne in mind, however, that the accuracy of such estimates is obviously limited. For example,

application of the general version of the one-component parameterization to all our data covering different periods of the year understandably leads to a practically zero systematic error of this estimate, although a significant statistical error remains. The latter may be characterized by standard error factors from 1.37 to 1.51 in the vast majority of the spectral ranges tested. However, since the real values of $a_{ph}$ vary in Baltic Sea conditions over almost three orders of magnitude, even an estimation accuracy such as this appears satisfactory. Our study has also shown that further improvement in the

accuracy of the approximate description of $a_{ph}$ spectra is possible, at least in some applications. This can be achieved, *inter alia*, by using parameterization variants specially matched to individual months of the year or by using the new two-component parameterization. One can, for example, achieve low levels of systematic error even in relation to data subsets limited to individual months, and obtain an additional noticeable reduction in the statistical error. Moreover, this paper records that in case of Baltic Sea data, regardless of the parameterization chosen, each of its newly developed variants

permits a more accurate estimate of $a_{ph}$ spectra than other, similar formulas known to date from the literature. The other important qualitative observation is that new variants of monthly parameterizations have a range of variability of coefficients similar to that between different literature parameterizations established on the basis of data from various aquatic environments. This particular observation reminds us that all such parameterizations are always quite far-reaching simplifications of relationships occurring in nature. The variability of these relationships that we recorded throughout the

25    year in the Baltic Sea seems to indicate that only the use of a much more elaborate mathematical apparatus, using a much larger number of variables describing the composition of pigments and other features of the phytoplankton population, could further and more radically improve the accuracy of the spectral description of the light absorption coefficient (see, e.g., the multi-component models presented earlier in the papers by B. Woźniak and his collaborators (Woźniak et al. 1999, Woźniak et al. 2000, Majchrowski et al. 2000, Ficek et al. 2004)). In our opinion, however, the practical value of the simple

parameterizations presented in this work should be seen in the opportunities for applying them to the development of methods, whose specificity from the very beginning requires the use of simplifications (e.g. which do not permit the use of an extensive set of descriptors of phytoplankton population characteristics). Practical remote sensing algorithms are a good example in this respect. Their task is often to solve complicated "reverse" problems. Starting with measurements of sea colour, such algorithms, through intermediate steps during which different inherent optical properties of seawater (including



light absorption) are retrieved, should yield basic features of different seawater components in the final stage of their operation. One such feature may be the concentration of chlorophyll *a*, which is often treated as a practical measure of the biomass of live phytoplankton contained in water. In our opinion the parameterizations presented in this work could be used in practice for such purposes.

5 **5 Data availability**

All data used in this study will be freely available, for scientific use only, upon request. Anyone interested in using this data set for scientific research should contact the corresponding author via e-mail.

**Appendix**

The spectral coefficients of certain variants of the parameterizations obtained in this work are given in the tables below with
10 either 2 or 5 nm spectral steps. The values of the coefficients with 1 nm resolution and for other cases not presented below are available from the authors on request.

**Table A1.** Spectral coefficients *A* and *E* of the one-component parameterization described by Eq. (8.a) in its general variant (i.e. the variant obtained when all data, regardless of the month of acquisition, were taken into account) and the corresponding determination coefficients $R^2$.

| λ [nm] | A | E | $R^2$ | λ [nm] | A | E | $R^2$ | λ [nm] | A | E | $R^2$ |
|---|---|---|---|---|---|---|---|---|---|---|---|
| *1* | *2* | *3* | *4* | *1* | *2* | *3* | *4* | *1* | *2* | *3* | *4* |
| **350** | 0.0358 | 0.833 | *0.69* | **468** | 0.0394 | 0.823 | *0.86* | **586** | 0.0078 | 0.859 | *0.84* |
| **352** | 0.0360 | 0.831 | *0.69* | **470** | 0.0387 | 0.822 | *0.86* | **588** | 0.0078 | 0.858 | *0.84* |
| **354** | 0.0362 | 0.826 | *0.70* | **472** | 0.0379 | 0.821 | *0.86* | **590** | 0.0078 | 0.859 | *0.84* |
| **356** | 0.0364 | 0.823 | *0.70* | **474** | 0.0371 | 0.821 | *0.86* | **592** | 0.0077 | 0.861 | *0.84* |
| **358** | 0.0368 | 0.820 | *0.70* | **476** | 0.0361 | 0.822 | *0.86* | **594** | 0.0076 | 0.863 | *0.83* |
| **360** | 0.0374 | 0.818 | *0.71* | **478** | 0.0353 | 0.823 | *0.86* | **596** | 0.0074 | 0.862 | *0.83* |
| **362** | 0.0382 | 0.816 | *0.72* | **480** | 0.0345 | 0.823 | *0.86* | **598** | 0.0073 | 0.861 | *0.83* |
| **364** | 0.0391 | 0.815 | *0.72* | **482** | 0.0338 | 0.824 | *0.86* | **600** | 0.0072 | 0.861 | *0.83* |
| **366** | 0.0401 | 0.814 | *0.73* | **484** | 0.0330 | 0.825 | *0.86* | **602** | 0.0072 | 0.862 | *0.83* |
| **368** | 0.0410 | 0.812 | *0.73* | **486** | 0.0323 | 0.827 | *0.86* | **604** | 0.0073 | 0.863 | *0.83* |
| **370** | 0.0414 | 0.812 | *0.74* | **488** | 0.0316 | 0.830 | *0.86* | **606** | 0.0073 | 0.864 | *0.83* |
| **372** | 0.0413 | 0.816 | *0.75* | **490** | 0.0310 | 0.832 | *0.86* | **608** | 0.0074 | 0.868 | *0.83* |
| **374** | 0.0413 | 0.820 | *0.77* | **492** | 0.0303 | 0.833 | *0.86* | **610** | 0.0075 | 0.875 | *0.84* |
| **376** | 0.0418 | 0.820 | *0.78* | **494** | 0.0295 | 0.835 | *0.86* | **612** | 0.0076 | 0.881 | *0.85* |
| **378** | 0.0427 | 0.818 | *0.78* | **496** | 0.0286 | 0.837 | *0.87* | **614** | 0.0077 | 0.885 | *0.85* |



| | | | | | | | | | | |
|---|---|---|---|---|---|---|---|---|---|---|
| **380** | 0.0434 | 0.817 | *0.79* | **498** | 0.0277 | 0.838 | *0.87* | **616** | 0.0078 | 0.885 | *0.85* |
| **382** | 0.0437 | 0.818 | *0.79* | **500** | 0.0268 | 0.840 | *0.87* | **618** | 0.0080 | 0.885 | *0.85* |
| **384** | 0.0440 | 0.819 | *0.80* | **502** | 0.0257 | 0.842 | *0.87* | **620** | 0.0082 | 0.886 | *0.86* |
| **386** | 0.0444 | 0.819 | *0.80* | **504** | 0.0246 | 0.844 | *0.87* | **622** | 0.0083 | 0.887 | *0.86* |
| **388** | 0.0449 | 0.819 | *0.80* | **506** | 0.0236 | 0.845 | *0.87* | **624** | 0.0084 | 0.886 | *0.86* |
| **390** | 0.0451 | 0.818 | *0.80* | **508** | 0.0227 | 0.844 | *0.87* | **626** | 0.0085 | 0.886 | *0.86* |
| **392** | 0.0454 | 0.818 | *0.81* | **510** | 0.0217 | 0.844 | *0.87* | **628** | 0.0085 | 0.888 | *0.87* |
| **394** | 0.0458 | 0.818 | *0.81* | **512** | 0.0208 | 0.844 | *0.86* | **630** | 0.0085 | 0.889 | *0.87* |
| **396** | 0.0462 | 0.819 | *0.81* | **514** | 0.0199 | 0.844 | *0.86* | **632** | 0.0085 | 0.888 | *0.87* |
| **398** | 0.0465 | 0.819 | *0.81* | **516** | 0.0191 | 0.845 | *0.86* | **634** | 0.0086 | 0.888 | *0.87* |
| **400** | 0.0468 | 0.820 | *0.82* | **518** | 0.0183 | 0.845 | *0.86* | **636** | 0.0087 | 0.888 | *0.87* |
| **402** | 0.0473 | 0.820 | *0.82* | **520** | 0.0176 | 0.845 | *0.86* | **638** | 0.0088 | 0.887 | *0.87* |
| **404** | 0.0481 | 0.821 | *0.83* | **522** | 0.0170 | 0.845 | *0.86* | **640** | 0.0089 | 0.881 | *0.87* |
| **406** | 0.0491 | 0.820 | *0.83* | **524** | 0.0165 | 0.844 | *0.86* | **642** | 0.0091 | 0.875 | *0.87* |
| **408** | 0.0502 | 0.819 | *0.83* | **526** | 0.0160 | 0.843 | *0.85* | **644** | 0.0092 | 0.871 | *0.87* |
| **410** | 0.0512 | 0.817 | *0.84* | **528** | 0.0156 | 0.843 | *0.85* | **646** | 0.0093 | 0.869 | *0.87* |
| **412** | 0.0521 | 0.817 | *0.84* | **530** | 0.0151 | 0.843 | *0.85* | **648** | 0.0094 | 0.868 | *0.86* |
| **414** | 0.0528 | 0.817 | *0.84* | **532** | 0.0147 | 0.844 | *0.85* | **650** | 0.0096 | 0.869 | *0.86* |
| **416** | 0.0533 | 0.817 | *0.84* | **534** | 0.0144 | 0.844 | *0.85* | **652** | 0.0099 | 0.873 | *0.87* |
| **418** | 0.0537 | 0.816 | *0.84* | **536** | 0.0141 | 0.843 | *0.85* | **654** | 0.0104 | 0.877 | *0.87* |
| **420** | 0.0539 | 0.814 | *0.84* | **538** | 0.0139 | 0.841 | *0.85* | **656** | 0.0112 | 0.881 | *0.88* |
| **422** | 0.0541 | 0.814 | *0.85* | **540** | 0.0137 | 0.839 | *0.85* | **658** | 0.0121 | 0.885 | *0.88* |
| **424** | 0.0544 | 0.813 | *0.85* | **542** | 0.0134 | 0.836 | *0.85* | **660** | 0.0133 | 0.888 | *0.89* |
| **426** | 0.0548 | 0.812 | *0.85* | **544** | 0.0132 | 0.832 | *0.84* | **662** | 0.0148 | 0.889 | *0.89* |
| **428** | 0.0552 | 0.811 | *0.85* | **546** | 0.0129 | 0.829 | *0.84* | **664** | 0.0164 | 0.889 | *0.89* |
| **430** | 0.0556 | 0.810 | *0.85* | **548** | 0.0126 | 0.827 | *0.84* | **666** | 0.0181 | 0.888 | *0.90* |
| **432** | 0.0561 | 0.809 | *0.85* | **550** | 0.0123 | 0.825 | *0.84* | **668** | 0.0197 | 0.887 | *0.90* |
| **434** | 0.0565 | 0.809 | *0.86* | **552** | 0.0120 | 0.823 | *0.83* | **670** | 0.0208 | 0.888 | *0.90* |
| **436** | 0.0565 | 0.809 | *0.86* | **554** | 0.0116 | 0.821 | *0.83* | **672** | 0.0216 | 0.889 | *0.90* |
| **438** | 0.0561 | 0.810 | *0.86* | **556** | 0.0113 | 0.819 | *0.83* | **674** | 0.0220 | 0.889 | *0.90* |
| **440** | 0.0553 | 0.812 | *0.86* | **558** | 0.0110 | 0.817 | *0.82* | **675** | 0.0220 | 0.889 | *0.90* |
| **442** | 0.0541 | 0.813 | *0.86* | **560** | 0.0107 | 0.816 | *0.82* | **676** | 0.0219 | 0.889 | *0.90* |
| **444** | 0.0525 | 0.815 | *0.86* | **562** | 0.0105 | 0.816 | *0.82* | **678** | 0.0214 | 0.889 | *0.90* |




| 446 | 0.0506 | 0.817 | *0.86* | 564 | 0.0102 | 0.819 | *0.82* | 680 | 0.0204 | 0.889 | *0.90* |
| 448 | 0.0486 | 0.819 | *0.86* | 566 | 0.0100 | 0.822 | *0.82* | 682 | 0.0190 | 0.890 | *0.90* |
| 450 | 0.0469 | 0.821 | *0.86* | 568 | 0.0098 | 0.824 | *0.82* | 684 | 0.0170 | 0.894 | *0.90* |
| 452 | 0.0453 | 0.824 | *0.86* | 570 | 0.0096 | 0.826 | *0.82* | 686 | 0.0146 | 0.898 | *0.90* |
| 454 | 0.0440 | 0.826 | *0.86* | 572 | 0.0094 | 0.827 | *0.82* | 688 | 0.0121 | 0.902 | *0.90* |
| 456 | 0.0429 | 0.827 | *0.86* | 574 | 0.0091 | 0.831 | *0.82* | 690 | 0.0098 | 0.904 | *0.89* |
| 458 | 0.0421 | 0.828 | *0.86* | 576 | 0.0089 | 0.836 | *0.83* | 692 | 0.0079 | 0.905 | *0.88* |
| 460 | 0.0416 | 0.827 | *0.86* | 578 | 0.0086 | 0.843 | *0.83* | 694 | 0.0062 | 0.906 | *0.86* |
| 462 | 0.0412 | 0.825 | *0.86* | 580 | 0.0082 | 0.851 | *0.83* | 696 | 0.0049 | 0.909 | *0.84* |
| 464 | 0.0407 | 0.824 | *0.86* | 582 | 0.0079 | 0.857 | *0.83* | 698 | 0.0039 | 0.912 | *0.81* |
| 466 | 0.0401 | 0.823 | *0.86* | 584 | 0.0078 | 0.859 | *0.83* | 700 | 0.0031 | 0.913 | *0.76* |

**Table A2**. Spectral coefficients *A* and *E* of the one-component parameterization described by Eq. (8.a) in its variants obtained in selected months and the corresponding determination coefficients $R^2$.

| λ [nm] | March | | | April | | | May | | | September | | |
|---|---|---|---|---|---|---|---|---|---|---|---|---|
| *1* | *A* | *E* | $R^2$ | *A* | *E* | $R^2$ | *A* | *E* | $R^2$ | *A* | *E* | $R^2$ |
| | *2* | *3* | *4* | *2* | *3* | *4* | *2* | *3* | *4* | *2* | *3* | *4* |
| 350 | 0.0368 | 0.971 | *0.75* | 0.0312 | 0.816 | *0.77* | 0.0352 | 0.876 | *0.82* | 0.0273 | 0.911 | *0.74* |
| 355 | 0.0371 | 0.961 | *0.75* | 0.0298 | 0.826 | *0.78* | 0.0352 | 0.866 | *0.83* | 0.0285 | 0.902 | *0.75* |
| 360 | 0.0371 | 0.959 | *0.75* | 0.0302 | 0.814 | *0.77* | 0.0364 | 0.865 | *0.83* | 0.0311 | 0.874 | *0.75* |
| 365 | 0.0384 | 0.946 | *0.75* | 0.0309 | 0.826 | *0.78* | 0.0382 | 0.865 | *0.83* | 0.0354 | 0.846 | *0.75* |
| 370 | 0.0396 | 0.945 | *0.77* | 0.0318 | 0.823 | *0.77* | 0.0399 | 0.866 | *0.84* | 0.0384 | 0.836 | *0.77* |
| 375 | 0.0395 | 0.950 | *0.81* | 0.0292 | 0.869 | *0.82* | 0.0412 | 0.865 | *0.86* | 0.0395 | 0.833 | *0.82* |
| 380 | 0.0412 | 0.943 | *0.82* | 0.0297 | 0.880 | *0.84* | 0.0441 | 0.853 | *0.86* | 0.0429 | 0.815 | *0.82* |
| 385 | 0.0417 | 0.945 | *0.83* | 0.0302 | 0.885 | *0.85* | 0.0452 | 0.853 | *0.87* | 0.0451 | 0.803 | *0.83* |
| 390 | 0.0421 | 0.943 | *0.83* | 0.0309 | 0.886 | *0.85* | 0.0462 | 0.851 | *0.88* | 0.0472 | 0.792 | *0.83* |
| 395 | 0.0430 | 0.936 | *0.83* | 0.0315 | 0.890 | *0.85* | 0.0469 | 0.853 | *0.88* | 0.0495 | 0.784 | *0.83* |
| 400 | 0.0439 | 0.935 | *0.84* | 0.0311 | 0.903 | *0.86* | 0.0471 | 0.858 | *0.89* | 0.0514 | 0.777 | *0.84* |
| 405 | 0.0452 | 0.934 | *0.85* | 0.0318 | 0.916 | *0.87* | 0.0487 | 0.858 | *0.89* | 0.0548 | 0.766 | *0.85* |
| 410 | 0.0476 | 0.921 | *0.85* | 0.0333 | 0.920 | *0.87* | 0.0519 | 0.854 | *0.89* | 0.0589 | 0.756 | *0.86* |
| 415 | 0.0489 | 0.914 | *0.86* | 0.0346 | 0.923 | *0.88* | 0.0541 | 0.852 | *0.90* | 0.0624 | 0.750 | *0.86* |
| 420 | 0.0496 | 0.905 | *0.86* | 0.0350 | 0.927 | *0.87* | 0.0554 | 0.847 | *0.90* | 0.0642 | 0.745 | *0.87* |
| 425 | 0.0499 | 0.898 | *0.87* | 0.0353 | 0.931 | *0.88* | 0.0569 | 0.842 | *0.90* | 0.0658 | 0.737 | *0.87* |





| | | | | | | | | | | | |
|---|---|---|---|---|---|---|---|---|---|---|---|
| **430** | 0.0504 | 0.893 | *0.87* | 0.0354 | 0.936 | *0.89* | 0.0586 | 0.837 | *0.90* | 0.0681 | 0.728 | *0.87* |
| **435** | 0.0508 | 0.889 | *0.87* | 0.0356 | 0.942 | *0.89* | 0.0608 | 0.832 | *0.90* | 0.0703 | 0.721 | *0.88* |
| **440** | 0.0493 | 0.892 | *0.88* | 0.0351 | 0.945 | *0.89* | 0.0602 | 0.835 | *0.90* | 0.0698 | 0.718 | *0.88* |
| **445** | 0.0458 | 0.900 | *0.88* | 0.0329 | 0.951 | *0.89* | 0.0568 | 0.834 | *0.90* | 0.0648 | 0.720 | *0.88* |
| **450** | 0.0417 | 0.912 | *0.88* | 0.0300 | 0.959 | *0.89* | 0.0514 | 0.841 | *0.90* | 0.0585 | 0.722 | *0.88* |
| **455** | 0.0386 | 0.921 | *0.88* | 0.0282 | 0.965 | *0.89* | 0.0481 | 0.844 | *0.90* | 0.0543 | 0.719 | *0.89* |
| **460** | 0.0366 | 0.926 | *0.88* | 0.0268 | 0.973 | *0.90* | 0.0464 | 0.841 | *0.90* | 0.0525 | 0.712 | *0.89* |
| **465** | 0.0354 | 0.918 | *0.88* | 0.0259 | 0.974 | *0.90* | 0.0449 | 0.839 | *0.90* | 0.0516 | 0.705 | *0.89* |
| **470** | 0.0338 | 0.913 | *0.88* | 0.0242 | 0.980 | *0.90* | 0.0428 | 0.840 | *0.89* | 0.0501 | 0.698 | *0.89* |
| **475** | 0.0318 | 0.911 | *0.88* | 0.0226 | 0.979 | *0.89* | 0.0405 | 0.839 | *0.89* | 0.0474 | 0.702 | *0.89* |
| **480** | 0.0356 | 0.915 | *0.88* | 0.0211 | 0.979 | *0.89* | 0.0382 | 0.842 | *0.89* | 0.0448 | 0.705 | *0.89* |
| **485** | 0.0281 | 0.920 | *0.88* | 0.0196 | 0.988 | *0.89* | 0.0364 | 0.843 | *0.89* | 0.0423 | 0.708 | *0.89* |
| **490** | 0.0266 | 0.930 | *0.88* | 0.0188 | 0.990 | *0.90* | 0.0346 | 0.844 | *0.90* | 0.0399 | 0.715 | *0.89* |
| **495** | 0.0250 | 0.939 | *0.89* | 0.0178 | 0.996 | *0.90* | 0.0323 | 0.845 | *0.90* | 0.0370 | 0.722 | *0.89* |
| **500** | 0.0233 | 0.945 | *0.89* | 0.0166 | 0.998 | *0.91* | 0.0291 | 0.853 | *0.91* | 0.0336 | 0.727 | *0.88* |
| **505** | 0.0213 | 0.950 | *0.89* | 0.0150 | 0.999 | *0.91* | 0.0255 | 0.862 | *0.91* | 0.0299 | 0.734 | *0.88* |
| **510** | 0.0193 | 0.952 | *0.88* | 0.0139 | 0.989 | *0.90* | 0.0230 | 0.859 | *0.91* | 0.0263 | 0.741 | *0.87* |
| **515** | 0.0173 | 0.960 | *0.88* | 0.0124 | 0.992 | *0.90* | 0.0203 | 0.862 | *0.91* | 0.0233 | 0.744 | *0.87* |
| **520** | 0.0157 | 0.964 | *0.88* | 0.0109 | 1.000 | *0.90* | 0.0179 | 0.864 | *0.90* | 0.0207 | 0.748 | *0.86* |
| **525** | 0.0147 | 0.958 | *0.88* | 0.0101 | 0.999 | *0.90* | 0.0162 | 0.864 | *0.90* | 0.0192 | 0.744 | *0.86* |
| **530** | 0.0139 | 0.953 | *0.88* | 0.0092 | 1.005 | *0.89* | 0.0148 | 0.867 | *0.90* | 0.0180 | 0.746 | *0.86* |
| **535** | 0.0132 | 0.953 | *0.87* | 0.0086 | 1.008 | *0.90* | 0.0133 | 0.881 | *0.90* | 0.0173 | 0.738 | *0.85* |
| **540** | 0.0128 | 0.939 | *0.87* | 0.0080 | 1.015 | *0.89* | 0.0125 | 0.881 | *0.90* | 0.0168 | 0.729 | *0.85* |
| **545** | 0.0121 | 0.934 | *0.87* | 0.0076 | 1.006 | *0.89* | 0.0116 | 0.875 | *0.90* | 0.0163 | 0.721 | *0.84* |
| **550** | 0.0113 | 0.931 | *0.87* | 0.0070 | 1.001 | *0.88* | 0.0106 | 0.878 | *0.90* | 0.0156 | 0.713 | *0.83* |
| **555** | 0.0106 | 0.917 | *0.86* | 0.0067 | 0.983 | *0.88* | 0.0097 | 0.872 | *0.89* | 0.0144 | 0.719 | *0.82* |
| **560** | 0.0123 | 0.911 | *0.87* | 0.0061 | 0.970 | *0.87* | 0.0090 | 0.873 | *0.89* | 0.0134 | 0.728 | *0.82* |
| **565** | 0.0092 | 0.910 | *0.85* | 0.0059 | 0.953 | *0.87* | 0.0081 | 0.892 | *0.88* | 0.0126 | 0.738 | *0.82* |
| **570** | 0.0089 | 0.903 | *0.85* | 0.0059 | 0.929 | *0.87* | 0.0075 | 0.906 | *0.88* | 0.0117 | 0.763 | *0.82* |
| **575** | 0.0084 | 0.912 | *0.85* | 0.0059 | 0.915 | *0.87* | 0.0074 | 0.904 | *0.89* | 0.0104 | 0.796 | *0.83* |
| **580** | 0.0081 | 0.911 | *0.84* | 0.0055 | 0.925 | *0.89* | 0.0065 | 0.934 | *0.87* | 0.0088 | 0.837 | *0.84* |
| **585** | 0.0077 | 0.930 | *0.85* | 0.0056 | 0.912 | *0.88* | 0.0068 | 0.908 | *0.89* | 0.0078 | 0.874 | *0.85* |
| **590** | 0.0076 | 0.931 | *0.85* | 0.0055 | 0.911 | *0.88* | 0.0071 | 0.902 | *0.89* | 0.0080 | 0.867 | *0.86* |



| | | | | | | | | | | | | |
|---|---|---|---|---|---|---|---|---|---|---|---|---|
| **595** | 0.0074 | 0.925 | *0.84* | 0.0053 | 0.907 | *0.88* | 0.0066 | 0.916 | *0.88* | 0.0077 | 0.873 | *0.85* |
| **600** | 0.0073 | 0.913 | *0.83* | 0.0055 | 0.872 | *0.85* | 0.0060 | 0.933 | *0.89* | 0.0075 | 0.881 | *0.85* |
| **605** | 0.0074 | 0.910 | *0.83* | 0.0056 | 0.863 | *0.86* | 0.0059 | 0.946 | *0.89* | 0.0076 | 0.883 | *0.84* |
| **610** | 0.0075 | 0.920 | *0.84* | 0.0060 | 0.864 | *0.87* | 0.0063 | 0.942 | *0.89* | 0.0076 | 0.899 | *0.84* |
| **615** | 0.0078 | 0.935 | *0.85* | 0.0061 | 0.882 | *0.87* | 0.0063 | 0.971 | *0.87* | 0.0080 | 0.902 | *0.86* |
| **620** | 0.0081 | 0.941 | *0.86* | 0.0062 | 0.906 | *0.87* | 0.0071 | 0.943 | *0.90* | 0.0086 | 0.893 | *0.86* |
| **625** | 0.0084 | 0.933 | *0.87* | 0.0067 | 0.907 | *0.89* | 0.0076 | 0.928 | *0.90* | 0.0090 | 0.886 | *0.86* |
| **630** | 0.0085 | 0.931 | *0.86* | 0.0071 | 0.893 | *0.89* | 0.0072 | 0.948 | *0.91* | 0.0091 | 0.883 | *0.86* |
| **635** | 0.0086 | 0.935 | *0.87* | 0.0071 | 0.897 | *0.90* | 0.0074 | 0.950 | *0.91* | 0.0093 | 0.878 | *0.87* |
| **640** | 0.0084 | 0.921 | *0.87* | 0.0073 | 0.894 | *0.90* | 0.0078 | 0.944 | *0.91* | 0.0099 | 0.861 | *0.86* |
| **645** | 0.0090 | 0.895 | *0.86* | 0.0072 | 0.895 | *0.90* | 0.0080 | 0.935 | *0.91* | 0.0106 | 0.848 | *0.86* |
| **650** | 0.0093 | 0.899 | *0.85* | 0.0072 | 0.903 | *0.89* | 0.0081 | 0.948 | *0.90* | 0.0116 | 0.830 | *0.85* |
| **655** | 0.0106 | 0.907 | *0.86* | 0.0082 | 0.915 | *0.90* | 0.0093 | 0.951 | *0.91* | 0.0130 | 0.831 | *0.87* |
| **660** | 0.0129 | 0.932 | *0.88* | 0.0101 | 0.934 | *0.91* | 0.0123 | 0.946 | *0.92* | 0.0161 | 0.829 | *0.88* |
| **665** | 0.0168 | 0.933 | *0.88* | 0.0130 | 0.954 | *0.92* | 0.0165 | 0.942 | *0.93* | 0.0215 | 0.809 | *0.89* |
| **670** | 0.0200 | 0.938 | *0.89* | 0.0155 | 0.966 | *0.92* | 0.0208 | 0.929 | *0.93* | 0.0266 | 0.791 | *0.89* |
| **675** | 0.0211 | 0.937 | *0.89* | 0.0164 | 0.970 | *0.93* | 0.0220 | 0.927 | *0.94* | 0.0284 | 0.787 | *0.89* |
| **680** | 0.0195 | 0.928 | *0.89* | 0.0152 | 0.966 | *0.93* | 0.0201 | 0.933 | *0.94* | 0.0265 | 0.789 | *0.89* |
| **685** | 0.0153 | 0.926 | *0.89* | 0.0120 | 0.954 | *0.92* | 0.0153 | 0.944 | *0.94* | 0.0196 | 0.817 | *0.88* |
| **690** | 0.0096 | 0.938 | *0.89* | 0.0077 | 0.932 | *0.91* | 0.0089 | 0.967 | *0.93* | 0.0112 | 0.862 | *0.86* |
| **695** | 0.0055 | 0.956 | *0.86* | 0.0045 | 0.908 | *0.89* | 0.0047 | 0.981 | *0.90* | 0.0057 | 0.899 | *0.82* |
| **700** | 0.0031 | 0.979 | *0.81* | 0.0023 | 0.906 | *0.82* | 0.0022 | 1.030 | *0.81* | 0.0029 | 0.938 | *0.72* |

**Table A3.** Spectral coefficients $A_0$ and $K$ of the two-component parameterization described by Eq. (10). The other coefficient $E$ is the same as in the case of the one-component parameterization (see Table A1).

| λ [nm] | $A_0$ | $K$ | λ [nm] | $A_0$ | $K$ | λ [nm] | $A_0$ | $K$ |
|---|---|---|---|---|---|---|---|---|
| *1* | *2* | *3* | *1* | *2* | *3* | *1* | *2* | *3* |
| **350** | 0.0356 | -0.007 | **468** | 0.0228 | -0.831 | **586** | 0.0075 | -0.053 |
| **352** | 0.0358 | -0.007 | **470** | 0.0223 | -0.836 | **588** | 0.0074 | -0.094 |
| **354** | 0.0361 | -0.005 | **472** | 0.0218 | -0.840 | **590** | 0.0072 | -0.127 |
| **356** | 0.0363 | -0.007 | **474** | 0.0213 | -0.842 | **592** | 0.0070 | -0.139 |
| **358** | 0.0364 | -0.018 | **476** | 0.0208 | -0.843 | **594** | 0.0070 | -0.130 |
| **360** | 0.0365 | -0.038 | **478** | 0.0203 | -0.842 | **596** | 0.0069 | -0.112 |

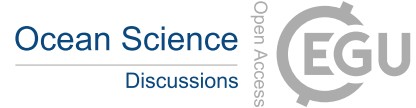

| | | | | | | | | |
|---|---|---|---|---|---|---|---|---|
| **362** | 0.0368 | -0.058 | **480** | 0.0199 | -0.840 | **598** | 0.0069 | -0.093 |
| **364** | 0.0371 | -0.077 | **482** | 0.0195 | -0.838 | **600** | 0.0069 | -0.063 |
| **366** | 0.0376 | -0.095 | **484** | 0.0191 | -0.833 | **602** | 0.0071 | -0.034 |
| **368** | 0.0379 | -0.119 | **486** | 0.0188 | -0.822 | **604** | 0.0072 | -0.018 |
| **370** | 0.0375 | -0.151 | **488** | 0.0186 | -0.805 | **606** | 0.0072 | -0.019 |
| **372** | 0.0367 | -0.181 | **490** | 0.0185 | -0.782 | **608** | 0.0073 | -0.020 |
| **374** | 0.0361 | -0.204 | **492** | 0.0184 | -0.758 | **610** | 0.0074 | -0.014 |
| **376** | 0.0360 | -0.228 | **494** | 0.0182 | -0.734 | **612** | 0.0076 | 0.000 |
| **378** | 0.0361 | -0.257 | **496** | 0.0180 | -0.706 | **614** | 0.0078 | 0.017 |
| **380** | 0.0360 | -0.285 | **498** | 0.0178 | -0.675 | **616** | 0.0079 | 0.020 |
| **382** | 0.0358 | -0.303 | **500** | 0.0175 | -0.643 | **618** | 0.0080 | -0.004 |
| **384** | 0.0359 | -0.309 | **502** | 0.0172 | -0.614 | **620** | 0.0079 | -0.045 |
| **386** | 0.0362 | -0.314 | **504** | 0.0167 | -0.587 | **622** | 0.0079 | -0.080 |
| **388** | 0.0363 | -0.323 | **506** | 0.0163 | -0.564 | **624** | 0.0080 | -0.083 |
| **390** | 0.0362 | -0.334 | **508** | 0.0158 | -0.546 | **626** | 0.0081 | -0.071 |
| **392** | 0.0362 | -0.345 | **510** | 0.0153 | -0.530 | **628** | 0.0081 | -0.061 |
| **394** | 0.0362 | -0.359 | **512** | 0.0149 | -0.512 | **630** | 0.0081 | -0.066 |
| **396** | 0.0362 | -0.372 | **514** | 0.0144 | -0.494 | **632** | 0.0081 | -0.082 |
| **398** | 0.0362 | -0.384 | **516** | 0.0139 | -0.477 | **634** | 0.0081 | -0.096 |
| **400** | 0.0361 | -0.395 | **518** | 0.0135 | -0.465 | **636** | 0.0081 | -0.109 |
| **402** | 0.0362 | -0.408 | **520** | 0.0130 | -0.455 | **638** | 0.0081 | -0.121 |
| **404** | 0.0364 | -0.423 | **522** | 0.0127 | -0.443 | **640** | 0.0081 | -0.144 |
| **406** | 0.0369 | -0.435 | **524** | 0.0124 | -0.428 | **642** | 0.0082 | -0.167 |
| **408** | 0.0374 | -0.447 | **526** | 0.0122 | -0.411 | **644** | 0.0081 | -0.190 |
| **410** | 0.0378 | -0.460 | **528** | 0.0121 | -0.389 | **646** | 0.0081 | -0.202 |
| **412** | 0.0382 | -0.472 | **530** | 0.0119 | -0.361 | **648** | 0.0082 | -0.209 |
| **414** | 0.0384 | -0.483 | **532** | 0.0118 | -0.333 | **650** | 0.0084 | -0.209 |
| **416** | 0.0385 | -0.493 | **534** | 0.0118 | -0.306 | **652** | 0.0087 | -0.202 |
| **418** | 0.0385 | -0.504 | **536** | 0.0117 | -0.281 | **654** | 0.0091 | -0.207 |
| **420** | 0.0384 | -0.518 | **538** | 0.0118 | -0.254 | **656** | 0.0096 | -0.231 |
| **422** | 0.0382 | -0.533 | **540** | 0.0118 | -0.227 | **658** | 0.0102 | -0.265 |
| **424** | 0.0380 | -0.548 | **542** | 0.0118 | -0.200 | **660** | 0.0109 | -0.299 |
| **426** | 0.0378 | -0.563 | **544** | 0.0118 | -0.172 | **662** | 0.0119 | -0.330 |





| | | | | | | | | |
|---|---|---|---|---|---|---|---|---|
| **428** | 0.0377 | -0.580 | **546** | 0.0117 | -0.143 | **664** | 0.0129 | -0.372 |
| **430** | 0.0376 | -0.596 | **548** | 0.0117 | -0.114 | **666** | 0.0137 | -0.428 |
| **432** | 0.0376 | -0.610 | **550** | 0.0116 | -0.087 | **668** | 0.0143 | -0.485 |
| **434** | 0.0374 | -0.627 | **552** | 0.0114 | -0.069 | **670** | 0.0148 | -0.524 |
| **436** | 0.0370 | -0.646 | **554** | 0.0112 | -0.055 | **672** | 0.0152 | -0.542 |
| **438** | 0.0363 | -0.664 | **556** | 0.0110 | -0.041 | **674** | 0.0154 | -0.542 |
| **440** | 0.0355 | -0.677 | **558** | 0.0108 | -0.027 | **675** | 0.0155 | -0.537 |
| **442** | 0.0345 | -0.686 | **560** | 0.0106 | -0.015 | **676** | 0.0155 | -0.531 |
| **444** | 0.0333 | -0.695 | **562** | 0.0105 | 0.000 | **678** | 0.0152 | -0.515 |
| **446** | 0.0319 | -0.702 | **564** | 0.0103 | 0.019 | **680** | 0.0148 | -0.491 |
| **448** | 0.0305 | -0.710 | **566** | 0.0102 | 0.039 | **682** | 0.0141 | -0.453 |
| **450** | 0.0292 | -0.721 | **568** | 0.0101 | 0.052 | **684** | 0.0131 | -0.393 |
| **452** | 0.0279 | -0.736 | **570** | 0.0099 | 0.050 | **686** | 0.0119 | -0.310 |
| **454** | 0.0268 | -0.754 | **572** | 0.0095 | 0.025 | **688** | 0.0105 | -0.217 |
| **456** | 0.0258 | -0.774 | **574** | 0.0091 | -0.006 | **690** | 0.0090 | -0.132 |
| **458** | 0.0250 | -0.791 | **576** | 0.0088 | -0.021 | **692** | 0.0076 | -0.054 |
| **460** | 0.0245 | -0.804 | **578** | 0.0085 | -0.007 | **694** | 0.0064 | 0.042 |
| **462** | 0.0242 | -0.811 | **580** | 0.0082 | 0.014 | **696** | 0.0054 | 0.157 |
| **464** | 0.0238 | -0.817 | **582** | 0.0079 | 0.012 | **698** | 0.0046 | 0.260 |
| **466** | 0.0233 | -0.825 | **584** | 0.0077 | -0.014 | **700** | 0.0038 | 0.306 |

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

## Acknowledgements

The data used in this work were gathered with financial assistance from the "SatBałtyk" project funded by the European Union through the European Regional Development Fund, (No. POIG.01.01.02-22-011/09, 'The Satellite Monitoring of the Baltic Sea Environment') and within the framework of the Statutory Research Project (No. I.1 and I.2) of the Institute of Oceanology Polish Academy of Sciences, to name but two sources. The subsequent analyses of the data were carried out as part of project N N306 041136, financed by the Polish Ministry of Science and Higher Education in 2009-2014 (grants awarded to B.W. and J.M.), and also the project funded by the National Science Centre, Poland, entitled "Advanced research into the relationships between optical, biogeochemical and physical properties of suspended particulate matter in the southern Baltic Sea" (contract No. 2016/21/B/ST10/02381) (awarded to S.B.W.) We thank Barbara Lednicka, Monika Zabłocka, Agnieszka Zdun and other colleagues from IOPAS for their help in collecting the empirical material.

The authors declare that they have no conflict of interest.