# Peer review of "Parameterization of the spectral light absorption coefficient of phytoplankton in the Baltic Sea: general, monthly and twocomponent variants of approximation formulas"

_Ocean Science, 2018_

## Referee Comment (RC1) · Anonymous Referee #1 · 2 Aug 2018

The article, **Parameterization of the spectral light absorption coefficient of phytoplankton in the Baltic Sea: general, monthly and two component variants of approximation formulas** by **Justina Meler**, presents a way of accounting for the variability in phytoplankton absorption coefficients using an additional term (∑Ci/Tchla) in the parameterization. A good attempt, as against the traditional method that takes into account only pigment concentrations. The dataset clearly presents the complexities and challenges, coastal systems have to offer in ocean color remote sensing, wherein regional parameterizations are highly essential. The article is therefore of relevance to the scientific community. However, in its current form it is lengthy (needs to be concise) and contains too many figures. It is therefore requested to perform major revisions (listed below), thoroughly check the text and get it checked from a native speaker.

**Major issues:**

- In the introduction, relevant background knowledge on how the ratio (∑Ci/Tchla) would account for the variability in phytoplankton absorption is missing. Its implications in inferring ecosystem dynamics etc. in different situations. What do high and low values of the ratio imply?

- Parameterizations are developed and tested on the same dataset. How about testing on an independent dataset on an independent dataset (especially when the authors have a huge dataset)? Test the parameterizations on a dataset (from the reported sampling sites, may be set aside data from a year) that was not included in developing the coefficients presented in tables A1, A2 and A3; refer to Mascarenhas et al. 2018 (http://www.mdpi.com/2072-4292/10/6/977/pdf)

- A different approach (than the monthly parameterizations) would be considering different ranges of Tchla concentration and corresponding values of the ratio (∑Ci/Tchla). This will provide an understanding of the effect of concentration ranges (low-medium-high) on parameterization parameters.

- In the introduction, in addition to studies in the Baltic and the Black Sea, consider also those of other ocean basins. For e.g. refer to the following
  o Mascarenhas et al. 2018 Parameterization of Spectral Particulate and Phytoplankton Absorption Coefficients in Sognefjord and Trondheimsfjord
  o Nima et al. 2016 Absorption properties of high-latitude norwegian coastal water
  o Stramska et al 2003 Bio-optical relationships and ocean color algorithms for the north polar region of the Atlantic.

- Matsuoka et al 2007 Bio-optical characteristics of the western arctic ocean: Implications for ocean color algorithms

- Figures could be combined representing the two different parameterization scenarios e.g. Fig 6 and Figures 11 c,d. Less relevant ones could be provided as supplementary material.

- Provide equations of trends in fig 6 and 11c,d.

- Figure captions need to be clearly distinguished from the normal text. Maintain appropriate spacing between the two.

- List out the objectives precisely in points instead of a paragraph.

- All along the article there is a constant effort to explain what this study has to offer in comparison to previously published works (using similar dataset). List out the features and make them clear to the reader in one instance, e.g. at the end of the introduction after listing your objectives or before. Avoid such statements in the methods or the results section.

- Pay attention to the formation of paragraphs (of a consistent size) throughout the article.

**Minor issues:**

- Watch over differences between British vs American English styles. Check for consistency throughout the article.

**Title**

Parameterization of the spectral light absorption coefficient of phytoplankton in the Baltic Sea: general, monthly and two component variants of approximation formulas

Instead,

**Parameterization of phytoplankton spectral absorption coefficients in the Baltic Sea**: general, monthly and two component variants of approximation formulas

**Page 1**

Line 8: approximate formulas, **empirical equations** instead

Line 12: varied between x and y; be precise no '> '

Line 18-20: sentence not clear, needs to be reframed.

Line 22: to fully describe **the process of** photosynthesis

Line 26: pigments  they contain.

**Page 2**

Line 1: …with which their populations absorb sunlight (**References???)**

Line 3: **of all these** relationships

Line 15: They proposed ….power function (provide the equation)

Line 20: empirical **data,** instead of empirical material

Line 20: **'case 1'** instead of "case 1" (single quote marks, also elsewhere in the manuscript)

Line 21: contents in enclosed parenthesis (chlorophyll a concentration ranging from 0.02 to 25mg m-3)

Line 27: concentration **ranging** from

Line 33: A more recent paper, Mascarenhas et al. 2018

**Page 3**

Line 16: 9 years or 10

Lines 22-31: instead of the paragraph, list the objectives as

Line 28: if possible??? This should be clear by now!

**Page 4**

Figure 1 caption: ……., **enlarged view of the enclosed area.**

**Page 5**

Lines 3 to 6: Authors attempt to emphasize differences in comparison to previously published works. This could be done at the end of section in brief. Not here and there or in the beginning of the section.

Line 13: pore size of GF/F

Line 15: kept  frozen

**Page 6**

Structure the paragraphs appropriately

**Page 7**

statistical formulas could be avoided, will help keep the length reasonable.

**Page 8**

Line 4:  **at** selected wavelengths

**Page 9**

Line 5: ….**and their ratio to Tchla**

**Page 14**

Figure 5 a,c:  provide detailed legends for every spectra

**Page 15**

Figure 6: (see the  legend  in panel (b))

**Page 19**

Line 1:   Two component parameterization

Line 2: ( Figure 3)

Line 5: As a step towards improving….

**Page 24**

Line 5: check the percentages

**Page 25**

Line 2: now we shall…..with those of **cite them here directly**

**Page 29**

Line 3: Importantly, **when** matched…..

**Page 30**

Line 1: seawater **optical** components

---

## Referee Comment (RC2) · Anonymous Referee #2 · 31 Aug 2018

General comments This paper is a continuation of a series of paper by the current authors on the parameterisation of the variability in phytoplankton absorption coefficients In the Baltic Sea. The analysis is extensive and thorough and is of some value for regional ocean colour remote sensing work.

I have the following major concerns: 1) The manuscript falls short of drawing necessary conclusion from the results presented here. There is very limited discussion on natural sources of observed variability, how much of it is real and what ecosystem processes can be inferred from findings. For example: does the data suggest that Printer-friendly version

an increased amount of photo-protective pigments influences the relationship in the summer months? And if not, why not?

2) I find myself unable to judge how much of the presented finding are actually novel and how much is a re-iteration of previous papers by the authors on the same dataset. Please clarify the progression from one paper to the next and highlight the additional value of this paper. Also justify why the results presented here were not published in previous paper.

3) The comparison of different parameterisations on this dataset does not add much value to the paper – especially if the performance is tested using the same dataset used for development.

4) The final recommendation as to which parameterization should be used (and under which conditions) is not clear. Quantify the benefit of the new parameterization and weigh it against processing time/costs.

5) Additionally, the language requires major revision (preferably from a native speaker) which should aim to shorten the article significantly and improve readability & understandability.

**Specific comments**

Explain the selection criteria for literature chosen for comparison. Can you use a more systematic approach to select a small (max. 5!) number of parametrisations from the hundreds available in the literature (e.g. only choose chase 2 waters or compare 2 case 1 and 2 case 2 water studies)? McKee et al. (2014), for example, did some analysis on the NOMAD data set which includes a variety of different water and geographic locations.

Revise terminology throughout article: - Use 'case 2 waters' rather than 'case 2' - Use 'at short wavelenths/in blue spectral region' instead of 'short-wave part of spectrum' - In my opinion, using 'absorption' rather than 'light absorption' is sufficient and would

OSD
improve readability

p. 4. Line 9: 'Because of 10 weather- and sea-state-related limitations, the proportion of data collected in open water regions exceeds ca 30% only for data collected in February, March, May, September and October (see Table 1).' – This is an example where the text can be shortened significantly: 'The amount of data collected in open waters was limited due to adverse weather (

analysis of data quality is required. Do you have any information with regards to bleach contamination resulting in low absorption values < 440 nm?

The absorption is expected to vary several orders of magnitude across all wavelengths and samples. This is not a new result. Interesting would be what the implications for your error analysis are? Do you see larger absolute errors for low absorption or do measurement errors increase linearly with strength of absorption?

p.9, I. 14: Use neutral and objective language (here and throughout document)! Another example: 'fairly well'.

p. 11, II. 4-5: Delete sentence, unnecessary self-citation at this point: 'We ourselves had already reported similar ranges of variability earlier (see Woźniak et al. 2011, Meler et al. 2016b, 5 2017a and b).'

Section 3.2.1: What happens if E(lambda) > 1?

Section 3.2.1: How can the annual variability in A be explained? Can it be linked to extent of packaging effect or the amount of photo-protective vs. photosynthetic pigments?

Section 3.2.1: What can be inferred from your analysis of R2? Does is correlate with any known ecosystem dynamics? Considerably shorting the paragraph unless you justify its value.

P.12 I. 8: Do you mean '690 - 700 nm'?

Section 3.2.1: What do you derive from the observed differences in performance compared to Bricaud's parameterisation? Why do you observe less variability and what does that mean for the ecosystem?

Please explain the metric of the colour index in more detail. What the reasons for the observed flattening (drop in colour index) of the absorption spectrum? Are you expecting a certain magnitude in your colour index drop and why?
p. 18: Shorten paragraph. Only highlight key finding shown in Table and consider deleting/rephrasing the following sentences: 'We have carried out such an analysis, but we do not present its detailed results due to the fact that they are very extensive. Here we limit ourselves only to stating that the use of general or monthly parameterizations for individual months 10 has little effect on the level of statistical error according to logarithmic statistics, although it may have, as generally expected, a very significant impact on the level of systematic error.'

Section 3.2.2 The use and exchange of a\_ph,cal/a\_ph,m and a\_ph in this section is confusing. What information can be gained from a\_ph,cal/a\_ph,m and how can the ratio be related to a\_ph in this context?

Fig. 8 (e) & (f): Justify the use of an exponential over a linear fit.

p. 27, II. 1-14: There is limited value in the comparison with other parameterisation using the same dataset (see comment above).

p. 27, II. 21 – end: The conclusion of superior performance based on the standard error factor cannot be justified. 'If we take into account both systematic and statistical errors, the superiority of the new parameterizations developed specifically for Baltic Sea conditions is undisputed.' Poor scientific language!

p. 29, 32: Delete sentences: 'Practical remote sensing algorithms are a good example in this respect. Their task is often to solve complicated "reverse" problems. Starting with measurements of sea colour, such algorithms, through intermediate steps during which different inherent optical properties of seawater (including light absorption) are retrieved, should yield basic features of different seawater components in the final stage of their operation. One such feature may be the concentration of chlorophyll a, which is often treated as a practical measure of the biomass of live phytoplankton contained in water. In our opinion the parameterizations presented in this work could be used in practice for such purposes.' OSD

---

## Author Comment (AC1)

**Responses to Reviewer #1**

**Dear Reviewer,**

Thank you very much for the extremely valuable critical comments on our work. Below we present our detailed point-by-point responses and the description of actions taken in regards to your comments. We believe that we have provided satisfactory explanations to your criticisms, and have made appropriate revisions in our paper.

**Reviewer's comment:**

The article, Parameterization of the spectral light absorption coefficient of phytoplankton in the Baltic Sea: general, monthly and two component variants of approximation formulas by Justina Meler, presents a way of accounting for the variability in phytoplankton absorption coefficients using an additional term ( $\Sigma$ Ci/Tchla) in the parameterization. A good attempt, as against the traditional method that takes into account only pigment concentrations. The dataset clearly presents the complexities and challenges, coastal systems have to offer in ocean color remote sensing, wherein regional parameterizations are highly essential. The article is therefore of relevance to the scientific community. However, in its current form it is lengthy (needs to be concise) and contains too many figures. It is therefore requested to perform major revisions (listed below), thoroughly check the text and get it checked from a native speaker.

**Author's response:**

Thank you for pointing out the positive aspects of our manuscript.

In order to make the work more concise, the new version of the manuscript text has been reorganized, some fragments has been shortened, and the number of drawings has been reduced. The new version of the text has also been corrected by a native speaker.

**Author's changes in the manuscript:**

The order of presentation of our own results has been changed. In section 3.2 one and twocomponent parameterizations are presented, while section 3.3 refers to errors of estimation. In the shortened section 3.4 we briefly refer to selected examples of parameterization known from the literature. Mathematical details regarding obtaining two-parameter parameterization have been moved to Appendix 2. The revised version of manuscript contains 9 reorganized figures in the main text, and one in the Appendix 2 (overall, the reorganized drawings contain 7 fewer individual panels than the original version).

**Reviewer's comment:**

Major issues: 1) In the introduction, relevant background knowledge on how the ratio  $(\Sigma Ci/Tchla)$  would account for the variability in phytoplankton absorption is missing. Its implications in inferring ecosystem dynamics etc. in different situations. What do high and low values of the ratio imply?

**Author's response:**

Generally, the composition of all accessory pigments may be different for various photosynthesizing marine species, it may reflect adaption of organism to different light conditions (processes of photo- and chromatic adaptations), as well as acclimation at the plant community level. The ratio ( $\Sigma$ Ci/Tchla) used by us for practical purposes allows, in the first approximation, to assess the differentiation between phytoplankton populations characterized by the same concentrations of the basic photosynthetic pigment - chlorophyll *a*.

**Author's changes in the manuscript:**

The information provided in the Introduction section has been supplemented.

**Reviewer's comment:**

2) Parameterizations are developed and tested on the same dataset. How about testing on an independent dataset on an independent dataset (especially when the authors have a huge dataset)? Test the parameterizations on a dataset (from the reported sampling sites, may be set aside data from a year) that was not included in developing the coefficients presented in tables A1, A2 and A3; refer to Mascarenhas et al. 2018 (http://www.mdpi.com/2072-4292/10/6/977/pdf)

**Author's response:**

Our main goal was to use all available information to determine new forms of parameterization tailored to the data collected on the Baltic, and we did not intend to carry out a strict validation. We realize that an independent data set would be needed for such an objective. We have tried to emphasize these issues in the improved manuscript. However, in order to satisfy the reviewer's curiosity, we conducted a supplementary calculation exercise. We divided our data dataset into a "training" and "validation" subsets. The "training" set consisted of 2/3 of data selected from each month. The remaining 1/3 of the data created a "validation" set. Using the "training" dataset we found alternative versions of one- and two-component parameterizations. We found that differences between A and E coefficients of one component parameterization were not more than 8% and 4% respectively (see additional Figure R1.a and b). Also relatively small were the differences between coefficients of two-component parameterizations (not shown). We found that these alternative versions of parameterizations have generally small systematic errors (either positive or negative, from -17% to +16.5%) (see Figure R1.c), and in terms of standard error factor x (characterizing the statistical error acc. to logarithmic statistics) they are only slightly less accurate in some portions of the spectrum (see Figure R1.d).

**Author's changes in the manuscript:**

The sentence stating that our intention was to briefly compare examples of parameterization known from the literature, and not to validate our own results has been added to the last paragraph of section 3.4 of the revised manuscript.

**Reviewer's comment:**

3) A different approach (than the monthly parameterizations) would be considering different ranges of Tchla concentration and corresponding values of the ratio ( $\Sigma$ Ci/Tchla). This will provide an understanding of the effect of concentration ranges (low-medium-high) on parameterization parameters.

**Author's response:**

Following the Reviewer's comment we have performed additional calculation exercise. The results are summarized in two additional Figures: R2 and R3. We have divided our dataset into separate groups according to different *Tchla* concentrations (33rd and 67th percentiles of *Tchla* distribution were used as limit values for these sets). However the obtained results reviled that changes in concentration ranges does not help to explain the differences in the classic one-component parameterizations matched for data from different months (see equations given in Figure R3).

**Author's changes in the manuscript:**

No changes were made to the manuscript in connection with this particular comment.

**Reviewer's comment:**

*4)* In the introduction, in addition to studies in the Baltic and the Black Sea, consider also those of other ocean basins. For e.g. refer to the following

- Mascarenhas et al. 2018 Parameterization of Spectral Particulate and Phytoplankton Absorption Coefficients in Sognefjord and Trondheimsfjord

- Nima et al. 2016 Absorption properties of high-latitude Norwegian coastal water

- Stramska et al 2003 Bio-optical relationships and ocean color algorithms for the north polar region of the Atlantic.

- Matsuoka et al 2007 Bio-optical characteristics of the western arctic ocean: Implications for ocean color algorithms

**Author's response:**

Thank you for indicating these literature items. They were all quoted in the manuscript. Some of them have been included in the examples demonstrating the diversity of coefficients reported in the literature.

**Author's changes in the manuscript:**

The indicated items are cited in the Introduction section. Selected items were used in section 3.4 (see Figures 4d and e, 5c and d, 9).

**Reviewer's comment:**

5) Figures could be combined representing the two different parameterization scenarios e.g. Fig 6 and Figures 11 c,d. Less relevant ones could be provided as supplementary material.

Author's response: Agreed.

**Author's changes in the manuscript:**

The suggested drawings have been merged according to the reviewer's suggestion (see new Figure 6). All figures have been rearranged, 7 panels from multi-panel figures have been removed, one six-panel figure has been moved to Appendix 2.

Reviewer's comment:*6)* Provide equations of trends in fig 6 and 11c,d.

Author's response: Agreed.

Author's changes in the manuscript: The equations have been added to the new Figure 6.

**Reviewer's comment:**

7) Figure captions need to be clearly distinguished from the normal text. Maintain appropriate spacing between the two.

**Author's response:**

We would like to apologize for this editing mistake, which appeared at the stage of creating an electronic document submitted to the editorial office of the journal.

Author's changes in the manuscript: The new version of the manuscript has been checked in this respect.

Reviewer's comment:*8)* List out the objectives precisely in points instead of a paragraph.

Author's response: Agreed.

Author's changes in the manuscript: The last paragraph of the Introduction section has been modified.

**Reviewer's comment:**

9) All along the article there is a constant effort to explain what this study has to offer in comparison to previously published works (using similar dataset). List out the features and make them clear to the reader in one instance, e.g. at the end of the introduction after listing your objectives or before. Avoid such statements in the methods or the results section.

Author's response: Agreed.

Author's changes in the manuscript:

Explanations which facts have already been documented in the previous work of our team have been grouped in the third paragraph of the Introduction section.

Reviewer's comment: 10) Pay attention to the formation of paragraphs (of a consistent size) throughout the article.

Author's response: Agreed.

Author's changes in the manuscript:

The sizes/lengths of the paragraphs have been modified (among others in section 2. Materials and methods).

Reviewer's comment:

Minor issues:- Watch over differences between British vs American English styles. Check for consistency throughout the article.

Author's changes in the manuscript: The language style has been checked (British English)

**Reviewer's comment:**

- Title

Parameterization of the spectral light absorption coefficient of phytoplankton in the Baltic Sea: general, monthly and two component variants of approximation formulas Instead,

**Parameterization of phytoplankton spectral absorption coefficients in the Baltic Sea**: general, monthly and two component variants of approximation formulas

Author's response: Agreed.

Author's changes in the manuscript:

The title has been corrected according to the Reviewer's suggestion. In the entire manuscript a shorter form was used: "phytoplankton absorption" (in this matter, the recommendation of the Reviewer #2 was also taken into account).

Reviewer's comment:

- Page 1 Line 8: approximate formulas, empirical equations instead Line 12: varied between x and y; be precise no '> '

Line 18-20: sentence not clear, needs to be reframed. Line 22: to fully describe **the process of** photosynthesis Line 26: pigments *it* they contain.

**Author's changes in the manuscript:**

Because the term 'approximation formulas' is used in title of the work, in the first sentence of the abstract, we decided to give both forms, i.e.: 'approximate formulas (empirical equations)". Other suggestions have been taken into account.

**Reviewer's comment:**

- Page 2

Line 1: ...with which their populations absorb sunlight (**References???**) Line 3: of all these relationships Line 15: They proposed ....power function (provide the equation) Line 20: empirical data, instead of empirical material Line 20: 'case 1' instead of "case 1" (single quote marks, also elsewhere in the manuscript) Line 21: contents in enclosed parenthesis (chlorophyll a concentration ranging from 0.02 to 25mg m-3) Line 27: concentration ranging from Line 33: A more recent paper, Mascarenhas et al. 2018

**Author's changes in the manuscript:**

(line1) - we have changed the place in the Introduction section, where the following literature items are mentioned for the first time: Morel and Bricaud 1981 and 1986; (line 3) - corrected; (line 15) - for brevity, we have provided the information that the appropriate mathematical formulas are given later in the text (see equation 3.a in Section 3.2); (line 21 and 27) - information on chlorophyll *a* ranges of was removed for brevity; (line 33) - the suggested position was added to the list of other examples.

**Reviewer's comment:**

- Page 3 Line 16: 9 years or 10 Lines 22-31: instead of the paragraph, list the objectives as Line 28: if possible??? This should be clear by now!

Author's changes in the manuscript:

(line 16) - the whole paragraph has been modified; (lines 22-31, line 28) - the paragraph has been modified according to these suggestions.

Reviewer's comment: - Page 4 Figure 1 caption: ....., enlarged view of the enclosed area.

Author's changes in the manuscript: Figure 1 caption has been modified.

Reviewer's comment:

- Page 5 Lines 3 to 6: Authors attempt to emphasize differences in comparison to previously published works. This could be done at the end of section in brief. Not here and there or in the beginning of the section. Line 13: pore size of GF/F Line 15: kept <del>deep</del> frozen

Author's changes in the manuscript:

(lines3 to 6) - as stated earlier all these information are now given in the third paragraph of the Introduction. (line 13) - information has been added; (line 15) - corrected.

Reviewer's comment: - Page 6 Structure the paragraphs appropriately

Author's changes in the manuscript: The sizes/lengths of the paragraphs have been modified.

Reviewer's comment:

- Page 7 statistical formulas could be avoided, will help keep the length reasonable.

Author's changes in the manuscript: Detailed mathematical formulas were transferred to the footnote under Table 3.

Reviewer's comment: - Page 8 Line 4: at selected wavelengths

Author's changes in the manuscript: Corrected.

Reviewer's comment: - Page 9 Line 5: ....and their ratio to Tchla

Author's changes in the manuscript: Corrected.

Reviewer's comment: - Page 14

**Figure 5 a,c: provide detailed legends for every spectra**

Author's changes in the manuscript: The new version of Figure 5 has been supplemented as suggested.

Reviewer's comment: - Page 15 Figure 6: (see the caption legend to in panel (b))

Author's changes in the manuscript: The caption to the new version of Figure 6 has been corrected as suggested.

Reviewer's comment: - Page 19 Line 1: Example of a Two component parameterization Line 2: (see also Figure 3)

Author's changes in the manuscript: Corrected.

Line 5: As a step towards improving....

Reviewer's comment: - Page 24 Line 5: check the percentages

Author's changes in the manuscript:

In the original manuscript, the ranges including data from all light wavelengths were given, while only selected wavelength were shown in tables. These particular fragment has been removed from the revised version of the manuscript.

Reviewer's comment: - Page 25 Line 2: now we shall.....with those of cite them here directly

Author's changes in the manuscript: Not applicable. The fragment has been modified.

Reviewer's comment: - Page 29 Line 3: Importantly, when matched.....

Author's changes in the manuscript: We have corrected the sentence fragment to avoid misunderstanding.

Reviewer's comment: - Page 30 Line 1: seawater optical components

Author's changes in the manuscript: The last five sentences of section 4 have been removed as suggested by the Reviewer # 2.

Figure R1. (a) and (b): Comparison of coefficients A and E of the one-component parameterization: for original variant matched to all data, and for the alternative variant matched to the data set limited to 2/3 of available data; (c) and (d) comparison of the main characteristics of the estimation error logarithmic statistics (mean logarithmic estimation error and standard error factor) calculated for original variants of one- and two-component parameterizations (all data used for training/all data used for error calculation) and for alternative variants validated against 1/3 of data.

---

## Author Comment (AC2)

**Responses to Reviewer #2**

Dear Reviewer,

Thank you very much for the extremely valuable critical comments on our work. Below we present our detailed point-by-point responses and the description of actions taken in regards to your comments. We believe that we have provided satisfactory explanations to your criticisms, and have made appropriate revisions in our paper.

Reviewer's comment:
*General comments: This paper is a continuation of a series of paper by the current authors on the parameterisation of the variability in phytoplankton absorption coefficients In the Baltic Sea. The analysis is extensive and thorough and is of some value for regional ocean colour remote sensing work.*

*I have the following **major concerns**:**1**) The manuscript falls short of drawing necessary conclusion from the results presented here. There is very limited discussion on natural sources of observed variability, how much of it is real and what ecosystem processes can be inferred from findings. For example: does the data suggest that an increased amount of photo-protective pigments influences the relationship in the summer months? And if not, why not?*

Author's response:
We would like to thank the Reviewer for noticing some of the positive aspects of our work. Regarding the major concern about the lack of necessary conclusions, we would like to point out that we tried to present only the most important observations and not to over-expand the work with other elements of a speculative nature. In our work we have documented that in the case of the Baltic Sea and the data collected during different periods of the year, one can see systematic differences when coefficients of classical one-component formulas for $a_{ph}$ (phytoplankton absorption coefficient) are matched to the data from various months. Because these differences are systematic, we believe that is not likely that they are caused only by possible measurement errors, because the latter should have rather a random character. Additionally, the differences observed between monthly specific parameterization variants qualitatively resemble the ones that occur between various parameterizations given by different authors for different marine environments. As we write in the final paragraph of our paper: '...*This particular observation reminds us that all such parameterizations are always quite far-reaching simplifications of relationships occurring in nature. The variability of these relationships that we recorded throughout the year in the Baltic Sea seems to indicate that only the use of a much more elaborate mathematical apparatus, using a much larger number of variables describing the composition of pigments and other features of the phytoplankton population* (i.e. 'packaging effect')*, could further and more radically improve the accuracy of the spectral description of the light absorption coefficient...'*. But, as we know for practical purposes there is often a need to look for simplified solutions. As we have documented, the simple parameterization of $a_{ph}$ as function of *Tchla* can be improved if the additional "degree of freedom" is added. We believe that it can be done either in a practical manner, by matching parameterizations to particular periods of the year, or alternatively by adding the additional variable, which would allow to get different estimated spectra of $a_{ph}$ coefficient while the concentration of chlorophyll *a* remains the same. From our analyses it has emerged, that if an addition of only one variable is concerned, the best candidate seems to be the ratio of $\sum C_i$ to *Tchla*. But obviously we have considered different possibilities; different ratios between

pigments groups (*PSC*, *PPC*, *Tchlb* and *Tchlc*) and *Tchla* were analysed. However, we understand the curiosity expressed by the Reviewer, and to satisfy it, we have prepared additional drawings documenting the scenario when as an additional variable the ratio of photoprotective carotenoids to chlorophyll *a* is considered (see Figure R4). These additional results can be compared with Figure A1 from the new version of the manuscript (or with Figure 8 in an original manuscript). Such comparison indicates that the influence of *PPC/Tchla* on the examined relationship is noticeable, but it is statistically less significant than the effect parameterized by changes of $\sum C_i/Tchla$.

Author's changes in the manuscript:
In order to respond to the Reviewer's concern, in addition to the arguments presented above, we have also expanded the scope of information given in the Introduction section regarding the influence of different pigments composition on the phytoplankton absorption spectra. In regards of the conclusions, we believe that the most important of them (not speculative in nature) were already given in the last paragraph of the article.

Reviewer's comment:
*2) I find myself unable to judge how much of the presented finding are actually novel and how much is a re-iteration of previous papers by the authors on the same dataset. Please clarify the progression from one paper to the next and highlight the additional value of this paper. Also justify why the results presented here were not published in previous paper.*

Author's response:
As a first part of our answer let us quote a new, modified version of the third paragraph of the Introduction section: *'...In one of earlier work by our research team (Meler et al. 2017a) we have already provided initial version of power function parameterisation adjusted to the data collected in the southern Baltic Sea. However, these preliminary analyzes were mainly focused on the joint data from marine and lacustrine environments, were based on a much smaller set of data than the one we currently have, and the initial parameterization was provided only for all the data pulled together regardless of the time of acquisition. On the other hand, in other paper published lately we have been able also to identify significant differences in the absorption properties of Baltic phytoplankton at different times of the year (see Meler et al. 2016b). In another preliminary study, limited to just a single light wavelength of 440 nm, we have also demonstrated differences between the coefficients of the relevant simplified parameterizations when they have been tailored to data gathered at specific times of the year (see Meler et al. 2017b). It is in this context, therefore, that we have decided in the present paper to re-address the problem of determining practical forms of a simplified parameterization of the phytoplankton absorption coefficient appropriate to Baltic Sea conditions....'*
When the first work cited above (Meler et al 2017a) was originally prepared, only much smaller dataset was available to its authors. Only having an extended set of data allowed us this time to carry out analyzes taking in which variations occurring through the year could be taken into account. Please also note that in our current work we have followed suggestions given by the scientific community. We have decided to use currently preferred expression for the "beta factor" for the T-R method (according to Stramski et al. (2015)), and *Tchla* concentration data from HPLC analyses (and not, as before, according to the spectrophotometric measurements). In a series of our works we try to document our findings, but obviously this must be done in a gradual way, due to the limitations in the length of individual scientific articles.

Author's changes in the manuscript:

The third paragraph of Introduction has been modified.

Reviewer's comment:
*3) The comparison of different parameterisations on this dataset does not add much value to the paper – especially if the performance is tested using the same dataset used for development.*

Author's response:
In the revised version of manuscript, we tried to clearly separate two issues. In the Section 3.3, we refer to estimation errors of our new formulas (obviously these errors are calculated at the same dataset as the one used for "training", and this is by no mean a validation of our formulas). And in the section 3.4 we presented the variability of coefficients between selected examples of parameterizations from literature and also the errors made when they are applied to our dataset.We believe that results presented in both of these sections, can be valuable to the interested reader, even if the 'classic' validation of our new formulas was not performed. All these results, in general, help to document the limitations of simple one- (or two-) component parameterizations.

Author's changes in the manuscript:
Sections 3.3 and 3.4 have been significantly modified

Reviewer's comment:
*4) The final recommendation as to which parameterization should be used (and under which conditions) is not clear. Quantify the benefit of the new parameterization and weigh it against processing time/costs.*

Author's response:
On the basis of our analysis of estimation errors, it can be expected that in the case of data sets from different periods of the year, on average only small differences between the various versions of parameterization will be noticed. However, when the information on the $\sum C_i/Tchla$ is available, it is, in our opinion, advisable to try using the two-component version, in order to get a slightly better accuracy of estimating of $a_{ph}$ in the most-important ranges around 440 and 675 nm absorption peaks. In the case of analyzing the data from selected months only, the differences between parameterizations may be much more pronounced. In practice the properly selected monthly parameterization variants may be the best solution. The qualitative benefit of two-component parameterization and "dynamically selected" monthly parameterization is the fact that they allowed for different absorption magnitudes and shapes for the same chlorophyll *a* concentrations. In our opinion the differences between processing time is not an important issue in resolving "forward" problems (finding $a_{ph}$, when the *Tchla* or *Tchla* and $\sum C_i$ are known). Obviously differences will arise when trying to solve "inverse" problems (estimating *Tchla* or *Tchla* and $\sum C$, when the absorption spectrum is known).

Author's changes in the manuscript:
We have modified section 3.3 and clarified a recommendation that has been given in the final section of manuscript.

Reviewer's comment:
*5) Additionally, the language requires major revision (preferably from a native speaker) which should aim to shorten the article significantly and improve readability & understandability.*

Author's response:

We have rearranged some sections of the text to improve readability and understandability. The revised manuscript has been also corrected by a native speaker.

Author's changes in the manuscript:
See various changes marked in colour throughout the revised version of manuscript.

**Specific comments**:

Reviewer's comment:
*1) Explain the selection criteria for literature chosen for comparison. Can you use a more systematic approach to select a small (max. 5!) number of parameterizations from the hundreds available in the literature (e.g. only choose chase 2 waters or compare 2 case1 and 2 case 2 water studies)? McKee et al. (2014), for example, did some analysis on the NOMAD data set which includes a variety of different water and geographic locations.*

Author's response:
In the first version of the manuscript, literature items were selected for comparisons, which concerned both case 1 waters and case 2 waters from different marine and lacustrine environments. Indication of the results achieved in lake waters was aimed at referring to an earlier work by Meler et al. (2017a), in which one-component parameterization was presented for the combined data set acquired in the southern part of the Baltic Sea and in lakes in the Polish coastal zone, pointing to their similar optical properties.
In the current version of the manuscript, as suggested by both Reviewers, the selection of literature items for comparison has been modified and limited. In addition to the classic work by Bricaud at al. (1995), we are now referring mainly to five other examples documenting the diversity of results obtained in different marine environments (Stramska et al. (2003), Staehr and Markager (2004), Matsuoka et al. (2007), Nima et al. (2016), Churilova et al. (2017))

Author's changes in the manuscript:
Appropriate changes were made in the text and in Figures 4, 6 and 9.

Reviewer's comment:
*2) Revise terminology throughout article: - Use 'case 2 waters' rather than 'case 2' – Use 'at short wavelengths/in blue spectral region' instead of 'short-wave part of spectrum' -In my opinion, using 'absorption' rather than 'light absorption' is sufficient and would improve readability.*

Author's response:
Corrected in accordance with the reviewer's suggestion.

Reviewer's comment:
*3) p. 4. Line 9: 'Because of 10 weather- and sea-state-related limitations, the proportion of data collected in open water regions exceeds ca 30% only for data collected in February, March, May, September and October (see Table 1).' – This is an example where the text can be shortened significantly: 'The amount of data collected in open waters was limited due to adverse weather (< 30% for the majority of months).'*

Author's response:
Corrected.

Reviewer's comment:
*4) Methods, Table 1: Were any spectra excluded from the data – if yes, what were the selection/data quality control criteria?*

Author's response:
The data set was first subjected to a preliminary qualitative control - data for which a set of associated data was missing was removed, non-physical data was removed, e.g. negative absorbance values. Secondly, after the parameterization, estimation errors were calculated and the so-called outliers - values clearly differing from the typical (laying outside the limits of +/- 3 SD) were removed.

Reviewer's comment:
*5) p.5, paragraph 1: Justify why you did not correct for scattering offsets by forcing the spectra through zero, in red/NIR? Are you aware of the systematic errors your choice potentially introduces to the data. Lefering et al. 2016 for detailed analysis.*

Author's response:
In general, we did not use correction due to the scattering effect, because in the spectrophotometric measurements we use the so-called T-R (transmission-reflectance) method according to Tassan and Ferrari (1999, 2002). It is generally accepted that such a correction is not needed when using this particular method. Our final data was corrected, however, by forcing final values in the range of 740-750 nm to be close to zero. The methodical section has been improved and supplemented in this matter.

Reviewer's comment:
*6) p. 6, l. 19: How did you assess reproducibility of samples? Did you measure any replicates at all, if yes how?*

Author's response:
In the collected data set, due to logistic limitations, generally no measurements were made on multiple samples. However, the separate tests has revealed that the average uncertainty of $a_{ph}$ measurements due to subsampling was 9.1% (± 1.5%, SD). The methodological section has been supplemented with this information.

Reviewer's comment:
*7) p. 6, ll.22-24: Moving averages can lead to a reduction of chl a absorption in the red. Did you observe this effect? Consider lo-ess smoothing.*

Author's response:
The procedure of smoothing absorption spectra carried out by us does not significantly reduce the value of light absorption coefficients in any band. This information was added to the text of the work.

Reviewer's comment:
*8) p. 7, l. 11: Highlight that you calculate the RMSE/standard deviation on relative errors. I would be interested to see the range absolute errors as well.*

Author's response:
The data set analyzed in the work covers a wide range of variability, i.e. almost three orders of magnitude. We believe that in this situation, the statistics of relative error and, above all, the

statistics on logarithmic values should be used for analyzes (see the last paragraph of Methods Section). However, to satisfy the Reviewer's curiosity, we present the version of Table 3 supplemented with the arithmetic statistics of absolute error (see Table R1).

Reviewer's comment:
*9) How is the standard error factor interpreted? What is the ideal, what a reasonable value?*

Author's response:
The standard error factor allows to calculate the range of statistical errors according to logarithmic statistics. As in standard arithmetic statistics we can calculate the error range by adding or subtracting the value of standard deviation (SD), in logarithmic statistics similar range can by calculated by multiplying/dividing by the standard error factor (x)). The ideal value is the case when x = 1, because it would mean that all approximate values accurately reflect empirical values. When the target physical quantities change for about three orders of magnitude, we believe that standard error factors of the order of 1.5 and less should be treated as reasonable (x=1.5 means the main range of variability is between 67% and 150% of original value, and the statistical error range is between -33% and +50%).

Reviewer's comment:
*10) p.7, l.28: I understand that low measurement sensitivity can cause issues at blue/UV wavelengths. How do you explain the observed artefacts at 550 – 650 nm.*

Author's response:
This is the range where absorption takes small values compared to other bands, they are close to noise, and the smoothing procedure is not fully effective there. In this spectral area, absorption by phycobilins is also possible.

Reviewer's comment:
*11) p.7, ll. 26-29: Use neutral and objective language (here and throughout document)!For example, avoid expressions like 'undesirable artefacts'.*

Author's response:
Corrected across the manuscript in accordance with the reviewer's suggestion.

Reviewer's comment:
*12) Overall the absorption data appears to be of low quality. I know that it's not possible to repeat the measurements with an improved protocol but a thorough and more detailed analysis of data quality is required. Do you have any information with regards to bleach contamination resulting in low absorption values < 440 nm?*

Author's response:
We are aware of the fact that the data obtained by spectrophotometric method can be generally characterized with inaccuracies, associated with many factors. We tried different techniques to avoid such inaccuracies. The use of bleaching agent to obtain spectra of light absorption coefficients by non-algal particles has been 'neutralized' by rinsing bleached filters with clean particle-free seawater. It is possible, however, that in some cases a slight amount of the suspension collected on the filter could also be rinsed during the flushing, resulting in the $a_{NAP}$ measurement being underestimated. Practically, we tried to correct for these possible undesirable effects by correcting to zero the final $a_{ph}$ values in the 740-750 nm range. In this matter, we have completed the methodical section of the manuscript. The effectiveness of such

procedure was checked on blank filters. The average relative difference were not higher than 2.8%.

Reviewer's comment:
*13) The absorption is expected to vary several orders of magnitude across all wavelengths and samples. This is not a new result. Interesting would be what the implications for your error analysis are? Do you see larger absolute errors for low absorption or do measurement errors increase linearly with strength of absorption?*

Author's response:
Please note that in our measurements we used filter pad technique and different volumes of water were filtered for different samples (see the supplemented methods section). In this sense, the signal from weakly absorbing samples was always 'amplified'. The proportions between absorption signal at the blue peak to the instrument noise, being a main source of uncertainty, was kept often at similar level, regardless of the fact the absolute values of absorption in original water samples changed over almost three orders of magnitude. For that reason the relative uncertainty due to instrument noise is at the similar level, regardless of magnitude of absorption of phytoplankton in seawater.

Reviewer's comment:
*14) p.9, l. 14: Use neutral and objective language (here and throughout document)! Another example: 'fairly well'.*

Author's response:
Corrected across the manuscript in accordance with the reviewer's suggestion.

Reviewer's comment:
*15) p. 11, ll. 4-5: Delete sentence, unnecessary self-citation at this point: 'We ourselves had already reported similar ranges of variability earlier (see Wozniak et al. 2011,Meler e t al. 2016b, 5 2017a and b).'*

Author's response:
Deleted.

Reviewer's comment:
*16) Section 3.2.1: What happens if E(lambda) > 1?*

Author's response:
E> 1 means the statistically observed increase in $a^*_{ph}$ with increasing chlorophyll. It has been explained in the revised manuscript text.

Reviewer's comment:
*17) Section 3.2.1: How can the annual variability in A be explained? Can it be linked to extent of packaging effect or the amount of photo-protective vs. photosynthetic pigments?*

Author's response:
The general relationships between absorption and the size of phytoplankton cells, intracellular pigments concentration and pigments composition are mentioned in the Introduction section (which was also supplemented). The data collected by us does not contain direct information about particle sizes. However, the analyzes carried out show that the influence of the proportion

of accessory pigments to *Tchla* is significant, and may be used to 'construct' the two-component parameterization. An example of similar analyzes concerning photoprotective pigments as separate group is presented in the additional Figure R4. I the case of our data, however, it turned out that a better candidate for an additional variable is $\Sigma C_i/Tchla$ and not *PPC/Tchla*.

Reviewer's comment:
*18) Section 3.2.1: What can be inferred from your analysis of R2? Does is correlate with any known ecosystem dynamics? Considerably shorting the paragraph unless you justify its value.*

Author's response:
$R^2$ is only a parameter initially describing the quality of parameterization when matched to the measured data.

Reviewer's comment:
*19) P.12 l. 8: Do you mean '690 – 700 nm'?*

Author's response:
In the new version of the manuscript, this fragment has been modified.

Reviewer's comment:
*20) Section 3.2.1: What do you derive from the observed differences in performance compared to Bricaud's parameterisation? Why do you observe less variability and what does that mean for the ecosystem?*

Author's response:
The following sentence has been added to the text: '...These differences indicate that the combined influence of the packaging effect and the decrease in relative accessory pigment concentrations manifests itself differently in our Baltic Sea dataset than in the original oceanic dataset of Bricaud et al. (1995)....'

Reviewer's comment:
*21) Please explain the metric of the colour index in more detail. What the reasons for the observed flattening (drop in colour index) of the absorption spectrum? Are you expecting a certain magnitude in your colour index drop and why?*

Author's response:
In the Introduction section, the sentence has been modified / added: '...*Another important factor, where absorption by phytoplankton is concerned, is how densely the strongly light-absorbing pigments are 'packed' within the internal structures of individual cells (the so-called 'packaging effect', see e.g. Morel and Bricaud 1981, 1986). Increasing intracellular pigment concentrations and cell size flatten the real absorption spectra compared to what one may expect from a simple addition of the absorption coefficients of individual pigments....*'
In section 3.2.1, the sentence is added: '...*As a simplified measure of spectra 'flattening', one can analyse, for example, the changes in the ratio of $a_{ph}(440)$ to $a_{ph}(675)$: this ratio is sometimes referred to as the 'colour' or 'pigment' index (see e.g. Woźniak and Ostrowska 1990 a and b; see also Bricaud et al. 1995)....*'

Reviewer's comment:
*22) p. 18: Shorten paragraph. Only highlight key finding shown in Table and consider deleting/rephrasing the following sentences: 'We have carried out such an analysis, but we do*

*not present its detailed results due to the fact that they are very extensive. Here we limit ourselves only to stating that the use of general or monthly parameterizations for individual months 10 has little effect on the level of statistical error according to logarithmic statistics, although it may have, as generally expected, a very significant impact on the level of systematic error.'*

Author's response:
The paragraph has been reformulated. Tables 3 and 4 are now discussed in one paragraph entitled: 3.3 Estimation errors of the different variants of parameterizations.

Reviewer's comment:
*23) Section 3.2.2 The use and exchange of a_ph,cal/a_ph,m and a_ph in this section is confusing. What information can be gained from a_ph,cal/a_ph,m and how can the ratio be related to a_ph in this context?*

Author's response:
These are only mathematical details - at the request of Reviewer #1 they are now moved to the new Appendix 2; ratio of $a_{ph}$ calculated to $a_{ph}$ measured was used to determine the functional form of the "patch" for equation and 3.a and allowed to find the final form of equation 4.

Reviewer's comment:
*24) Fig. 8 (e) & (f): Justify the use of an exponential over a linear fit.*

Author's response:
The analyzes were carried out on logarithmic values, therefore the fits we found in the logarithmic space are linear fits.

Reviewer's comment:
*25) p. 27, ll. 1-14: There is limited value in the comparison with other parameterisation using the same dataset (see comment above).*

Author's response:
Comparison with selected examples of parameterizations from the literature was only done for illustrative purposes. In no way was it an attempt to validate our results (obviously, for such a purpose we would need to acquire a separate dataset, not used in parameterization 'training')). In new Figure 9 (simplified version of original Figure 12) we only wanted to document that using different literature parameterization on our dataset is often burdened with a large either positive or negative systematic errors and with statistical errors often higher, then errors which we were able to achieve with our new parameterizations matched to our data. (please, see also our answer to comment number 2 of Reviewer#1 and an additional Figure R1).

Reviewer's comment:
*26) p. 27, ll. 21 – end: The conclusion of superior performance based on the standard error factor cannot be justified. 'If we take into account both systematic and statistical errors, the superiority of the new parameterizations developed specifically for Baltic Sea conditions is undisputed.' Poor scientific language!*

Author's response:
Changed; the final sentence of the last paragraph of section 3.4 is now as follows: '*(...) But since for the total estimation accuracy the contributions of both systematic and statistical errors have*

*to be taken into account, one can expect that overall, none of the literature examples can attain the accuracy that we achieved by matching our new parameterizations to our own dataset....'*

Reviewer's comment:
*27) p. 29, 32: Delete sentences: 'Practical remote sensing algorithms are a good example in this respect. Their task is often to solve complicated "reverse" problems. Starting with measurements of sea colour, such algorithms, through intermediate steps during which different inherent optical properties of seawater (including light absorption)are retrieved, should yield basic features of different seawater components in the final stage of their operation. One such feature may be the concentration of chlorophyll a, which is often treated as a practical measure of the biomass of live phytoplankton contained in water. In our opinion the parameterizations presented in this work could be used in practice for such purposes.'*

Author's response:
That sentence has been deleted.

[Figure]

**Figure R1.** (a) and (b): Comparison of coefficients *A* and *E* of the one-component parameterization: for original variant matched to all data, and for the alternative variant matched to the data set limited to 2/3 of available data; (c) and (d) comparison of the main characteristics of the estimation error logarithmic statistics (mean logarithmic estimation error and standard error factor) calculated for original variants of one- and two-component parameterizations (all data used for training/all data used for error calculation) and for alternative variants validated against 1/3 of data.

[Figure]

**Figure R2.** (*extended variant of Figure 3 from the revised manuscript*). (a): Box plot presenting the range of variation of chlorophyll *a* concentration (*Tchla*) for all the data analysed, for each sampling month, and for three selected ranges of *Tchla*; (b): as (a) but showing the sum of accessory pigments to the chlorophyll *a* concentration ($\Sigma C_i$); (c): as (a) but showing the ratio of the sum of accessory pigments to the chlorophyll *a* concentration ($\Sigma C_i/Tchla$); (d): graph illustrating the relationship between $\Sigma C_i$ and *Tchla* – solid lines represent simple functional approximations of the relationship (the equations are given below the panel).

[Figure]

$Tchla$ [mg m$^{-3}$]:

0.31-2.57: $a_{ph}^{*}(440) = 0.056*Tchla^{-0.178}$

2.57-5.38: $a_{ph}^{*}(440) = 0.0499*Tchla^{-0.112}$

5.38-141.8: $a_{ph}^{*}(440) = 0.048*Tchla^{-0.139}$

$a_{ph}^{*}(675) = 0.022*Tchla^{-0.085}$

$a_{ph}^{*}(675) = 0.0224*Tchla^{-0.115}$

$a_{ph}^{*}(675) = 0.0194*Tchla^{-0.068}$

**Figure R3.** (*alternative to Figure 6a and b from the revised manuscript*) (a): Relationship between coefficient $a_{ph}^{*}(440)$ and the chlorophyll *a* concentration *Tchla* and its functional approximations determined in this study for all the data analysed, for selected sampling months, and for limited ranges of *Tchla* (see the legend to panel b; equations are given below the panel); (b): as (a) but for $a_{ph}^{*}(675)$. The grey dots on each panel represent individual data points from our database.

[Figure]

**Figure R4.** (a): Box plot presenting the range of variation of the ratio of the photoprotective carotenoids to chlorophyll *a* concentration (*PPC/Tchla*) for all the data analysed, and for each sampling month; (b) and (c) relations between the ratio $a_{ph}(\lambda)_{cal}/a_{ph}(\lambda)_m$ and the pigment concentration ratio *PPC/Tchla*, and their functional approximations (the equations are given in the panels).

**Table R1.** Arithmetic statistics of estimation absolute errors* of coefficient $a_{ph}(\lambda)$ in selected spectral bands when the different variants of the parameterization derived in this study were applied to the entire dataset (n= 1002). The calculated values are given for three scenarios: when the general variant of the one-component parameterization was used; when variants specific to individual months were chosen (the first alternative value is given in parentheses); and when the two-component parameterization was used (the second alternative value is given in parentheses).

| $\lambda$ [nm] | absolute systematic error $\langle \varepsilon_a \rangle [m^{-1}]$ | absolute statistical error $\sigma_{\varepsilon_a} [m^{-1}]$ |
|---|---|---|
| 350 | -0.044 (-0.037; -0.044) | 0.22 (0.21; 0.22) |
| 400 | -0.025 (-0.023; -0.027) | 0.17 (0.16; 0.18) |
| 440 | -0.019 (-0.017; -0.022) | 0.15 (0.14; 0.15) |
| 500 | -0.009 (-0.008; -0.010) | 0.09 (0.08; 0.07) |
| 550 | -0.005 (-0.005; -0.005) | 0.05 (0 04; 0.05) |
| 600 | -0.005 (-0.005; -0.005) | 0.03 (0.03; 0.03) |
| 675 | -0.007 (-0.007; -0.009) | 0.06 (0.06; 0.06) |
| 690 | -0.005 (-0.004; -0.005) | 0.04 (0.04; 0.04) |
| 700 | -0.003 (-0.003; -0.003) | 0.02 (0.02; 0.02) |

**\*) Arithmetic statistics of the absolute error:**
- mean of the absolute error (representing the systematic error according to arithmetic statistics):
$\langle \varepsilon' \rangle = N^{-1} \sum_{i=1}^{N} \varepsilon'_i$ , where $\varepsilon' = P_i - O_i$, $O_i$ - observed/measured values, $P_i$ - predicted/estimated values
- the standard deviation of the absolute error (representing the statistical error according to arithmetic statistics):
$\sigma_{\varepsilon'} = \sqrt{\frac{1}{N} \left( \sum_{i=1}^{N} (\varepsilon'_i - \langle \varepsilon' \rangle)^2 \right)}$

---

## Author Comment (AC3)

**Parameterization of phytoplankton spectral absorption coefficients in the Baltic Sea: general, monthly and two-component variants of approximation formulas**

Justyna Meler[1], Sławomir B. Woźniak[1], Joanna Stoń-Egiert[1], Bogdan Woźniak[1†]

[1]Institute of Oceanology, Polish Academy of Sciences, Powstańców Warszawy 55, 81-712 Sopot, Poland

*Correspondence to*: Justyna Meler (jmeler@iopan.pl)

**Abstract.** The paper presents approximate formulas (empirical equations) for parameterizing the coefficient of light absorption by phytoplankton $a_{ph}(\lambda)$ in Baltic Sea surface waters. Over a thousand absorption spectra (in the 350-750 nm range), recorded during nine years of research carried out in different months of the year and in various regions of the southern and central Baltic, were used to derive these parameterizations. The empirical material was characterized by a wide range of variability: the total chlorophyll *a* concentration (*Tchla*) varied between 0.31 and 142 mg m$^{-3}$, the ratio of the sum of all accessory pigment concentrations to chlorophyll *a* ($\sum C_i$/*Tchla*) ranged between 0.21 and 1.5, and the absorption coefficients $a_{ph}(\lambda)$ at individual light wavelengths varied over almost three orders of magnitude. Different versions of the parameterization formulas were derived on the basis of these data: a one-component parameterization in the "classic" form of a power function with *Tchla* as the only variable, and a two-component formula – the product of the power and exponential functions – with *Tchla* and $\sum C_i$/*Tchla* as variables. We found distinct differences between the general version of the one-component parameterization and its variants derived for individual months of the year. In contrast to the general variant of parameterization, the new two-component variant takes account of the variability of pigment composition occurring throughout the year in Baltic phytoplankton populations.

**1 Introduction**

If we wish to fully describe the process of photosynthesis in the seas and oceans, and to correctly interpret remote observations of water bodies, it is important to obtain an accurate quantitative description of the spectral characteristics of light absorption by living phytoplankton, a significant constituent of seawater (see e.g. Kirk 1994, Mobley 1994, or Woźniak and Dera 2007). The efficiency of sunlight absorption by this phytoplankton generally depends on a number of factors. The major, strongly absorbing components of phytoplankton cells are the pigments they contain. The principal photosynthetic pigment is chlorophyll *a*, which absorbs visible light mainly in two bands, one situated in the blue region, with a maximum around 440 nm, and another one in the red, with a maximum around 675 nm (see e.g. Bidigare et al. 1990, Bricaud et al. 2004 or Woźniak and Dera 2007). There are also various accessory pigments, like chlorophylls *b* and *c*, carotenoids and phycobilins. These latter

pigments have different spectral absorption characteristics and may be involved in photosynthetic, photoprotective or other processes in marine organisms. The pigment composition of photosynthesizing marine species can differ, since it can in a general sense reflect the adaptation of an organism to different light conditions (photo- and chromatic adaptation) as well as acclimation at the plant community level (see e.g. Kirk 1994, Woźniak and Dera 2007). Overall, analysis of pigment composition data obtained for marine phytoplankton assemblages from various ocean and sea waters has revealed a general trend indicating that the proportion of accessory pigments relative to chlorophyll *a* decreases with increasing chlorophyll *a* concentration (increasing trophicity) (see e.g. Woźniak and Ostrowska 1990, Trees et al. 2000 or Babin et al. 2003). Even so, there is substantial variability around this trend. Hence, the main factor that light absorption by phytoplankton assemblages depends on is the pigment composition. Another important factor, where absorption by phytoplankton is concerned, is how densely the strongly light-absorbing pigments are 'packed' within the internal structures of individual cells (the so-called 'packaging effect', see e.g. Morel and Bricaud 1981, 1986). Increasing intracellular pigment concentrations and cell size flatten the real absorption spectra compared to what one may expect from a simple addition of the absorption coefficients of individual pigments. Given the complexity of all these relationships, it is often necessary (or even required) for practical purposes to take a highly simplified approach, for example, when constructing models of bio-optical processes for interpreting remote sea observations. It is a common simplification to assume that all the relevant properties of a phytoplankton population can be roughly parameterized with the aid of just one variable – the concentration of chlorophyll *a*: indeed, the total biomass of an entire phytoplankton population as well as its diverse optical properties are often parameterized in this way. Earlier authors addressing this question applied this kind of simplification in attempts to determine typical values of the 'chlorophyll *a*-specific' absorption coefficient (defined as the light absorption coefficient of phytoplankton normalized to the chlorophyll *a* concentration). In practice, therefore, the adoption of one averaged value of this coefficient should enable the relationship between the phytoplankton absorption coefficient and the chlorophyll *a* concentration in seawater to be described using the simplest possible, i.e. linear, functional relationship. As measured in nature, however, values of the specific absorption coefficient have proved to be highly variable. The papers by Bricaud et al. (1995, 1998), often cited by other authors, were among the first to introduce for practical purposes a different, non-linear, approximate description of the light absorption vs chlorophyll *a* concentration relationship. They proposed using a power function to account for the general decrease in light absorption efficiency per unit chlorophyll *a* concentration that occurs with increasing absolute values of this concentration in seawater (the relevant mathematical formulas are given later in the text). Bricaud et al. (1995) also gave a theoretical explanation of these effects, suggesting that there might be a correlation between the increase in the absolute chlorophyll *a* concentration and the increasing contribution of the pigment packaging effect and, concurrently, the decreasing proportion of pigments other than chlorophyll *a*. The papers by Bricaud et al. (1995, 1998) were based on extensive empirical material gathered in different regions of open, oceanic waters, classified as 'case 1 waters'. Different authors addressed the same problem in many later papers: examples of spectral power function parameterizations for different marine environments can be found in, for example, Stramska et al. (2003), Staehr and Markager (2004), Matsuoka et al. (2007), Dmitriev et al. (2009), Nima et al. (2016), Churilova et al. (2017) and Mascarenhas et al. (2018). The subject literature also provides examples of similar

parameterizations derived for inland water bodies, for example, by Reinart et al. (2004), Ficek et al. (2012a, b), Ylöstalo et al. (2014) and Paavel et al. (2016). All of these papers give spectral coefficients of parameterizations tailored to specific datasets differing from each other to a greater or lesser extent. Obviously, all such parameterizations are far-reaching simplifications of the complex dependences observed in nature. That there might be significant deviations from the approximate "average"

5 relationship was already made clear by Bricaud et al. (1995) in their original work; these authors subsequently analysed the potential causes of this differentiation (Bricaud et al. 2004). Using indirectly reconstructed information regarding the size structure of the ocean phytoplankton population, they were able to estimate separately the impacts of the differences in the dominant sizes of plankton populations and the differences in pigment composition on the relationship in question. In general, they found that for oligo- and mesotrophic waters (i.e. waters with chlorophyll $a$ concentrations < 2 mg m$^{-3}$), the variability

10 associated with the packaging effect might be exerting a more significant influence, whereas in eutrophic waters (with higher chlorophyll $a$ concentrations) both effects might be of equal weight. Generally, however, the observed variability indicates that one should expect both regionally and seasonally differentiated forms of such simplified relationships to occur instead of one universal, approximate statistical relationship between $a_{ph}(\lambda)$ and the chlorophyll $a$ concentration.

The Baltic Sea, the region we have been studying, is a semi-enclosed, brackish water basin classified as an example

15 of 'case 2 waters'. It is characterized by usually high concentrations of terrigenous dissolved organic substances (see e.g. Kowalczuk 1999). The biomass and species composition of living phytoplankton in the Baltic is known to vary during the year. Usually there are three main phytoplankton blooms: a spring bloom of cryophilous diatoms, which transforms into a bloom of dinoflagellates (early March – May), a summer bloom of cyanobacteria (July and August), and an autumn bloom of thermophilous diatoms (September – October) (see Wasmund et al. 1996 and 2001, Witek and Pliński 1998, Wasmund and

20 Uhlig, 2003, Thamm et al. 2004).

In an earlier paper by our research team (Meler et al. 2017a) we provided an initial version of the power function parameterization, adjusted to data collected in the southern Baltic Sea. However, these preliminary analyses, focusing mainly on pooled data from marine and lacustrine environments, were based on a much smaller set of data than the one we currently have, and the initial parameterization was carried out for all the data pooled, regardless of when they were acquired. On the

25 other hand, another recent paper of ours identified significant differences in the absorption properties of Baltic phytoplankton at different times of the year (Meler et al. 2016b). In a further preliminary study, limited to just the single light wavelength of 440 nm, we demonstrated differences between the coefficients of the relevant simplified parameterizations when they were tailored to data gathered at specific times of the year (Meler et al. 2017b). It is in this context, therefore, that we have decided in the present paper to re-address the problem of determining practical forms of a simplified parameterization of the

30 phytoplankton absorption coefficient appropriate to Baltic Sea conditions.

The objectives of this work are twofold:

- the main one is to find new forms of the classic, one-component power function parameterization for the phytoplankton absorption coefficient adapted to the specific conditions of the Baltic Sea. An important aspect of this is to record the extent of the differences between the coefficients of spectral parameterization when they are derived separately for data from selected

periods of the year. These new analyses, as opposed to the preliminary results published earlier, have to be performed over a wide spectral range, with a sufficiently high resolution, on the currently available extended dataset and also in accordance with the latest recommended calculation procedures;

- an additional aim of this work is to propose a modified, but still relatively simple, new form of parameterization enabling the

5    diversity of phytoplankton absorption properties observed in the study area during the year to be taken into account.

The new forms of parameterization that we are seeking can be used, among other things, to develop and improve the accuracy of practical, local algorithms for interpreting remote observations of the Baltic Sea.

**2 Materials and methods**

The empirical data used in this study were collected at more than 170 measuring stations in various parts of the

10    southern and central Baltic Sea, though mainly in the Polish economic zone, from 2006 to 2014 (see Figure 1). These data were acquired principally during 42 short research cruises on board r/v Oceania at different times of the year, but mostly from March to May and from September to October (about 80% of the data analysed in this paper are from these periods). The practice during each cruise was to select measuring stations that were maximally diverse with respect to their optical properties, i.e. in the vicinity of river mouths and estuaries (the Rivers Vistula, Oder, Reda, Łeba and Świna; the Szczecin Lagoon), bays

15    and offshore waters (Gulf of Gdańsk, Puck Bay and Pomeranian Bay), and open southern Baltic waters. During three cruises (in May of 2010, 2012 and 2014), measurements were also made in the open waters of the central Baltic. However, because of weather- and sea-state-related limitations, only 32% of the data are from open water regions (see Table 1). In addition to the cruise measurements, data were gathered throughout the year by sampling the seawater at the end of the ca 400 m long pier in Sopot, on the Gulf of Gdansk coast (< 7% of the overall number of data analysed).

[Figure]

**Figure 1.** Locations of all the sampling stations in the Baltic Sea; enlarged view of the study area.

**Table 1.** Numbers of seawater samples analysed, divided into the months and areas of their acquisition.

| Time of the year | Open Baltic Sea areas | Coastal areas and bays | All regions |
|---|---|---|---|
| January | 4 | 6 | 10 |
| February | 16 | 36 | 52 |
| March | 52 | 122 | 174 |
| April | 22 | 74 | 96 |
| May | 80 | 121 | 201 |
| June | - | 5 | 5 |
| July | - | 7 | 7 |
| August | 4 | 21 | 25 |
| September | 75 | 136 | 211 |
| October | 51 | 101 | 152 |
| November | 15 | 42 | 57 |
| December | 1 | 11 | 12 |
| All year | 320 | 682 | 1002 |

During the research cruises, a diversity of physical and optical parameters of seawater were measured *in situ* at each sampling station, and discrete seawater samples were collected for further laboratory analysis of certain optical properties (spectra of coefficients of light absorption by phytoplankton) and biogeochemical properties (concentrations of chlorophyll *a* and other phytoplankton pigments). These samples were collected with a Niskin bottle (height ca 0.9 m, capacity 25L) immersed just below the surface; in shallow estuarine areas and river mouths (sampled from a pontoon) or off the end of the Sopot pier, they were obtained with a bucket. Immediately after collection, all samples were passed through glass fibre filters (Whatman, GF/F, 25 mm, nominal retention of particles with sizes down to 0.7 μm) at a pressure not exceeding 0.4 atm. The volumes filtered were chosen on a case-by-case basis; between 2 and 1150 ml of seawater were filtered for later absorption measurements, and generally between 150 and 1000 ml for phytoplankton pigment concentration analysis). All sample filters were immersed in a Dewar flask containing liquid nitrogen (at about -196°C) and then kept frozen (at about -80°C) for further analysis in the laboratory on land.

In order to determine the spectra of the phytoplankton absorption coefficient $a_{ph}$, we measured the absorption coefficient spectra for all suspended particles retained on filters ($a_p$), and also, after chemical bleaching of the pigments in our samples, the corresponding spectra of non-algal particles ($a_{NAP}$). We performed the optical measurements in the 350-750 nm spectral range with a UNICAM UV4-100 double-beam spectrophotometer equipped with an integrating sphere of external diameter 66 mm (LABSPHERE RSA-UC-40). For the reference measurements we used clean filters rinsed with particle-free seawater. The methodology of combined light-transmission and light-reflection measurements (known as the T-R method) was that described by Tassan and Ferrari (1995, 2002). From the pooled results of these measurements (several scans in

different configurations), we calculated the optical density $OD_s(\lambda)$ representing each filtered sample. As opposed to standard spectrophotometric analyses performed in transmission mode only, at least in theory, the results of T-R method should not need to be corrected for the so-called 'scattering error'. But in order to calculate absorption coefficients of particles in solution, an additional correction has to be made to compensate for the elongation of the optical path of the light owing to the multiple scattering occurring in the filtered material. This is done by applying the dimensionless path length amplification, the β-factor, which converts the measured optical density of particles collected on the filter ($OD_s(\lambda)$) into the optical density characterizing these particles in solution ($OD_{sus}(\lambda)$) (Mitchell 1990). In our analyses we used the new β-factor formula proposed by Stramski et al. (2015) for the T-R method:

$$OD_{sus}(\lambda) = 0.719 OD_s(\lambda)^{1.2287}. \tag{1}$$

The coefficient of light absorption by all suspended particles was then calculated using the formula:

$$a_p(\lambda) = [\ln(10) \cdot OD_{sus}(\lambda)]/l, \tag{2}$$

where $l$ [m] is the hypothetical optical path in solution, determined as the ratio of the volume of filtered water to the effective area of the filter. The absorption by non-algal particles $a_{NAP}(\lambda)$ was determined in an analogous way, after the phytoplankton pigments had been bleached for 2-3 minutes with a 2% solution of calcium hypochlorite $Ca(ClO)_2$ (Koblentz-Mishke et al. 1995, Woźniak et al. 1999); thereafter the sample filter was rinsed with a small volume of particle-free seawater to remove any bleach residue, as this could additionally absorb light at short wavelengths. Finally the sought-after coefficient $a_{ph}$ was calculated as the difference between $a_p$ and $a_{NAP}$.

In practice, however, we had to add two corrective procedures to the protocols described above. One related to the noise which appeared in the individual spectra recorded with our spectrophotometer. To partially eliminate it, we applied a 'spectral smoothing' procedure – the spectral 5-point 'moving average' was repeated 3 times – to the calculated individual spectra of both coefficients $a_p$ and $a_{NAP}$. This procedure partially eliminated fluctuations of the signal between adjacent light wavelengths but did not significantly affect the magnitude of the major absorption peaks, the 'half-widths' of which are of the order of tens of nm. The other corrective procedure related to the small deviations from zero of the calculated coefficients $a_{ph}$ at wavelengths close to 750 nm. It is generally assumed that the light absorption of phytoplankton pigments in this spectral range should be negligible (Babin and Stramski, 2002). The occurrence of non-zero values in this range may be due to several different factors: differences in the optical properties of individual glass fiber filters, the difficulty of maintaining ideally repeatable filter moisture during measurements, and the possible partial loss of sample material as a result of the filters being rinsed after bleaching. To correct for all these effects we applied a simple 'null-point' correction (see e.g. Mitchell et al. 2002). The average coefficient $a_{ph}$ calculated in the range between 740 and 750 nm was subtracted from the values of $a_{ph}$ across the entire spectrum. Generally different factors may have influenced the final uncertainty of our absorption measurements. Among them is the instrument noise present in each individual spectrophotometric scan (in each configuration), and also the possible uncertainty in path length amplification factor, volume filtered and filter area used in subsequent calculations, as well as

uncertainty coming from subsampling from larger volumes of water. As a strict estimation of all these inaccuracies would be very complicated (due to the mathematical complexity of the algorithm used according to Tassan and Ferrari (1995, 2002)), here we limit ourselves only to estimating some of these inaccuracies. The mean inaccuracy caused by instrument noise occurring in separate measurements of light absorption by particles before and after bleaching we estimated to be $\Delta a_{ph} =$ 5.68*10$^{-3}$ m$^{-1}$ (± 9.13*10$^{-3}$ m$^{-1}$, standard deviation (SD)). We assumed that the uncertainty of $a_{ph}$ ($\Delta a_{ph}$) can be calculated as a square root of a sum of squares of uncertainties $\Delta a_p$ and $\Delta a_{NAP}$. The latter were estimated as 95% prediction intervals (=1.96 SD) for $a_p$ and $a_{NAP}$ values between 740 and 750 nm, where it is assumed that the absorption signal should be flat. Dividing the mean value of $\Delta a_{ph}$ by corresponding measured values of $a_{ph}(440)$ or $a_{ph}(675)$ gave percentage error distributions with mean values of 2.3% and 5.9%, respectively (with SD of 2.3% and 7.6%, respectively). In the collected data set, due to logistic limitations, generally no measurements were made on multiple samples. However, in the separate tests we estimated the average uncertainty of $a_{ph}$ measurements due to subsampling to be 9.1% (± 1.5%, SD).

HPLC was used to determine phytoplankton pigment concentrations; the methodology is described in detail in Meler et al. (2017b), Stoń and Kosakowska (2002) and Stoń-Egiert and Kosakowska (2005). In this work we refer mainly to the total chlorophyll $a$ concentration ($Tchla$) (defined as the sum of chlorophyll $a$, allomer and epimer, chlorophyllide $a$ and phaeophytin $a$), and to the sum of the concentrations of all accessory pigments $\Sigma C_i$, i.e. the sum of chlorophylls $b$ ($Tchlb$), chlorophylls $c$ ($Tchlc$), photosynthetic carotenoids ($PSC$) and photoprotective carotenoids ($PPC$). The precision of HPLC measurements was estimated as equal to 2.9% (±1.5%, SD), and an error related to subsampling as 9.7% (±6.4%, SD) (Stoń-Egiert et al., 2010).

The data were analysed statistically in order to characterize their variability and to find approximate empirical relationships between them. The variability of the target optical and biogeochemical quantities ranged over almost three orders of magnitude. Therefore, to assess the uncertainty of our empirical parameterizations, we applied standard arithmetic statistics of relative error, and also separate statistics of logarithmically transformed data (so-called logarithmic statistics). The exact formulas are given as a footnote to one of the tables later in the paper.

**3 Results and discussion**

**3.1 General characteristics of the data**

Figure 2 exemplifies selected spectra of $a_{ph}$ that we recorded in the Baltic Sea. Even though they were smoothed using the previously described raw data procedure, some still contain artefacts related to the noise occurring in our measurement system. These artefacts are particularly visible in the 350-400 nm range, where the accuracy of measurements is limited owing to the strong light attenuation by the glass fibre from which the filters are made, and also in the 550-650 nm range, where the absorption signal is small compared to other bands. In spite of these imperfections, 80% of these spectra exhibit the expected characteristic absorption maxima in both the blue (ca 440 nm) and red (ca 675 nm) bands. Some of the spectra in our set, however, do not show a significant increase in light absorption with increasing wavelength in the 350-440 nm range: in 20% of the spectra recorded $a_{ph}(400)/a_{ph}(440)$ is > 0.95 (in some cases as high as 1.42), mainly for samples from near the mouth of

[revised manuscript text omitted]

$$a_{ph}(\lambda) = A(\lambda) \cdot Tchla^{E(\lambda)},\hspace{4cm}(3.a)$$

Note that coefficient $A(\lambda)$ determined in this way reflects the numerical value of the light absorption coefficient $a_{ph}(\lambda)$ that the approximated relationship assigns to the case when the *Tchla* is exactly 1 mg m$^{-3}$. The coefficient $E(\lambda)$ of equation (3.a) is a dimensionless quantity, which is the exponent of the power to which the chlorophyll *a* concentration is raised. If its value is

5   <1, there is a statistical tendency for the phytoplankton absorption efficiency to decrease per unit mass of chlorophyll *a* with increasing absolute chlorophyll *a* concentration. A value of $E=1$ would mean a stable $a_{ph}$ to *Tchla* ratio, while a value of $E>1$ would imply a statistical tendency for $a_{ph}$/*Tchla* to increase with increasing *Tchla*. By performing linear regression of the logarithms of the input data, we were also able to calculate the determination coefficients $R^2$ for the approximated parameterization at the individual wavelengths of light. The parameterization coefficients *A* and *E* given by formula (3.a) can

10  be easily used to determine the specific coefficient of light absorption by phytoplankton $a_{ph}^{*}(\lambda)$ [m$^2$mg$^{-1}$] (defined as values of $a_{ph}(\lambda)$ normalized with respect to *Tchla*):

$$a_{ph}^{*}(\lambda) = A(\lambda) \cdot Tchla^{E(\lambda)-1}.\hspace{4cm}(3.b)$$

The coefficients of the approximate formula (3.a) were determined over the entire available spectral range from 350 to 700 nm with a resolution of 1 nm. Figures 4a, b and c present different variants of the spectra of coefficients $A(\lambda)$ and $E(\lambda)$,

15  along with the respective values of $R^2$. These variants represent parameterizations based on all available data (a general variant) as well as alternative parameterizations derived for data subsets relating to particular months (monthly variants). The parameterization coefficients for the general and selected monthly variants are listed in the Appendix 1 (see Tables A1 and A2). Analysis of the curves in Figures 4a and b shows that the coefficients of the monthly parameterizations differ, exhibiting larger or smaller deviations from the course of the general variant's coefficients. In the case of coefficient *A*, the differences

20  between 350-590 nm and around 675 nm are particularly conspicuous. For example, the highest values of coefficient *A* for the 440 nm band were obtained in the case of parameterizations derived for September and December-January, and the lowest for April. With regard to the spectral slope of coefficient *A* in the 350-440 nm range, the largest deviations from the typical course were recorded for December-January, April, and February. In contrast, the parameterizations obtained for March, May and October are the closest to the general variant with respect to *A*. As regards coefficient *E*, there are differences between the

25  alternative parameterizations over the entire spectral range. In the general variant, *E* changes only slightly, between 0.81 and 0.91. On the other hand, the values of *E* for the parameterizations derived for individual months are spectrally more differentiated, with more pronounced local maxima and minima. The deviations from the general case of the parameterizations are the largest for March and April (upward) and for December-January (downward). The determination coefficients $R^2$, which may initially characterize the accuracy of the absorption coefficient parameterization using formula (3.a), are relatively high

30  in the case of general variant of parameterization, i.e. no less than 0.8, over almost the entire visible light range. Lower values of $R^2$ are found only at the edges of the spectral range examined, where either the accuracy of measurements is expected to be lower (short wavelengths), or the values of the absorption coefficient are close to zero (long wavelengths). In the case of the

monthly parameterizations, only the formulas obtained for months with relatively large amounts of data take equally high values of $R^2$ (i.e. March, April, May and September). For the other months, values of $R^2$ are <0.8, at least in significant parts of the spectral range examined.

[Figure]

**Figure 4.** (a) and (b): Spectral plots of coefficients *A* and *E* of the parameterizations described by Eq. (3.a), and (c): the corresponding values of coefficient $R^2$, determined for all the data analysed and for each sampling month. The coefficients of the preliminary parameterization given in Meler et al. (2017a) are also plotted; (e) and (f) examples of coefficients *A* and *E* given by various authors (see the legend in panel (e)).

Figure 5 illustrates important aspects of the variability in magnitude and spectral shape of the absorption coefficient when certain variants of the parameterization are used to calculate it. Figure 5a illustrates the family of curves representing the specific coefficients of light absorption by phytoplankton $a_{ph}^{*}(\lambda)$ calculated according to the general variant of our new

parameterization. These curves are plotted for a few chlorophyll *a* concentrations from the 0.3-100 mg m$^{-3}$ range (corresponding more or less to the range that we recorded in the Baltic Sea). The bold line in Figure 5a outlines the spectrum calculated for *Tchla* = 1 mg m$^{-3}$ (corresponding to the numerical value of coefficient $A(\lambda)$). In addition, to better visualize the 'evolution' of the spectral shape of the predicted spectra of $a_{ph}$, another family of curves is plotted. The spectra of $a_{ph}$,

5   normalized with respect to 440 nm, are plotted in Figures 5b and f for values of *Tchla* from the same range. To provide some background, Figures 5c and d show two analogous diagrams obtained using the 'classic' parameterization developed by Bricaud et al. (1995) (although it should be mentioned that the two highest *Tchla* values – 30 and 100 mg m$^{-3}$ – generally lie beyond the range for which Bricaud et al. (1995) originally developed their parameterization). Both parameterizations, our new one and the 'classic' one according to Bricaud et al. (1995), clearly predict drops in $a_{ph}^{*}(\lambda)$ with increasing *Tchla*. But

10  where changes in spectral shapes are concerned, our general parameterization predicts significant changes only in the 600-680 nm spectral range, whereas according to Bricaud et al. (1995) the variations should occur over a much broader spectral range (see Figures 5 b and d). Both parameterizations qualitatively predict the well-known phenomenon of absorption spectra 'flattening' with increasing *Tchla* (see e.g. Morel and Bricaud 1981), but these predictions are quantitatively different. As a simplified measure of spectra 'flattening', one can analyse, for example, the changes in the ratio of $a_{ph}(440)$ to $a_{ph}(675)$: this

15  ratio is sometimes referred to as the 'colour' or 'pigment' index (see e.g. Woźniak and Ostrowska 1990 a and b; see also Bricaud et al. 1995). When the parameterization by Bricaud et al. (1995) is applied to the range of *Tchla* changes assumed here (from 0.3 to 100 mg m$^{-3}$), the colour index changes roughly threefold, i.e. it decreases from 2.69 to 0.88, the latter value signifying a greater absorption of light in the red band than in the blue. By contrast, with our new parameterization, the colour index drops by a factor of only around 1.57 (from 2.76 to 1.76). These differences indicate that the combined influence of the

20  packaging effect and the decrease in relative accessory pigment concentrations manifests itself differently in our Baltic Sea dataset than in the original oceanic dataset of Bricaud et al. (1995). Besides these differences, however, we would also like to point out clear differences that become apparent when different parameterizations matched to individual months are applied to our dataset. The next four panels in Figure 5 (e, f, g and h) show similar spectral curves for two contrasting months: April and September. The family of $a_{ph}^{*}(\lambda)$ curves plotted for April (Figure 5e) shows generally much lower values in the blue light

25  maximum, and less steep slopes around this maximum than the corresponding curves for September (Figure 5g). Also, the variability in the normalized shapes of coefficient $a_{ph}$ is greater and more complex for these two particular months (Figures 5f to h) than was the case with the general parameterization (Figure 5b). The colour index changes only by a factor of 1.16 (a drop from 2.2 to 1.9) for April, while for September the corresponding change is by a factor of 1.49 (a drop from 2.7 to 1.8). There are, moreover, differences in the 'evolution' of slopes in the short-wave part of the spectrum between these two months

30  that were not manifested by the general version of our parameterization.

[Figure]

**Figure 5.** Example spectra of the specific coefficients of light absorption by phytoplankton $a^*_{ph}(\lambda)$ (left-hand panels) and spectra of $a_{ph}$ normalized to its own value in the 440 nm band (right-hand panels) for some values of *Tchla* between 0.3 and 100 mg m$^{-3}$: (a) and (b): estimated using the general variant of the parameterization obtained in this paper; (c) and (d): estimated on the basis of the parameterization by Bricaud et al. (1995) (the grey lines on panels (c) and (d) represent chlorophyll *a*

concentrations *Tchla* that go beyond the range for which the Bricaud et al. parameterization was originally developed); (e) and (f): estimated using the variant of parameterization obtained in this paper for April data only; (g) and (h): estimated using the variant for September data only.

Distinct differences between different months can also be visualized by plotting the mass-specific absorption coefficients $a_{ph}^*$ at selected bands against chlorophyll *a* concentrations. Figures 6a and 6b illustrate such plots for 440 and 675 nm bands. Evident differences in the slopes of approximate curves for the selected four months can be seen. Although for May and March we obtain slopes of the $a_{ph}^*$ vs *Tchla* relationships relatively close to those obtained for the whole dataset, quite different values are obtained for April and September.

[Figure]

**Figure 6.** (a): Relationship between coefficient $a_{ph}^*(440)$ and the chlorophyll *a* concentration *Tchla* and its functional approximations determined in this study for all the data analysed and for selected sampling months (see the legend to panel b; the equation given in the panel represents approximations of all the data available; for coefficients obtained for particular months, see Table A2 in Appendix 1); (b): as (a) but for $a_{ph}^*(675)$; (c) and (d) plots similar to (a) and (b) but presenting examples of functional approximations determined by various authors (see the legend to panel d). The grey dots on each panel represent individual data points from our database.

**3.2.2 Two-component parameterization**

As already indicated in section 3.1, there is a noticeable variation in the proportion between *Tchla* and the concentrations of other phytoplankton pigments in particular months of the year within our dataset (Figure 3). This variability initially indicated the limitations that may crop up when the chlorophyll *a* concentration is used as the only variable for parameterizing the spectra of $a_{ph}(\lambda)$. Such limitations became clear when we recorded the differences between the parameterizations matched to the data from selected months. As a step towards improving the accuracy of $a_{ph}$ parameterization, while retaining the relative simplicity of the mathematical formalism used, we decided to search for one additional variable. Different candidates for this variable were tested: various ratios between concentrations of different groups of accessory pigment concentrations (*Tchlb*, *Tchlc*, *PSC*, *PSP*, their partial sums, and the sum *ΣCi*) and *Tchla*. As a result of these tests we found that the best for this particular purpose was the ratio of all accessory pigments to chlorophyll *a* (*ΣCi/Tchla*). The new expression that approximates $a_{ph}(\lambda)$ by treating it as a function of two variables at each light wavelength can be written as follows (more details on how the new formula was derived are given in Appendix 2):

$$a_{ph}(\lambda) = A_0(\lambda) \cdot e^{K(\lambda) \cdot \frac{\Sigma C_i}{Tchla}} \cdot Tchla^{E(\lambda)}. \tag{4}$$

The numerical coefficients of the new parameterization, i.e. $A_0(\lambda)$ [m$^2$ mg$^{-1}$] and $K(\lambda)$ [no units], are summarized in Table A3 in Appendix 1 (with a spectral resolution of 2 nm). Note that coefficient $E(\lambda)$ [no units] takes the same values as those in the general variant of the one-component parameterization. Note, too, that the product of the new coefficient $A_0(\lambda)$ and the exponential function appearing in equation (4) allows one, with the adopted value of the ratio *ΣCi/Tchla*, to calculate the value corresponding to coefficient $A(\lambda)$ from the parameterization given by formula (3.a). We define this product as:

$$A'\left(\lambda, \frac{\Sigma C_i}{Tchla}\right) = A_0(\lambda) \cdot e^{K(\lambda) \cdot \frac{\Sigma C_i}{Tchla}}. \tag{5}$$

Spectral values of the new coefficients of equation (4) (coefficients $A_0(\lambda)$ and $K(\lambda)$) are shown in Figure 7.a, and Figure 7.b illustrates the family of *A'(λ,ΣCi/Tchla)* curves plotted for some values of *ΣCi/Tchla* in our database.

[Figure]

**Figure 7.** (a): Spectral plots of coefficients $A_0$ and $K$ of the two-component parameterization described by Eq (4); (b): examples of curves representing coefficent $A'(\lambda, \Sigma C_i/Tchla)$, defined by Eq. (5), for the following values of $\Sigma C_i/Tchla$: the average value of 0.62; selected values between 0.39 and 1.15 (representing the range from the 10th percentile for the December-January period to the 90th percentile for May); the hypothetical value of 0.

As in the case of single-variable parameterizations, example families of $a_{ph}^*$ curves are now presented for the new two-component parameterization (Figure 8) for two values of $\Sigma C_i/Tchla$, i.e. 0.47 and 0.88, corresponding to the 10th and 90th percentiles from the observed distribution of that ratio. There are conspicuous differences in this respect between both the values and the shapes of the $a_{ph}^*$ spectra. As expected, the new two-component parameterization generally predicts lower values of $a_{ph}^*(\lambda)$ for lower values of $\Sigma C_i/Tchla$ than for higher ones. If we assume a low proportion of accessory pigments, i.e. for $\Sigma C_i/Tchla = 0.47$ and $Tchla$ increasing from 0.3 to 100 mg m$^{-3}$, the colour index falls from 2.69 to 1.72, i.e. by a factor of 1.57. In contrast, if we assume a higher proportion of accessory pigments (=0.88), the colour index decreases by the same factor (1.57), but from a higher starting value of 2.85, to 1.82. However, none of these differences are as distinct as those between the families of $a_{ph}^*$ curves, drawn earlier according to the one-component parameterizations obtained for particular months (see Figure 5). Generally speaking, we can expect the use of the two-component parameterization to introduce an additional 'degree of freedom' to the description of the variability of parametrized light absorption spectra. But it also seems likely that even with the new two-component parameterization it will not be possible to explain all the differences manifested

by the monthly one-component parameterizations. This intuitive expectation can be quantitatively checked by analysing in detail the estimation errors calculated for different variants of the parameterizations.

[Figure]

**Figure 8**. Example spectra of the specific coefficient of light absorption by phytoplankton $a^*_{ph}$ (panels a and b) and spectra of coefficient $a_{ph}$ normalized to its value in the 440 nm band (panels c and d), for some values of *Tchla* between 0.3 and 100 mg m$^{-3}$, estimated by the two-component parameterization Eq. (4), and for two different assumed constant values of the pigment concentration ratio $\Sigma C_i/Tchla$: 0.47 (representing the 10th percentile for all data), and 0.88 (representing the 90th percentile for all data).

**3.3 Estimation errors of the different variants of parameterizations**

We performed an extensive analysis of the errors arising out of the different variants of the proposed approximation formulas (analysis of estimation errors). Different cases were considered: the formulas were tested on the whole dataset as well as on data from particular months only. In our analyses we used both the arithmetic statistics of relative errors and also so-called logarithmic statistics, as these are generally appropriate when the variation of tested/estimated quantities spans several orders of magnitude. Below we present the most important results of these analyses.

Table 3 sets out the statistics of estimation errors when different variants of the parameterizations were tested on the whole dataset. Three scenarios were considered: first, when the general variant of the parameterization was used to calculate absorption coefficients; second, when the relevant variants of monthly parameterizations were used, depending on the month of data acquisition; and third, when the two-component parameterization was used (the values for the second and third scenarios are given in parentheses). All these results are given for nine light wavelengths chosen to cover the spectral range under consideration and include the characteristic maxima of light absorption by chlorophyll $a$. We found that the estimation errors of coefficients $a_{ph}$ obtained using the general parameterization were relatively stable from 400 to 690 nm. In this range, the systematic error according to arithmetic statistics remains at the relatively low level of 5-9%, while the statistical error varies from 34% to just over 50%. Because the general parameterization was developed using linear least-squares regression applied to the logarithms of $Tchla$ and $a_{ph}$, the systematic error according to logarithmic statistics is always equal or very close to zero. The standard error factor $x$, which enables the statistical error range according to logarithmic statistics to be determined (by multiplying or dividing by its value, see formulas given as a footnote to Table 3), varies between 1.37 and 1.52 for wavelengths from 400 to 690 nm; values are higher only at the edges of the spectral range under investigation. This means that the statistical error according to logarithmic statistics in the 400-690 nm range varies from -34% to 52%; if the entire spectral range is considered, it varies from -45% to 81%. For the second scenario of calculations done over the entire dataset, i.e. when different monthly parameterization were used on an entire dataset, the errors are only slightly lower than the previous ones. Applying logarithmic statistics to this scenario leads to a standard error factor varying from 1.34 to 1.49 in the 400-690 nm range, and taking values of ≤1.75 at the edges of this range. Hence, the statistical error according to logarithmic statistics in the 400-690 nm range varies from -33% to 49%, and from -45% to 75% if the entire spectral range is considered. In the third scenario, i.e. when the new two-component parameterization was applied to a whole dataset, we found estimation errors to lie generally between the errors of the first and second scenarios. In terms of logarithmic statistics, again, as expected, the systematic errors are close to zero, and the standard error factor in the 400-690 nm range varies from 1.35 to 1.52. A detailed comparison of the results obtained indicates that applying the two-component parameterization to a whole dataset leads to a small but noticeable reduction in the errors compared with use of the general version of the one-component parameterization (Eq. 3.a) only in the 390-530 nm and 665-685 nm spectral ranges. These are the ranges in which significant differences in the family of $A'(\lambda, \Sigma C_i/Tchla)$ curves have been observed (see Figure 7b). However, comparison of the estimation errors associated with the two-component parameterization with the scenario of using monthly variants of the one-component parameterization slightly favours the latter.

**Table 3.** Statistics of estimation errors* of coefficient $a_{ph}(\lambda)$ in selected spectral bands when the different variants of the parameterization derived in this study were applied to the entire dataset (n= 1002). The calculated values are given for three scenarios: when the general variant of the one-component parameterization was used; when variants specific to individual months were chosen (the first alternative value is given in parentheses); and when the two-component parameterization was used (the second alternative value is given in parentheses).

| $\lambda$ [nm] | Arithmetic statistics of relative error | | | Logarithmic statistics | | |
|---|---|---|---|---|---|---|
| | systematic error | statistical error | systematic error | standard error factor | statistical error | |
| | $\langle\varepsilon\rangle$ [%] | $\sigma_\varepsilon$ [%] | $\langle\varepsilon\rangle_g$ [%] | $x$ | $\sigma_+$ [%] | $\sigma_-$ [%] |
| 350 | 17.9 (16.0; 18.0) | 68 (65; 68) | 0 (-0.0; -0.1) | 1.81 (1.75; 1.82) | 81 (75; 82) | -45 (-43; -45) |
| 400 | 8.5 (7.7; 8.1) | 45 (42; 43) | 0 (-0.0; -01) | 1.51 (1.48; 1.51) | 51 (48; 51) | -34 (-33; -34) |
| 440 | 6.3 (5.2; 5.4) | 39 (34; 35) | 0 (-0.0; -0.1) | 1.42 (1.38; 1.40) | 42 (38; 40) | -30 (-28; -28) |
| 500 | 6.1 (5.2; 5.2) | 37 (34; 34) | 0 (-0.0; -0.2) | 1.42 (1.38; 1.39) | 42 (38; 40) | -29 (-28; -28) |
| 550 | 7.6 (6.7; 7.6) | 42 (39; 43) | 0 (-0.1; -0.1) | 1.47 (1.44; 1.48) | 48 (44; 48) | -32 (-31; -33) |
| 600 | 8.8 (8.1; 8.8) | 47 (46; 48) | 0 (-0.2; -0.1) | 1.52 (1.49; 1.52) | 52 (49; 52) | -34 (-33; -34) |
| 675 | 5.0 (4.2; 4.4) | 34 (31; 32) | 0 (-0.0; -0.2) | 1.37 (1.34; 1.35) | 37 (34; 36) | -27 (-25; -26) |
| 690 | 5.8 (5.4; 5.8) | 37 (37; 38) | 0 (-0.1; -0.2) | 1.40 (1.39; 1.41) | 40 (39; 41) | -29 (-28; -29) |
| 700 | 23.8 (21.8; 24.5) | 199 (193; 194) | -0.9 (-0.2; 0.8) | 1.69 (1.70; 1.74) | 69 (70; 74) | -41 (-41; -42) |

**\*) Arithmetic statistics of the relative error:**

- mean of the relative error (representing the systematic error according to arithmetic statistics):

$\langle\varepsilon\rangle = N^{-1}\sum_{i=1}^{N}\varepsilon_i$ , where $\varepsilon = \frac{(P_i - O_i)}{O_i}$, $O_i$ - observed/measured values, $P_i$ - predicted/estimated values

- the standard deviation of the relative error (representing the statistical error according to arithmetic statistics):

$\sigma_\varepsilon = \sqrt{\frac{1}{N}\left(\sum_{i=1}^{N}(\varepsilon_i - \langle\varepsilon\rangle)^2\right)}$

**Logarithmic statistics:**

- the mean logarithmic error (representing the systematic error according to logarithmic statistics):

$\langle\varepsilon\rangle_g = 10^{\langle\log(\frac{P_i}{O_i})\rangle} - 1$ , where $\langle\log\left(\frac{P_i}{O_i}\right)\rangle$ is the mean of $\log\left(\frac{P_i}{O_i}\right)$

- the standard error factor (the quantity which allows the range of statistical errors to be calculated according to logarithmic statistics):

$x = 10^{\sigma_{log}}$, where $\sigma_{log}$ is the standard deviation of the set $\log\left(\frac{P_i}{O_i}\right)$

- statistical logarithmic errors (representing the range of statistical errors according to logarithmic statistics):

$\sigma_- = \frac{1}{x} - 1$, $\sigma_+ = x - 1$

The above estimation errors were calculated over the entire available dataset. Therefore these results do not address the question of how much more accurate the results might be if different variants of parameterization were tested on data from just one particular month. Table 4 gathers some results which address the latter question. It lists results obtained for data subsets limited to four separate months, and to light wavelengths representing only selected bands where the phytoplankton absorption is relatively high. Additionally, for brevity, only certain characteristics of the logarithmic statistics are presented. We generally found that using the monthly parameterization instead of a general variant for individual months often reduces

the level of statistical error according to both arithmetic and logarithmic statistics by only a small amount, although it may have a significant impact by strongly reducing the systematic error. Applying the general variant of the parameterization to particular months may overestimate or underestimate the values of coefficients $a_{ph}$ by a few percent to as much as 30% and more in extreme situations (up to 25% for the spectral bands shown in Table 4). If we take the case of April, the general parameterization variant overestimates $a_{ph}(\lambda)$ by ca 12% to 34%, depending on the light wavelength. For September, on the other hand, $a_{ph}(\lambda)$ is underestimated by ca 2 to 9% in the majority of the visible range. In other months $a_{ph}(\lambda)$ may be overestimated in some spectral ranges and underestimated in others. In contrast, according to expectations, using appropriately matched monthly parameterizations in all of these cases enables one to eliminate the systematic error according to logarithmic statistics and to significantly reduce the systematic error of the arithmetic statistics. Applying the two-component parameterization rather than the general variant of the one-component parameterization usually leads to a reduction in the statistical errors in the vicinity of phytoplankton absorption peaks. The systematic error is reduced only in two of the months analysed (May and September), while in the others, there is no influence or even a slight increase (April and March).

**Table 4.** Selected characteristics of the estimation error logarithmic statistics calculated according to different parameterizations and for particular months: March, April, May and September (the numbers of data considered were 174, 96, 201 and 211, respectively).

| Month | Variant of parameterization $\lambda$ [nm] | one-component general systematic error $\langle \varepsilon \rangle_g$ [%] | standard error factor $x$ | one-component month specific systematic error $\langle \varepsilon \rangle_g$ [%] | standard error factor $x$ | two-component systematic error $\langle \varepsilon \rangle_g$ [%] | standard error factor $x$ |
|---|---|---|---|---|---|---|---|
| March | | | | | | | |
| | 400 | -7.4 | 1.56 | 0 | 1.54 | -9.8 | 1.54 |
| | 440 | 1.7 | 1.44 | 0 | 1.42 | -2.6 | 1.41 |
| | 500 | 0.8 | 1.45 | 0 | 1.42 | -3.3 | 1.43 |
| | 675 | -1.5 | 1.41 | 0 | 1.41 | -4.8 | 1.39 |
| April | | | | | | | |
| | 400 | 24.9 | 1.46 | 0 | 1.44 | 25.3 | 1.45 |
| | 440 | 17.3 | 1.42 | 0 | 1.39 | 17.9 | 1.39 |
| | 500 | 13.4 | 1.43 | 0 | 1.38 | 13.9 | 1.38 |
| | 675 | 11.8 | 1.33 | 0 | 1.31 | 12.2 | 1.32 |
| May | | | | | | | |
| | 400 | -5.7 | 1.47 | 0 | 1.46 | -0.6 | 1.43 |
| | 440 | -11 | 1.40 | 0 | 1.39 | -2.6 | 1.34 |
| | 500 | -9.8 | 1.39 | 0 | 1.39 | -1.7 | 1.34 |
| | 675 | -4.9 | 1.45 | 0 | 1.35 | 2.2 | 1.34 |
| September | | | | | | | |
| | 400 | -1.7 | 1.41 | 0 | 1.41 | -0.5 | 1.42 |
| | 440 | -7.5 | 1.33 | 0 | 1.31 | -5.6 | 1.33 |
| | 500 | -4.6 | 1.41 | 0 | 1.31 | -2.7 | 1.33 |
| | 675 | -8.5 | 1.35 | 0 | 1.32 | -7.0 | 1.34 |

**3.4 Comparison with selected examples of parameterizations from the literature**

So far, when discussing our own results, we have referred only to the 'classic' version of the parameterization given by Bricaud et al. (1995). Now we shall briefly compare our results with other examples of parameterizations from the literature. In two additional panels of Figure 5 (d and e) we have plotted the coefficients of the parameterization by Bricaud et al. (1995), as well as coefficients of four other variants obtained for different marine environments by different authors: Stramska et al. (2003), Matsuoka et al. (2007), Nima et al. (2016) and Churilova et al. (2017). These examples were chosen from among the many known in the literature, in order to illustrate the possible variability occurring between coefficients of different parameterizations that were originally matched to different datasets. In the case of coefficients $A$, all the spectral shapes presented in Figure 5d generally reflect the characteristic absorption maxima in the blue and red spectral ranges. Quantitatively, however, there are significant differences between these examples, the largest being in the wavelength range from about 400 to 480 nm. Interestingly, such a range of $A$ coefficient variability resembles the one we obtained with our own Baltic data when we developed separate variants of the one-component parameterization for individual months (see Figure 5a). With regard to the values of coefficients $E$, the literature examples presented in Figure 5e differ significantly from each other and all exhibit a distinct variation of values across the spectrum. According to these literature sources, coefficients $E$ can take values from less than 0.5 to even more than 1 in different spectral ranges. In our analyses we also found spectral variations of $E$ values, but only for parameterization variants that were matched to the data from separate months; on pooling all our data, we found that the resulting spectral shape of coefficient $E$ was relatively flat (with values between 0.8 and 0.9 for the general parameterization variant; see Figure 5b). The fact that the various literature parameterizations clearly differ in their coefficients $E$ can be additionally illustrated by different slopes of curves plotted on graphs showing estimated dependences of specific absorption coefficients $a_{ph}*(440)$ and $a_{ph}*(675)$ as functions of $Tchla$ (see Figures 6c and d). In addition to the examples mentioned earlier, we also plotted curves according to Staehr and Markager (2004) as examples representing a wide range of $Tchla$. Again, we would like to point out that the pattern of different slopes among literature examples resembles the differences we obtained from analysing the data for different months.

As a final aspect of this brief comparison, Figure 9 shows the main characteristics of the logarithmic statistics describing the accuracy of the formulas chosen from the literature when they were applied to calculate coefficient $a_{ph}$ of our whole dataset. This was only done for illustrative purposes – in no way was it an attempt to validate our results. As may be seen from Figure 9a, all the literature formulas compared reveal significant systematic errors when they were tested on our dataset. The systematic estimation errors of $a_{ph}(\lambda)$ in the 'classic' parameterization according to Bricaud et al. (1995) range from -57 to -21% over almost the entire spectral range. Other examples show significant systematic errors at least in some portions of the light spectrum analysed (from almost -60 to about +60% at some cases). In Figure 9a we also plotted systematic errors calculated now for our own, previous preliminary version of the Baltic Sea parameterization (Meler et al. 2017a). We now see that, apart from the UV range, values of $a_{ph}$ are generally overestimated by up to 20% and more by this earlier version of the formula. In regard to the standard error factor, we can see that only some of the literature examples in the vicinity of

phytoplankton light absorption peaks achieve similarly low values as represented by our new two-component parameterization. But since for the total estimation accuracy the contributions of both systematic and statistical errors have to be taken into account, one can expect that overall, none of the literature examples can attain the accuracy that we achieved by matching our new parameterizations to our own dataset.

[Figure]

**Figure 9.** Comparison of the main characteristics of the estimation error logarithmic statistics calculated for various parameterizations of phytoplankton absorption coefficients: (a): the mean logarithmic errors of estimation $\langle\varepsilon\rangle_g$, and (b): the standard error factor $x$. Different curves in these panels represent: the two-component parameterization from this study, the preliminary parameterization according to Meler et al. (2017a), and selected examples of parameterizations gleaned from the literature (according to Bricaud et al. (1995), Churilova et al. (2017), Nima et al. (2016), Matsuoka et al. (2007) and Stramska (2003)) (see the legend given in panel b). All these parameterizations were applied to our entire dataset.

**4 Final remarks**

The empirical material for this work was acquired in a relatively small geographical area, mainly the southern Baltic Sea. However, because it was gathered in various parts of this basin, from coastal areas to open waters, and at different times of the year, the recorded light absorption coefficients and concentrations of phytoplankton pigments have large ranges of variability, in both cases reaching almost three orders of magnitude. Based on such a dataset, it was possible to derive a number

of new variants of the parameterization of coefficient $a_{ph}$: they should be treated as simplified and practical relationships of a local character, tailored to the specifics of the target environment. The new empirical formulas include classic one-component parameterizations, where the only variable is the concentration of chlorophyll *a*. Parameterizations of this type have been developed both as a general version, i.e. one matched to all the data collected in different periods of the year, and in the form

5   of separate variants adjusted to the individual months of data collection. Importantly, we found that the coefficients of monthly variants could differ from each other very significantly, thus indirectly reflecting the annual variation in the proportions between chlorophyll *a* and other photosynthetic or photoprotective pigments. The paper also presents a new, slightly more complex form of parameterization that uses one additional variable: the ratio of the concentrations of accessory pigments to the concentration of chlorophyll *a*. With all the variants of this parameterization, spectra of coefficient $a_{ph}$ can be estimated

10  fairly simply, and with few requirements as to input data. Such estimates can be made over a wide spectral range (from 350 to 700 nm) and with a high spectral resolution (1 nm). It should be borne in mind, however, that the accuracy of such estimates is obviously limited. For example, application of the general version of the one-component parameterization to all our data covering different periods of the year understandably leads to a practically zero systematic error of this estimate, although a significant statistical error remains. The latter may be characterized by standard error factors from 1.37 to 1.51 in the vast

15  majority of the spectral ranges tested. However, since the real values of $a_{ph}$ vary in Baltic Sea conditions over almost three orders of magnitude, even an estimation accuracy such as this appears satisfactory. Our study has also shown that further improvement in the accuracy of the approximate description of $a_{ph}$ spectra is possible, at least in some applications. In the case of datasets acquired in different times of the year, such an improvement can be achieved by using either 'dynamically selected' monthly variants of parameterizations, or, when pigment composition data are available, by using the new two-component

20  parameterization. In the case of data limited to particular months, it is possible to prevent the occurrence of significant systematic errors especially by using the appropriately selected monthly parameterization. An important qualitative observation from our analyses is that the new variants of monthly parameterizations have a range of variability of coefficients similar to that between the different literature parameterizations established on the basis of data from diverse aquatic environments. This particular observation reminds us that all such parameterizations are always quite far-reaching

[revised manuscript text omitted]

**Appendix 2**

In order to derive the two-component parameterization, the relationship described earlier by formula (3.a) was treated as a first, intermediate stage in its construction (see the examples plotted at two wavelengths in Figures A1a and b). To distinguish between them, the values calculated according to formula (3.a) are now denoted $a_{ph}(\lambda)_{cal}$, whereas the actually measured values of the absorption coefficient are $a_{ph}(\lambda)_m$. In the next step, the relationship between the ratio $a_{ph}(\lambda)_{cal}/a_{ph}(\lambda)_m$ and the ratio of the sum of accessory pigments to the concentration of chlorophyll $a$ $\Sigma C_i/Tchla$ was analysed. Figures A1c and d illustrate the frequency distributions of the ratio $a_{ph}(\lambda)_{cal}/a_{ph}(\lambda)_m$, while Figures A1e and f show the relationships between the ratios $a_{ph}(\lambda)_m/a_{ph}(\lambda)_{cal}$ and $\Sigma C_i/Tchla$. Despite the large dispersion of individual data points on the latter two panels, the general tendency for the logarithm of $a_{ph}(\lambda)_m/a_{ph}(\lambda)_{cal}$ to decrease with increasing $\Sigma C_i/Tchla$ is evident. This tendency can be approximated by a linear function, which effectively allows one to establish coefficients of an approximate exponential relationship between the two ratios investigated:

[revised manuscript text omitted]

N N306 041136, financed by the Polish Ministry of Science and Higher Education in 2009-2014 (grants awarded to B.W. and J.M.), and also the project funded by the National Science Centre, Poland, entitled "Advanced research into the relationships between optical, biogeochemical and physical properties of suspended particulate matter in the southern Baltic Sea" (contract No. 2016/21/B/ST10/02381) (awarded to S.B.W.) We thank Barbara Lednicka, Monika Zabłocka, Agnieszka Zdun and other colleagues from IOPAS for their help in collecting the empirical material.

The authors declare that they have no conflict of interest.

---

## Referee Report (RR1)

Authors have done considerable efforts in improving the manuscript, **Parameterization of phytoplankton spectral absorption coefficients in the Baltic Sea: general, monthly and two-component variants of approximation formulas** by Justina Meler, especially in terms of rearranging figures, including studies in different ocean basins, as suggested. The paper is worth publishing, if the authors work at further improving the quality of the manuscript.

The manuscript still needs:

- extensive editing of English language (consider verification of language by a native speaker or a professional proof reader)
- rearrangement of text into paragraphs of a consistent size

Some necessary corrections are listed below, the authors are required to carefully follow the examples and check the entire text for similar errors.

1. Check the reference list thoroughly. References in the introduction like Bidigare et al. 1990, Reinart et al. 2004, and Mascarenhas et al. 2018 are not listed in the references.
2. When adding references, throughout the manuscript avoid the word 'see', the reader will obviously look for the reference if interested.
   The same applies when referring to figures in the text.
3. 'Case 1' waters and not 'case 1 waters'
4. Figure 4 caption mentions a sub figure (f) and legend in panel (e), but no figure (f) and a legend in (e) appears in the figure.